# Target Concrete Score Matching: A Holistic Framework for Discrete Diffusion

**Ruixiang Zhang** [1]   **Shuangfei Zhai** [1]   **Yizhe Zhang** [1]   **James Thornton** [1 2]   **Zijing Ou** [1 3]   **Joshua Susskind** [1]
**Navdeep Jaitly** [1]

## Abstract

Discrete diffusion is a promising framework for modeling and generating discrete data. In this work, we present Target Concrete Score Matching (TCSM), a novel and versatile objective for training and fine-tuning discrete diffusion models. TCSM provides a general framework with broad applicability. It supports pre-training discrete diffusion models directly from data samples, and many existing discrete diffusion approaches naturally emerge as special cases of our more general TCSM framework. Furthermore, the same TCSM objective extends to post-training of discrete diffusion models, including fine-tuning using reward functions or preference data, and distillation of knowledge from pre-trained autoregressive models. These new capabilities stem from the core idea of TCSM, estimating the concrete score of the target distribution, which resides in the original (clean) data space. This allows seamless integration with reward functions and pre-trained models, which inherently only operate in the clean data space rather than the noisy intermediate spaces of diffusion processes. Our experiments on language modeling tasks demonstrate that TCSM matches or surpasses current methods. Additionally, TCSM is versatile, applicable to both pre-training and post-training scenarios, offering greater flexibility and sample efficiency.

## 1. Introduction

Discrete diffusion models have emerged as a transformative paradigm in generative modeling, achieving remarkable success across diverse domains. Despite their advancements in closing the performance gap with autoregres-

sive (AR) models through innovative training techniques, these models still face fundamental limitations that impede their broader adoption and practical use.

The current landscape of discrete diffusion models reveals two critical shortcomings. First, existing approaches are fragmented in their theoretical foundations and training methodologies. Methods such as SEDD (Lou et al., 2024) employ denoising score entropy, while CTMC (Campbell et al., 2022) derives objectives from continuous-time Markov chains, and approaches like those in (Shi et al., 2024; Sahoo et al., 2024; Xu et al., 2024a) specialize in absorbing state diffusion models with specific assumptions. This fragmentation creates a barrier to developing unified and theoretically grounded approaches.

Second, and perhaps more significantly, current discrete diffusion models predominantly focus on pre-training, largely neglecting the crucial post-training phase that has proven essential for downstream task optimization in autoregressive models. While AR models benefit from well-established post-training techniques such as reinforcement learning with human feedback (Ziegler et al., 2019; Ouyang et al., 2022; Bai et al., 2022), direct preference optimization (Rafailov et al., 2023), and knowledge distillation (Gu et al., 2024), discrete diffusion models lack comparable capabilities. This limitation significantly restricts their practical applicability and prevents them from achieving performance parity with AR counterparts in many real-world scenarios.

**Contributions** We introduce Target Concrete Score Matching (TCSM), a novel framework for discrete diffusion models based on the concrete score (Meng et al., 2022). By operating in the clean data space, TCSM seamlessly integrates reward functions and pre-trained models while integrating pre-training and post-training. Our key contributions are:

- We develop the general TCSM framework for discrete diffusion models (Sec. 3), which provides flexibility across various diffusion formulations and model parameterization.

- We showcase the effectiveness of TCSM in pre-training contexts (Sec. 4). This includes the development of ef-

---

[1]Apple Inc.  [2]Now at Google DeepMind  [3]Imperial College London, work done during internship at Apple. Correspondence to: Ruixiang Zhang <ruixiangz@apple.com>.

*Proceedings of the $42^{nd}$ International Conference on Machine Learning*, Vancouver, Canada. PMLR 267, 2025. Copyright 2025 by the author(s).

ficient Monte Carlo estimation techniques for training discrete diffusion models directly from data samples (Sec. 4.1), methods to expedite training through the use of parametric target distribution models (Sec. 4.2), and offers a perspective for contextualizing several existing discrete diffusion methods within our framework.

• We explore the application of TCSM in various post-training scenarios (Sec. 5). This encompasses reward-guided fine-tuning for optimizing downstream tasks (Sec. 5.2), preference-based fine-tuning (Sec. 5.3), and the distillation of knowledge from pre-trained autoregressive models (Sec. 5.4).

## 2. Preliminaries

**Notation** Let $\mathcal{S} = \mathcal{X}^L$ be our discrete state space, where $\mathcal{X} = \{1, \ldots, V\}$ is the vocabulary, and $L$ is the sequence length. $\mathbf{x} := [x^1, \ldots, x^L] \in \mathcal{S}$, where $x^i \in \mathcal{X}$ is the $i$-th token in the sequence. The notation $\mathbf{x}^{\neq i}$ is used to indicate all tokens in the sequence except for the one at position $i$. When referring to a sequence with a specific token $y_i$ at position $i$, we write $[y^i, \mathbf{x}^{\neq i}] = [x^1, \ldots, x^{i-1}, y^i, x^{i+1}, \ldots, x^L]$. For any token $x \in \mathcal{X}$, we denote its one-hot vector representation as $\mathbf{e}_x \in \mathbb{R}^V$. The function $\delta(x, y)$ returns $1$ if $x = y$ and $0$ otherwise. Additionally, we designate a special mask token $\mathsf{M} \in \mathcal{X}$ to serve as an absorbing state in the discrete diffusion model.

**Continuous Time Markov Chains Model** The Continuous Time Markov Chain (CTMC) model is an $\mathcal{S}$-valued time-dependent family of random variables $(\mathbf{x}_t)_{t \in [0,1]}$ that form a Markov chain characterized by the probability transition kernel $p_{t+\Delta t|t}(\mathbf{y}|\mathbf{x}) = \delta(\mathbf{y}, \mathbf{x}) + u_t(\mathbf{y}, \mathbf{x})\Delta t + o(\Delta t)$ with the initial distribution of the process at time $t = 0$ as $p_0(\mathbf{x}_0)$. $u_t(\mathbf{y}, \mathbf{x}) : \mathcal{S} \times \mathcal{S} \to \mathbb{R}$ is called the velocity or the rate matrix, which indicate the speed at which the probability transitions between states. To make sure the transition probabilities $p_{t+\Delta t|t}(\mathbf{y}|\mathbf{x})$ are normalized, $u_t(\mathbf{y}, \mathbf{x})$ need to satisfy $u_t(\mathbf{y}, \mathbf{x}) \geq 0$ for all $\mathbf{y} \neq \mathbf{x}$ and $\sum_{\mathbf{y}} u_t(\mathbf{y}, \mathbf{x}) = 0$.

**Discrete Flow Matching** We use the discrete flow matching (Campbell et al., 2024; Gat et al., 2024) as a general framework to introduce the discrete diffusion models. Our goal is to transfer samples $\mathbf{x}_0 \sim p_0(\mathbf{x}_0)$ from a *source* distribution $p_0$ to samples $\mathbf{x}_1 \sim p_1(\mathbf{x}_1)$ from a *target* distribution $p_1$. Source and target samples can be related by means of the independent coupling $(\mathbf{x}_0, \mathbf{x}_1) \sim p_0(\mathbf{x}_0)p_1(\mathbf{x}_1)$, or associate by means of a general coupling $\pi_{0,1}(\mathbf{x}_0, \mathbf{x}_1)$. For independent coupling, common choices for the source distribution is either $p_0^{\text{unif}}(\mathbf{x}_0) = \prod_{i=1}^{L} \frac{1}{V}$, a uniform distribution over $\mathcal{S}$; and (ii) $p_0^{\text{mask}}(\mathbf{x}_0) = \prod_{i=1}^{L} \delta\{\mathsf{M}, x_0^i\}$, a delta measure concentrated on the absorbing state $\mathsf{M}$.

Similar to the continuous flow matching model (Lipman et al., 2023; Liu et al., 2023), we construct a probability path $p_t(\mathbf{x}_t)$ interpolating between $p_0$ and $p_1$. By conditioning on $\mathbf{x}_1$, we build a probability path $p_t(\mathbf{x}_t) = \mathbb{E}_{p_1(\mathbf{x}_1)} p_{t|1}(\mathbf{x}_t|\mathbf{x}_1)$. The marginal velocity $u_t(\mathbf{y}, \mathbf{x})$ generating probability path $p_t(x_t)$ can be computed by $u_t(\mathbf{y}_t, \mathbf{x}_t) = \mathbb{E}_{p_{1|t}(\mathbf{x}_1|\mathbf{x}_t)} u_t(\mathbf{y}_t, \mathbf{x}_t|\mathbf{x}_1)$, where $p_{1|t}(\mathbf{x}_1|\mathbf{x}_t) = \frac{p_1(\mathbf{x}_1)p_{t|1}(\mathbf{x}_t|\mathbf{x}_1)}{p_t(\mathbf{x}_t)}$ is the true conditional distribution predicting clean data $\mathbf{x}_1$ from noisy data $\mathbf{x}_t$, and $u_t(\mathbf{y}_t, \mathbf{x}_t|\mathbf{x}_1)$ is the conditional velocity generating $p_{t|1}(\mathbf{x}_t|\mathbf{x}_1)$.

**Training** The goal is to approximate the velocity $u_t(\mathbf{y}, \mathbf{x})$ using a neural network. We can parameterize the velocity $u_t^\theta(\mathbf{y}, \mathbf{x})$ directly, and optimize the conditional flow matching loss $\mathcal{L}_{\text{CFM}}^{\text{vel}} = \mathbb{E}_{\omega(t)p_1(\mathbf{x}_1)p_{t|1}(\mathbf{x}_t|\mathbf{x}_1)} \mathcal{D}_F \left( u_t(\mathbf{y}_t, \mathbf{x}_t), u_t^\theta(\mathbf{y}_t, \mathbf{x}_t) \right)$, where we sample time $t$ from distribution $\omega(t)$, and $\mathcal{D}_F(\mathbf{u}, \mathbf{v}) = F(\mathbf{u}) - F(\mathbf{v}) - \langle \nabla F(\mathbf{v}), \mathbf{u} - \mathbf{v} \rangle$ is the Bregman divergence with respect to the strictly convex function $F$. We also need to make sure that $u_t^\theta(\mathbf{y}_t, \mathbf{x}_t)$ satisfies the rate conditions.

As shown above, the velocity is governed by the true denoising distribution $p_{1|t}(\mathbf{x}_1|\mathbf{x}_t)$, so instead of parameterizing the velocity directly, we can use a model $p_{1|t}^\theta(\mathbf{x}_1|\mathbf{x}_t)$ to approximate $p_{1|t}(\mathbf{x}_1|\mathbf{x}_t)$ by minimizing the loss

$$\mathcal{L}_{\text{CFM}}^{\text{d}} = \mathbb{E}_{\omega(t)p_1(\mathbf{x}_1)p_{t|1}(\mathbf{x}_t|\mathbf{x}_1)} \mathbb{D}\left( p_{1|t}(\mathbf{x}_1|\mathbf{x}_t) \, \| \, p_{1|t}^\theta(\mathbf{x}_1|\mathbf{x}_t) \right) \quad (1)$$

where $\mathbb{D}(\cdot \| \cdot)$ is some statistical divergence. For example (Campbell et al., 2024) uses the KL divergence which gives rise to the cross-entropy loss $\mathbb{E}_{t, \mathbf{x}_1, \mathbf{x}_t} - \log p_{1|t}^\theta(\mathbf{x}_1|\mathbf{x}_t)$, which has been shown to be an upper bound on the negative model log-likelihood of the target data distribution. $\mathcal{L}_{\text{CFM}}^{\text{d}}$ is often called the *data-prediction* loss, as the model $p_{1|t}^\theta(\mathbf{x}_1|\mathbf{x}_t)$ is trained to predicts the clean data $\mathbf{x}_1$ from the noisy data $\mathbf{x}_t$ by aligning to the true denoising distribution $p_{1|t}(\mathbf{x}_1|\mathbf{x}_t)$.

## 3. Target Concrete Score Matching

In this section, we introduce Target Concrete Score Matching (TCSM), a novel framework for training discrete diffusion models. We first present the general formulation before exploring specific instantiations in subsequent sections.

At the heart of our approach lies the concrete score (Meng et al., 2022), which serves as a discrete analog to the continuous score function $\nabla_{\mathbf{x}} \log p(\mathbf{x})$ used in continuous diffusion models.

**Definition 3.1** (Concrete Score (Meng et al., 2022)). *Let $p(\mathbf{x})$ be any discrete distribution over $\mathcal{S}$. We denote $\mathcal{N} : \mathcal{S} \to \mathcal{S}^{K_{\mathbf{x}}}$ as the function mapping each example $\mathbf{x} \in \mathcal{S}$ to a (multi)set of neighbors, such that $\mathcal{N}(\mathbf{x}) = \{\mathbf{x}_{n_1}, \ldots, \mathbf{x}_{n_k}\}$ and $K_{\mathbf{x}} = |\mathcal{N}(\mathbf{x})|$. The neighborhood-*

| Domain | Approach | Target Object | Target Quantity |
|--------|----------|---------------|-----------------|
| Discrete | *Target* CSM (Ours) | Concrete Score of $p_1$ | $\left[\frac{p_1(\mathbf{y}_1)}{p_1(\mathbf{x}_1)}\right]_{\mathbf{y}_1 \neq \mathbf{x}_1}$ |
| Discrete | Denoising CSM (Lou et al., 2024; Meng et al., 2022) | Concrete Score of $p_{t|1}(\cdot|\mathbf{x}_1)$ | $\left[\frac{p_{t|1}(\mathbf{y}_t|\mathbf{x}_1)}{p_{t|1}(\mathbf{x}_t|\mathbf{x}_1)}\right]_{\mathbf{y}_t \neq \mathbf{x}_t}$ |
| Continuous | *Target* SM (Bortoli et al., 2024) | Score of $p_1$ | $\nabla_{\mathbf{x}_1} \log p_1(\mathbf{x}_1)$ |
| Continuous | Denoising SM (Vincent, 2011; Song et al., 2021) | Score of $p_{t|1}(\cdot|\mathbf{x}_1)$ | $\nabla_{\mathbf{x}_t} \log p_{t|1}(\mathbf{x}_t|\mathbf{x}_1)$ |

Table 1: Comparison of score matching objectives across continuous and discrete domains. The key distinction lies in whether the target quantity is derived from the clean data distribution ($p_1$) or the forward noising kernel ($p_{t|1}$). SM = Score Matching, CSM = Concrete Score Matching.

*induced graph $G$ is the directed graph which results from adding a directed edge from $\mathbf{x}$ to each node in its neighborhood set $\mathbf{x}_n \in \mathcal{N}(\mathbf{x})$, for all $\mathbf{x} \in supp(p(\mathbf{x}))$. The concrete score for a given distribution $p(\mathbf{x})$ evaluated at $\mathbf{x}$ is $\left[\frac{p(\mathbf{x}_{n_1})}{p(\mathbf{x})} - 1, \ldots, \frac{p(\mathbf{x}_{n_k})}{p(\mathbf{x})} - 1\right]^\top$. We define $\mathbf{c}_p(\mathbf{x}; \mathcal{N}) : \mathcal{S} \to \mathbb{R}^{|\mathcal{N}(\mathbf{x})|}$ by a constant shift of $\mathbf{1}$, for notational convenience.*

$$\mathbf{c}_p(\mathbf{x}; \mathcal{N}) := \left[\frac{p(\mathbf{x}_{n_1})}{p(\mathbf{x})}, \ldots, \frac{p(\mathbf{x}_{n_k})}{p(\mathbf{x})}\right]^\top. \quad (2)$$

Our approach builds upon the discrete flow matching framework (Campbell et al., 2024; Gat et al., 2024) by adopting the *data-prediction* objective in Eq. (1). This objective offers crucial flexibility, remaining valid for various model architectures and naturally supporting different probability paths without structural changes.

**Target Concrete Score Matching** We now introduce the target concrete score matching (TCSM) objective, which aims to align our model denoising distribution $p_{1|t}^\theta(\mathbf{x}_1|\mathbf{x}_t)$ with the true denoising distribution $p_{1|t}(\mathbf{x}_1|\mathbf{x}_t)$, by matching their respective concrete scores, $\mathbf{c}_{p_{1|t}^\theta}(\mathbf{x}_1; \mathcal{N}|\mathbf{x}_t)$ and $\mathbf{c}_{p_{1|t}}(\mathbf{x}_1; \mathcal{N}|\mathbf{x}_t)$. The general TCSM objective function is given by:

$$\mathcal{L}_{\text{TCSM}}(\theta; \mathcal{N}, \mathcal{D}, h) = \mathbb{E}_{\omega(t)p(\mathbf{x}_t)h(\mathbf{x}_1|\mathbf{x}_t)} \mathcal{D}\left(\mathbf{c}_{p_{1|t}}, \mathbf{c}_{p_{1|t}^\theta}\right), \quad (3)$$

where $h(\mathbf{x}_1|\mathbf{x}_t)$ serves as a proposal distribution - a probability mass function that ensures $supp(p_{1|t}(\mathbf{x}_1|\mathbf{x}_t)) \subseteq supp(h(\mathbf{x}_1|\mathbf{x}_t))$. The term $\mathcal{D}$ represents a general divergence measure that quantifies the discrepancy between the concrete scores.

**Proposition 1.** *Let $\mathcal{N}$ define a neighborhood structure that induces a weakly connected graph $G$ over the support of $p_{1|t}(\cdot|\mathbf{x}_t)$. Assuming mild regularity conditions on the divergence measure $\mathcal{D}$, the global minimum of the TCSM objective $\mathcal{L}_{TCSM}$ in Eq. (3) guarantees that $p_{1|t}^\theta(\cdot|\mathbf{x}_t)$ equals $p_{1|t}(\cdot|\mathbf{x}_t)$ almost everywhere with respect to $p(\mathbf{x}_t)$.*

*Proof.* Please refer to Sec. B.1. □

The effectiveness of our approach fundamentally relies on the connectivity of the graph $G$ induced by the neighborhood definition $\mathcal{N}$. To satisfy this requirement while offering flexible levels of granularity, we introduce a family of neighborhood structures based on Hamming distance.

**Definition 3.2** ($k$-Hamming Neighborhood). *For any sequence $\mathbf{x} \in \mathcal{S}$ and integer $k \geq 1$, the $k$-Hamming neighborhood is defined as $\mathcal{N}^k(\mathbf{x}) := \{\mathbf{y} \in \mathcal{S} \mid \text{Hamming-distance}(\mathbf{x}, \mathbf{y}) \leq k\}$, comprising all sequences that differ from $\mathbf{x}$ in at most $k$ positions.*

This family of neighborhood structures provides a flexible framework for TCSM, as $\mathcal{N}^k$ induces a weakly connected graph for any $1 \leq k \leq L$. By varying $k$, we can create a spectrum of TCSM objectives that balance local and global perspectives. The smallest neighborhood $\mathcal{N}^1$ focuses on immediate neighbors with single token differences, while $\mathcal{N}^{\text{full}} := \mathcal{N}^L$ encompasses the entire sequence space.

**TCSM with $1$-Hamming Neighborhood** When applying the TCSM framework to the 1-Hamming neighborhood - where sequences differ by at most one token - we can represent the concrete score $\mathbf{c}_p(\mathbf{x}; \mathcal{N}^1|\mathbf{x}_t)$ as a $V \times L$ matrix by replicating the original sequence $\mathbf{x}$ $L$ times, with each column $i$ defined as: $\left[\frac{p(x^i=j, \mathbf{x}^{\neq i}|\mathbf{x}_t)}{p(\mathbf{x}|\mathbf{x}_t)}\right]_{1 \leq j \leq V}^\top$. By decomposing the TCSM objective in Eq. (3) into $L$ groups based on their sequence positions, the TCSM objective can be expressed as:

$$\mathcal{L}_{\text{score}}(\theta; \mathcal{N}^1, \mathcal{D}, h) = \mathbb{E}_{\omega(t)p(\mathbf{x}_t)h(\mathbf{x}_1|\mathbf{x}_t)} \sum_{i=1}^{L} \ell_{\text{score}}^i, \quad (4)$$

$$\ell_{\text{score}}^i = \mathcal{D}\left(\left[\frac{p_{1|t}(y_1^i, \mathbf{x}_1^{\neq i}|\mathbf{x}_t)}{p_{1|t}(x_1^i, \mathbf{x}_1^{\neq i}|\mathbf{x}_t)}\right]_{y_1^i=1}^{V}, \left[\frac{p_{1|t}^\theta(y_1^i, \mathbf{x}_1^{\neq i}|\mathbf{x}_t)}{p_{1|t}^\theta(x_1^i, \mathbf{x}_1^{\neq i}|\mathbf{x}_t)}\right]_{y_1^i=1}^{V}\right).$$

This objective is termed the *score-based* TCSM ($\mathcal{L}_{\text{score}}$) as it directly operates on concrete scores. Alongside the score-based objective, we propose another objective centered on distribution matching:

$$\mathcal{L}_{\text{distrib}}(\theta; \mathcal{N}^1, \mathcal{D}, h) = \mathbb{E}_{\omega(t)p(\mathbf{x}_t)} \sum_{i=1}^{L} \mathbb{E}_{h(\mathbf{x}_1^{\neq i}|\mathbf{x}_t)} \ell_{\text{distrib}}^i, \quad (5)$$

$$\ell_{\text{distrib}}^i = \mathbb{D}\left(p_{1|t}(x_1^i|\mathbf{x}_1^{\neq i}, \mathbf{x}_t) \,\|\, p_{1|t}^\theta(x_1^i|\mathbf{x}_1^{\neq i}, \mathbf{x}_t)\right)$$

The $\mathcal{L}_{\text{distrib}}$ objective transitions from matching joint distributions $\mathbf{c}_{p_{1|t}}(\mathbf{x}_1|\mathbf{x}_t)$ via concrete score matching to aligning conditional distributions $p_{1|t}(\cdot|\mathbf{x}_1^{\neq i}, \mathbf{x}_t)$. This objective uses a statistical divergence $\mathbb{D}(\cdot \| \cdot)$ to quantify differences in probability distribution space, setting it apart from the score-based method.

The following theorem demonstrates that both $\mathcal{L}_{\text{score}}$ and $\mathcal{L}_{\text{distrib}}$ are effective for aligning the concrete score between the true distribution and the model distribution.

**Proposition 2.** *Under mild regularity conditions, the score-based objective $\mathcal{L}_{\text{score}}$ Eq. (4) achieves its global minimum if and only if the distribution-based objective $\mathcal{L}_{\text{distrib}}$ Eq. (5) achieves its global minimum, where the general TCSM objective 3 is minimized.*

*Proof.* Please refer to Sec. B.2. □

Practical implementation of $\mathcal{L}_{\text{score}}$ and $\mathcal{L}_{\text{distrib}}$ requires choosing two essential elements: the divergence metrics $\mathcal{D}(\cdot, \cdot)$ (or $\mathbb{D}(\cdot \| \cdot)$) and the proposal distribution $h(\mathbf{x}_1|\mathbf{x}_t)$. We'll explore a specific example of these choices to better understand how the score-based and distribution-based objectives are implemented and connected.

**Example: TCSM with Gen KL** Let us employ the generalized KL divergence, a specific instance of the Bregman divergence $\mathcal{D}_F(\cdot, \cdot)$ with function $F(\mathbf{u}) = \sum_j u_j \log u_j$, which takes the form $\mathcal{D}_F(\mathbf{u}, \mathbf{v}) = \sum_j u_j \log \frac{u_j}{v_j} - u_j + v_j$. To streamline our notation, let us define the ratio of conditional probabilities as $w_{1|t}^i(y) := {p_{1|t}(x_1^i=y, \mathbf{x}_1^{\neq i}|\mathbf{x}_t)}/{p_{1|t}(x_1^i, \mathbf{x}_1^{\neq i}|\mathbf{x}_t)}$ and $w_{1|t}^{i,\theta}(y) := {p_{1|t}^\theta(x_1^i=y, \mathbf{x}_1^{\neq i}|\mathbf{x}_t)}/{p_{1|t}^\theta(x_1^i, \mathbf{x}_1^{\neq i}|\mathbf{x}_t)}$. Using this notation, we can express the objective $\ell_{\text{score}}$ in Eq. (4) as:

$$\ell_{\text{score}}^i = \sum_y \left( w_{1|t}^i(y) \left[ \log \frac{w_{1|t}^i(y)}{w_{1|t}^{i,\theta}(y)} \right] - w_{1|t}^i(y) + w_{1|t}^{i,\theta}(y) \right) \quad (6)$$

**Proposition 3.** *Under the proposal distribution $h(\mathbf{x}_1|\mathbf{x}_t) = p_{1|t}(\mathbf{x}_1|\mathbf{x}_t)$, the score-based objective with generalized KL divergence is equivalent to the distribution-based objective with a weighted combination of forward KL and Itakura-Saito (IS) divergences:*

$$\mathcal{L}_{\text{score}}(\theta; h = p_{1|t}, \mathcal{D} = \mathcal{D}_{\text{GKL}}(,)) \equiv$$
$$\mathcal{L}_{\text{distrib}}(\theta; h = p_{1|t}, \mathbb{D} = V\mathbb{D}_{\text{KL}} + \mathbb{D}_{\text{IS}})$$

*where $\mathbb{D}_{\text{KL}}$ represents the forward KL divergence, and $\mathbb{D}_{\text{IS}}$ denotes the Itakura-Saito divergence.*

*Proof.* Please refer to Sec. B.3. □

This equivalence demonstrates that the score-based and distribution-based approaches yield identical optimization objective when using the true conditional distribution as the proposal and appropriate divergence measures.

| Type | Source | Div. | Param. | Model |
|------|--------|------|--------|-------|
| $\mathcal{L}_{\text{distrib}}$ | M | KL | Fact.+ | MD4/MDLM |
| $\mathcal{L}_{\text{distrib}}$ | M/U | KL | Fact. | DFM |
| $\mathcal{L}_{\text{distrib}}$ | M | $f$-div | EBM | EDLM |

Table 2: Existing discrete diffusion models under the TCSM framework with different choices of source distribution (M=Mask, U=Uniform), divergence measure, proposal ($p_{1|t}(\mathbf{x}_1|\mathbf{x}_t)$ for all), and parameterization (Fact.=Factorized, Fact.+=Factorized with carry-over, EBM=Energy-Based Model).

**Target Concrete Score** To gain more insights into the $\mathcal{L}_{\text{score}}$ and $\mathcal{L}_{\text{distrib}}$ objectives, we examine their respective targets: the concrete score ratio $\left[ \frac{p_{1|t}(\mathbf{y}_1|\mathbf{x}_t)}{p_{1|t}(\mathbf{x}_1|\mathbf{x}_t)} \right]$ and the conditional distribution $p_{1|t}(\cdot|\mathbf{x}_1^{\neq i}, \mathbf{x}_t)$.

For the score-based objective, we can decompose the target as $\left[ \frac{p_{1|t}(\mathbf{y}_1|\mathbf{x}_t)}{p_{1|t}(\mathbf{x}_1|\mathbf{x}_t)} = \frac{p_1(\mathbf{y}_1)}{p_1(\mathbf{x}_1)} \frac{p_{t|1}(\mathbf{x}_t|\mathbf{y}_1)}{p_{t|1}(\mathbf{x}_t|\mathbf{x}_1)} \right]$. This shows that $p_{1|t}(\mathbf{x}_1|\mathbf{x}_t)$'s concrete score is a weighted version of $p_1(\mathbf{x}_1)$'s concrete score, with weights from the probability path $p_{t|1}(\mathbf{x}_t|\mathbf{x}_1)$:

$$\left[ \mathbf{c}_{p_{1|t}}(\mathbf{x}_1|\mathbf{x}_t) \right]_{\mathbf{y}_1} = \left[ \mathbf{c}_{p_1}(\mathbf{x}_1) \right]_{\mathbf{y}_1} \frac{p_{t|1}(\mathbf{x}_t|\mathbf{y}_1)}{p_{t|1}(\mathbf{x}_t|\mathbf{x}_1)} \quad (7)$$

Here, $[\mathbf{c}]_{\mathbf{y}_1}$ indexes the concrete score $\mathbf{c}$ at position $\mathbf{y}_1$. The distribution-based objective reveals an analogous relationship:

$$p_{1|t}(x_1^i|\mathbf{x}_1^{\neq i}, \mathbf{x}_t) \propto p_1(x_1^i|\mathbf{x}_1^{\neq i}) p_{t|1}(\mathbf{x}_t|\mathbf{x}_1) \quad (8)$$
$$p_{1|t}(x_1^i|\mathbf{x}_1^{\neq i}, \mathbf{x}_t) = \text{Cat}\left( x_1^i; \text{softmax}\left( \log \mathbf{c}_{p_{1|t}}(x_1^i|\mathbf{x}_1^{\neq i}, \mathbf{x}_t) \right) \right)$$

Thus $p_{1|t}(\cdot|\mathbf{x}_1^{\neq i}, \mathbf{x}_t)$ constitutes a weighted transformation of $p_1(\cdot|\mathbf{x}_1^{\neq i})$ within the target distribution space. The conditional distribution $p_{1|t}(\cdot|\mathbf{x}_1^{\neq i}, \mathbf{x}_t)$ can be interpreted as a probability-normalized instance of the concrete score $\mathbf{c}_{p_{1|t}}$.

These highlight a crucial distinction between our *target concrete score matching* (TCSM) framework and traditional denoising score matching approaches (Song et al., 2021; Lou et al., 2024). Unlike denoising score matching, which operates through the lens of the noising process $p_{t|1}(\mathbf{x}_t|\mathbf{x}_1)$, TCSM directly engages with the clean data distribution $p_1$. TCSM aligns with established methodologies in continuous diffusion models (Bortoli et al., 2024). We summarize the relationships and the contrast with conventional denoising score matching objectives across both discrete and continuous domains in Table 1.

## 4. Pre-training with TCSM

Building upon the general TCSM framework in Sec. 3, we present two approaches for pre-training discrete diffusion models. First, in Sec. 4.1, we develop Monte Carlo estimation methods for the $\mathcal{L}_{\text{score}}$ and $\mathcal{L}_{\text{distrib}}$ objectives using only empirical data samples from the target distribution $p_1$. Second, in Sec. 4.2, we demonstrate how TCSM allows one

to incorporate parametric models of $p_1$ to significantly accelerate the training of discrete diffusion models.

## 4.1. TCSM with Data Samples $\mathbf{x}_1 \sim p_1$

**Problem setting** The target distribution is the true data distribution $p_1(\mathbf{x}_1) \coloneqq p_{\text{data}}(\mathbf{x}_1)$, and we only have an empirical dataset sampled from $p_{\text{data}}(\mathbf{x}_1)$. We want to match $p_{1|t}^\theta(\mathbf{x}_1|\mathbf{x}_t)$ to $p_{1|t}(\mathbf{x}_1|\mathbf{x}_t)$ with the TCSM objective.

**Score based TCSM** We begin with the score-based $\mathcal{L}_{\text{score}}$ objective introduced in Eq. (4).

**Proposition 4.** *When using forward generalized KL divergence as the discrepancy measure and setting the proposal distribution to the true conditional distribution $p_{1|t}(\mathbf{x}_1|\mathbf{x}_t)$, the score-based $\mathcal{L}_{\text{score}}$ objective in Eq. (4) can be expressed as:*

$$\ell^i_{\text{score}} = [\ell^i_{\text{pseudo}} + \ell^i_{\text{entropy}}] + C$$
$$\ell^i_{\text{pseudo}} = \left(-\log p_{1|t}^\theta(x_1^i|\mathbf{x}_1^{\neq i}, \mathbf{x}_t) + \frac{1}{V p_{1|t}^\theta(x_1^i|\mathbf{x}_1^{\neq i}, \mathbf{x}_t)}\right)$$
$$\ell^i_{\text{entropy}} = \sum_{y_1^i} \frac{1}{V} \log p_{1|t}^\theta(y_1^i|\mathbf{x}_1^{\neq i}, \mathbf{x}_t)$$

*Proof.* Please refer to Sec. B.4. $\qquad\square$

**Analysis of the Objective** The objective consists of two additive terms that serve distinct purposes. The first term, $\ell_{\text{pseudo}}$, maximizes the pseudo-likelihood of the denoising model $p_{1|t}^\theta(\mathbf{x}_1|\mathbf{x}_t)$ with respect to the data distribution. The second term, $\ell^i_{\text{entropy}} = -\mathbb{H}(\text{Uniform}(\cdot), p_{1|t}^\theta(\cdot|\mathbf{x}_1^{\neq i}, \mathbf{x}_t))$, guides the denoising model toward making more precise and confident predictions through cross-entropy maximization for $p_{1|t}^\theta(\cdot|\mathbf{x}_1^{\neq i}, \mathbf{x}_t)$. This objective provides a practical optimization objective that relies solely on samples from the joint distribution $p(\mathbf{x}_1, \mathbf{x}_t)$.

**Distribution based TCSM** For the distribution-based $\mathcal{L}_{\text{distrib}}$ objective in Eq. (5), it is straightforward to derive a simple objective when using forward KL divergence and $p_{1|t}$ as the proposal distribution. After dropping constant terms, this yields a cross-entropy based objective:

$$\ell^i_{\text{distrib}} = -\mathbb{E}_{p_{1|t}} \log p_{1|t}^\theta(x_1^i|\mathbf{x}_1^{\neq i}, \mathbf{x}_t) + C, \qquad (9)$$

where $C$ is a constant term. In contrast to the objective in Eq. (1), which maximizes the conditional joint data likelihood $\log p_{1|t}^\theta(\mathbf{x}_1|\mathbf{x}_t)$, our approach maximizes the *pseudo-likelihood* of the denoising model $\sum_i \log p_{1|t}^\theta(x_1^i|\mathbf{x}_1^{\neq i}, \mathbf{x}_t)$.

**Flexible Model Parameterization** The $\mathcal{L}_{\text{score}}$ and $\mathcal{L}_{\text{distrib}}$ objectives are versatile and can be applied regardless of the specific parameterization of $p_{1|t}^\theta(\mathbf{x}_1|\mathbf{x}_t)$. The only requirement is the efficient estimation of the conditional distribution $p_{1|t}^\theta(x_1^i|\mathbf{x}_1^{\neq i}, \mathbf{x}_t)$ during training.

**Factorized Parameterization** Following established discrete diffusion models (Gat et al., 2024; Lou et al., 2024;

Shi et al., 2024; Sahoo et al., 2024), we can further simplify our objectives by adopting a factorized parameterization: $p_{1|t}^\theta(\mathbf{x}_1|\mathbf{x}_t) = \prod_{i=1}^L p_{1|t}^\theta(x_1^i|\mathbf{x}_t)$. This leads to the following simplified $\mathcal{L}_{\text{score}}$ objective:

$$\ell^i_{\text{score}} = \left(-\log p_{1|t}^\theta(x_1^i|\mathbf{x}_t) + \frac{1}{V p_{1|t}^\theta(x_1^i|\mathbf{x}_t)}\right) \qquad (10)$$
$$+ \frac{1}{V} \sum_y \log p_{1|t}^\theta(y|\mathbf{x}_t). \qquad (11)$$

The distribution-based TCSM objective also simplifies to: $\ell^i_{\text{distrib}} = -\mathbb{E}_{p_{1|t}} \log p_{1|t}^\theta(x_1^i|\mathbf{x}_t) + C$ .

**Joint Parameterization** In Sec. 5.1, we demonstrate example of applying our framework to models that parameterize the joint distribution without factorization assumption.

The TCSM framework offers a unifying perspective, allowing several existing discrete diffusion methods, including MD4 (Shi et al., 2024), MDLM (Sahoo et al., 2024), and DFM (Gat et al., 2024), to be viewed through the lens of target concrete score estimation under specific configurations (e.g., choices of divergence, model parameterization, and probability path). This viewpoint highlights common principles while acknowledging the unique aspects of each method. We summarize these relationships and differing choices in Table 2.

**Experiments** We now empirically validate the effectiveness of using TCSM for pre-training discrete diffusion models on language modeling tasks. We measure both perplexity. We use the same transformer-based model architecture as in (Lou et al., 2024) for all experiments. See Sec. C.1 for more experimental details.

TEXT8 We conduct experiments on TEXT8 character level language modeling tasks. We adopt a factorized model parameterization for all experiments. We explored using both $\mathcal{L}_{\text{score}}$ Eq. (10) and $\mathcal{L}_{\text{distrib}}$ Eq. (9) objectives for pretraining; as well as both uniform and absorbing source distribution for pre-training. We show the results in Fig. 1b.

OPENWEBTEXT We also conduct experiments on larger scale OPENWEBTEXT dataset. We pre-train the model with factorized parameterization using $\mathcal{L}_{\text{score}}$ and $\mathcal{L}_{\text{distrib}}$ objectives. Following previous works (Lou et al., 2024; Shi et al., 2024), we evaluate the zero-shot perplexity of trained models and show the results in Table 3.

## 4.2. TCSM with Parametric Model $p_1$

Discrete diffusion models often encounter challenges such as slow convergence and reduced sample efficiency compared to autoregressive models. We show that TCSM can help to mitigate these issues by employing parametric modeling of the target distribution $p_1(\mathbf{x}_1)$.

| | Method | LAMBADA | PTB | WikiText | 1BW |
|---|---|---|---|---|---|
| AR | GPT-2 (WebText)[*] | 45.04 | 138.43 | 41.60 | 75.20 |
| | D3PM | $\leq 93.47$ | $\leq 200.82$ | $\leq 75.16$ | $\leq 138.92$ |
| CD | Plaid | $\leq 57.28$ | $\leq 142.60$ | $\leq 50.86$ | $\leq 91.12$ |
| DD-U | SEDD (Lou et al., 2024) | $\leq 65.40$ | $\leq 140.12$ | $\leq 49.60$ | $\leq 101.37$ |
| DD-U | TCSM $\mathcal{L}_{\texttt{score}}$ ( Sec. 4.2) | $\leq 63.84$ | $\leq 138.95$ | $\leq 50.73$ | $\leq 100.46$ |
| DD-U | TCSM $\mathcal{L}_{\texttt{distrib}}$ ( Sec. 4.2) | $\leq 65.29$ | $\leq 133.67$ | $\leq 46.91$ | $\leq 98.52$ |
| DD-M | SEDD (Lou et al., 2024) | $\leq 50.92$ | $\leq 114.24$ | $\leq 40.62$ | $\leq 79.29$ |
| DD-M | MD4 (Shi et al., 2024) | $\leq 48.43$ | $\leq 102.26$ | $\leq 35.90$ | $\leq 68.10$ |
| DD-M | MDLM (Sahoo et al., 2024) | $\leq 47.52$ | $\leq 95.26$ | $\leq 32.83$ | $\leq 67.01$ |
| DD-M | TCSM $\mathcal{L}_{\texttt{distrib}}$ ( Sec. 4.2) | $\leq 48.37$ | $\leq 101.85$ | $\leq 34.92$ | $\leq 68.43$ |
| DD-M | TCSM $\mathcal{L}_{\texttt{distrib}}$ ( Sec. 5.1) | $\leq 47.29$ | $\leq 96.71$ | $\leq 31.56$ | $\leq 65.82$ |

Table 3: Zero-shot unconditional perplexity ($\downarrow$) of model trained on OPENWEBTEXT dataset. [*]The GPT-2 numbers are reported for the GPT-2 checkpoint pretrained on WebText instead of OPENWEBTEXT.

**Parametric Estimation of Target Score** Building on the observation in Eq. (8) that learning $p_{1|t}(\cdot|\mathbf{x}_1^{\neq i}, \mathbf{x}_t)$ effectively reduces to learning $p_1(\cdot|\mathbf{x}_1^{\neq i})$ in the target distribution space, we can employ a dedicated neural network to parameterize $p_1(x_1^i|\mathbf{x}_1^{\neq i})$, providing an efficient estimation of $p_{1|t}(\cdot|\mathbf{x}_1^{\neq i}, \mathbf{x}_t)$. We explore following strategies for parametric estimation of $p_1(x_1^i|\mathbf{x}_1^{\neq i})$: Importantly, the learned parametric target estimation remains invariant to the choice of probability path, making it reusable across different diffusion transition kernels.

Pre-trained BERT/AR Models Unlike previous approaches operating in noisy data spaces $\mathbf{x}_t$, our method focuses exclusively on clean data at $t = 1$. This perspective creates a valuable connection between TCSM diffusion models and other models trained on *clean* data. We can leverage existing pre-trained models like BERT (Devlin et al., 2019) or autoregressive language models to estimate $p_1(x_1^i|\mathbf{x}_1^{\neq i})$. While BERT directly provides this distribution through masked token prediction, autoregressive models require marginalizing over the vocabulary: $p_1(x_1^i|\mathbf{x}_1^{\neq i}) = p_1(\mathbf{x}_1)/\sum_{y_1^i} p_1(y_1^i, \mathbf{x}_1^{\neq i})$. See Sec. 5.4 dedicated to distilling autoregressive models.

Hollow Transformer As introduced in (Sun et al., 2023), the hollow transformer employs two autoregressive Transformers per layer, one operating left-to-right and the other right-to-left. In the final layer, the representations $f(\mathbf{x}_1^{<i})$ and $f(\mathbf{x}_1^{>i})$ are combined via attention to form $f(\mathbf{x}_1^{\neq i})$, which is used to predict the missing token $x_1^i$. This architecture allows for efficient estimation of $p_1(x_1^i|\mathbf{x}_1^{\neq i})$ for all $1 \leq i \leq L$ in a single forward pass.

**Experiments** To validate the effectiveness of parametric target estimation in accelerating discrete diffusion model training, we conducted experiments on language modeling. We explore three variants of parametric models of $p_1$: (i) pre-trained transformer autoregressive model, denoted as TCSM-AR; (ii) pre-trained BERT model, denoted

as TCSM-Bert; (iii) pre-trained hollow transformer model, denoted as TCSM-Hollow. We train the model for 26 billion tokens on OPENWEBTEXT dataset and report the perplexity on validation set in Fig. 1a. We also plot validation NLL loss curves in Fig. 4. We can see that with the help of parametric $p_1$ model, the training process of discrete diffusion model is consistently faster.

## 5. Post-training with TCSM

TCSM provides a versatile framework that extends beyond pre-training to enable effective adaptation across a wide range of post-training scenarios. By utilizing the same TCSM objectives outlined in Sec. 3, we can effortlessly adapt to diverse post-training scenarios through tailored instantiations of the target distribution, divergence measure, and model parameterization. We illustrate this adaptability through four key applications: (1) fine-tuning with pre-trained models as parametric estimators of $p_{1|t}$ (Sec. 5.1), (2) reward optimization for downstream tasks (Sec. 5.2), (3) preference-based fine-tuning (Sec. 5.3), and (4) knowledge distillation from autoregressive models (Sec. 5.4).

### 5.1. TCSM Fine-tuning with a Parametric Model $p_{1|t}$

In a similar spirit to Sec. 4.2 where we have a parametric model of $p_1$, we now consider scenarios where we have a parametric model of $p_{1|t}$, such as a pre-trained discrete diffusion model. This is particularly useful for post-training applications such as weak-to-strong fine-tuning (Burns et al., 2023; Chen et al., 2024), where we can enhance a weaker $p_{1|t}$ model to a stronger one with expanded capabilities.

**Problem Setting** We consider an unknown target distribution $p_{\text{target}} := p_1(\mathbf{x}_1)$ from which we can sample. We assume access to a parametric reference model $p_{1|t}^{\text{ref}}$, such as a pre-trained discrete diffusion model, a smaller version of the same model, or a weaker version from earlier train-

| $F(r)$ in objective Eq. (12) | (i) Parameterize ratio $r_{1|t}^\theta$ by model $p_{1|t}^\theta$ | (ii) Parameterize model $p_{1|t}^\theta$ by ratio $r_{1|t}^\theta = \exp(f_\theta)$ |
|---|---|---|
| LSIF $(r-1)^2/2$ | $\mathbb{E}_{p_{1|t}^{\text{ref}}} \left( 1/2 \left( p_{1|t}^\theta/p_{1|t}^{\text{ref}} \right)^2 \right) - \mathbb{E}_{p_{1|t}} \left( p_{1|t}^\theta/p_{1|t}^{\text{ref}} \right)$ | $\mathbb{E}_{p_{1|t}^{\text{ref}}} \left( \exp(2f_\theta)/2 \right) - \mathbb{E}_{p_{1|t}} \exp(f_\theta)$ |
| BCE $r \log r - (r+1)\log(r+1)$ | $\mathbb{E}_{p_{1|t}^{\text{ref}}} \log(1 - \sigma(\log p_{1|t}^\theta/p_{1|t}^{\text{ref}})) + \mathbb{E}_{p_{1|t}} \log(\sigma(\log p_{1|t}^\theta/p_{1|t}^{\text{ref}}))$ | $\mathbb{E}_{p_{1|t}^{\text{ref}}} \log(1 - \sigma(f_\theta)) + \mathbb{E}_{p_{1|t}} \log(\sigma(f_\theta))$ |
| GEN. KL $r \log r - r$ | $\mathbb{E}_{p_{1|t}^{\text{ref}}} \left( p_{1|t}^\theta/p_{1|t}^{\text{ref}} \right) - \mathbb{E}_{p_{1|t}} \log p_{1|t}^\theta/p_{1|t}^{\text{ref}}$ | $\mathbb{E}_{p_{1|t}^{\text{ref}}} \exp(f_\theta) - \mathbb{E}_{p_{1|t}} f_\theta$ |

Table 4: Objective functions for various density ratio parameterizations and choices of $F$ as in Sec. 5.1. $\sigma(x)$ is the sigmoid function.

ing steps. The goal is to leverage $p_{1|t}^{\text{ref}}$ to learn an improved model $p_{1|t}^\theta$ that better approximates the true distribution.

**Density Ratio Estimation** Our approach leverages the reference model $p_{1|t}^{\text{ref}}$ through density ratio estimation between the true and reference distributions. Building on the $\mathcal{L}_{\text{distrib}}$ objective Eq. (5) with $\mathcal{N}^1$ neighborhood structure, we denote the density ratio as $r_{1|t}(x_1^i|\mathbf{x}_1^{\neq i}, \mathbf{x}_t) = \frac{p_{1|t}(x_1^i|x_1^{\neq i}, \mathbf{x}_t)}{p_{1|t}^{\text{ref}}(x_1^i|x_1^{\neq i}, \mathbf{x}_t)}$. Given the true density ratio $r(x_1^i|\mathbf{x}_1^{\neq i}, \mathbf{x}_t)$, we minimize the divergence $\mathbb{D}\left(p_{1|t} \| p_{1|t}^\theta\right) = \mathbb{D}_f\left(r_{1|t}p_{1|t}^{\text{ref}} \| p^\theta\right)$ to align $p_{1|t}^\theta$ with $p_{1|t}$. The core challenge thus lies in estimating $r(x_1^i|\mathbf{x}_1^{\neq i}, \mathbf{x}_t)$. We address this by parameterizing our density ratio model as $r^\phi(x_1^i|\mathbf{x}_1^{\neq i}, \mathbf{x}_t)$ and using Bregman divergence (Sugiyama et al., 2012) to estimate it:

$$\mathbb{E}_{p_{1|t}^{\text{ref}}(x_1^i|x_1^{\neq i}, \mathbf{x}_t)} \mathcal{D}_F\left(r(x_1^i|\mathbf{x}_1^{\neq i}, \mathbf{x}_t), \, r^\phi(x_1^i|\mathbf{x}_1^{\neq i}, \mathbf{x}_t)\right) \quad (12)$$

**Density Ratio Parameterization** A straightforward method involves independently parameterizing both the density ratio model $r_{1|t}^\phi(x_1^i|\mathbf{x}_1^{\neq i}, \mathbf{x}_t)$ and the denoising model $p_{1|t}^\theta(x_1^i|\mathbf{x}_1^{\neq i}, \mathbf{x}_t)$. Once the density ratio model is optimized using Bregman divergence minimization, resulting in the optimal model $r^\star(x_1^i|\mathbf{x}_1^{\neq i}, \mathbf{x}_t)$, we face the task of solving the optimization problem $\min_\theta \mathcal{D}(r^\star p^{\text{ref}}, p^\theta)$ to align $p^\theta$ with $p$. However, this two-stage process, alternating between density ratio estimation and divergence minimization can be adversarial, unstable and is difficult to converge, we discuss more in Sec. E. Instead, we propose alternative strategies with *implicit* parameterization: (i) Parameterizing the density ratio model in terms of the denoising model as $r_{1|t}^{\phi:=\theta}(x_1^i|\mathbf{x}_1^{\neq i}, \mathbf{x}_t) = \frac{p_{1|t}^\theta(x_1^i|x_1^{\neq i}, \mathbf{x}_t)}{p_{1|t}^{\text{ref}}(x_1^i|x_1^{\neq i}, \mathbf{x}_t)}$; or (ii) Parameterizing the denoising model in terms of the density ratio model as $p_{1|t}^\theta(\mathbf{x}_1|\mathbf{x}_t) = p_{1|t}^{\text{ref}}(\mathbf{x}_1|\mathbf{x}_t)r_{1|t}^{\phi:=\theta}(\mathbf{x}_1|\mathbf{x}_t)$. The equality holds when the density ratio model is optimal where $p^{\text{ref}}r^\star$ is self-normalized. To ensure that $p_{1|t}^\theta$ is always properly normalized in practice, we define $p_{1|t}^\theta(\mathbf{x}_1|\mathbf{x}_t) = p_{1|t}^{\text{ref}}(\mathbf{x}_1|\mathbf{x}_t)r_{1|t}^\theta(\mathbf{x}_1|\mathbf{x}_t)/\sum_{\mathbf{x}_1} p_{1|t}^{\text{ref}}(\mathbf{x}_1|\mathbf{x}_t)r_{1|t}^\theta(\mathbf{x}_1|\mathbf{x}_t)$. The specific objectives resulting from these parameterizations under common Bregman divergences are summarized in Table 4.

**Reference Models** With the density ratio model parameterized, we consider two specific reference models $p^{\text{ref}}$.

Weak model as reference At each optimization step $k$, we can set the reference distribution to be the previous step de-

noising distribution $p^{\text{ref}} = p_{1|t}^{\theta_{k-1}}$. The density ratio model is parameterized as $r_{1|t}^\theta(x_1^i|\mathbf{x}_1^{\neq i}, \mathbf{x}_t) = \frac{p_{1|t}^\theta(x_1^i|x_1^{\neq i}, \mathbf{x}_t)}{p_{1|t}^{\theta_{k-1}}(x_1^i|x_1^{\neq i}, \mathbf{x}_t)}$. This will give us a procedure similar to (Chen et al., 2024). Also, we can use the exponential moving average of the denoising distribution as the reference distribution, $p^{\text{ref}} = p_{1|t}^{\theta_{\text{ema}}}$.

Pre-trained model as reference We can also set the reference distribution to be a pre-trained discrete diffusion model $p_{1|t}^{\text{ref}}(\mathbf{x}_1|\mathbf{x}_t) := p_{1|t}^{\text{pre}}(\mathbf{x}_1|\mathbf{x}_t)$. We use the (ii) parameterization strategy $p_{1|t}^\theta(\mathbf{x}_1|\mathbf{x}_t) \propto p_{1|t}^{\text{pre}}(\mathbf{x}_1|\mathbf{x}_t)r_{1|t}^\theta(\mathbf{x}_1|\mathbf{x}_t)$.

**Experiments** We evaluate our TCSM post-training density ratio estimator on language modeling, focusing on parameterization strategy (ii), which uses density ratios to characterize the denoising model (strategy (i) is explored in Sec. 5.3). Using pre-trained models with $\mathcal{L}_{\text{distrib}}$ (see Sec. 4.1), we train density ratio model with three estimators (LSIF, BCE, Generalized KL), as detailed in Alg. 1. We utilize pre-trained models from Sec. 4.1 on the TEXT8 and OPENWEBTEXT datasets, and enhance them by applying the proposed density ratio estimation post-training methods. The results are presented in Fig. 1b and Table 3. The results presented in Fig. 1b and Table 3 and summarized for different Bregman divergences in Table 5 consistently improve over the baseline across all configurations, showing robustness to divergence choice. See Sec. E for further analysis and implementation details.

| Model | Perplexity ($\downarrow$) |
|---|---|
| MDLM (Sahoo et al., 2024) | 23.83 |
| EDLM NCE (Xu et al., 2024a) | 21.52 |
| TCSM BCE (Reimpl.) | 21.87 |
| TCSM LSIF | 22.10 |
| TCSM Gen KL | 21.74 |

Table 5: Comparison of perplexity scores across different Bregman divergence formulations in TCSM framework. Baseline numbers are from (Xu et al., 2024a).

### 5.2. TCSM Fine-tuning with Reward Optimization

**Problem Setting** We address the challenge of fine-tuning pre-trained discrete diffusion models for specific reward functions $R : \mathcal{S} \to \mathbb{R}$. While rewards may sometimes require learning from external feedback (Ouyang et al.,

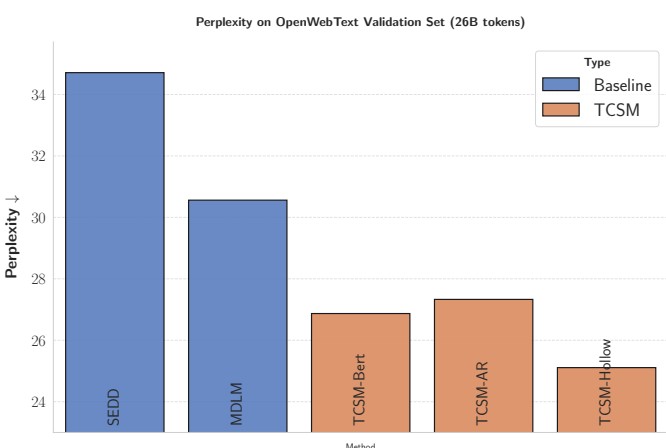

(a) Comparison of perplexity on the OPENWEBTEXT validation set after training for 26B tokens: TCSM vs. baseline models.

| Type | Method | BPC ($\downarrow$) |
|------|--------|------|
| CD | Plaid (Gulrajani & Hashimoto, 2023) | $\leq 1.48$ |
| CD | BFN (Graves et al., 2023) | $\leq 1.41$ |
| AO-AR | MAC (Shih et al., 2022) | $\leq 1.40$ |
| AR | Transformer AR (Austin et al., 2021) | **1.23** |
| DD | D3PM Uniform (Austin et al., 2021) | $\leq 1.61$ |
| DD | SEDD Uniform (Lou et al., 2024) | $\leq 1.47$ |
| DD | TCSM Uniform $\mathcal{L}_{\text{score}}$ (Sec. 4.2) | $\leq 1.47$ |
| DD | TCSM Uniform $\mathcal{L}_{\text{distrib}}$ (Sec. 4.2) | $\leq 1.45$ |
| DD | SEDD Absorb (Lou et al., 2024) | $\leq 1.39$ |
| DD | MD4 (Shi et al., 2024) | $\leq 1.37$ |
| DD | EDLM (Xu et al., 2024a) | $\leq 1.24$ |
| DD | TCSM Absorb $\mathcal{L}_{\text{score}}$ (Sec. 4.2) | $\leq 1.38$ |
| DD | TCSM Absorb $\mathcal{L}_{\text{distrib}}$ (Sec. 4.2) | $\leq 1.37$ |
| DD | TCSM Absorb $\mathcal{L}_{\text{distrib}}$ (Sec. 5.1) | $\leq 1.25$ |

(b) Bits Per Character (BPC) on TEXT8 test set. CD=Continuous Diffusion, DD=Discrete Diffusion, AR=Autoregressive, AO=Any-Order.

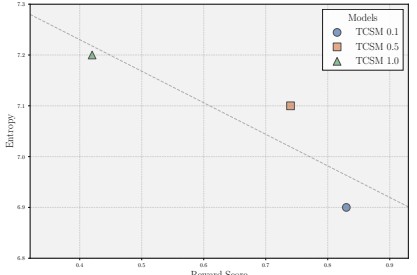

Figure 2: TCSM Reward vs. Entropy in IMDB sentiment fine-tuning (Sec. 5.3).

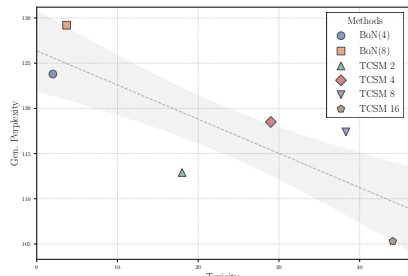

Figure 3: TCSM toxicity vs. generative perplexity in Sec. 5.2.

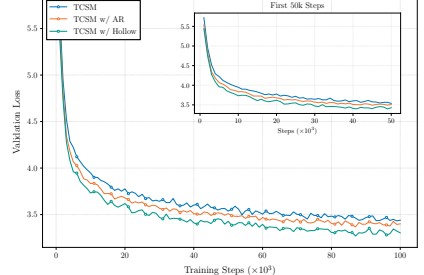

Figure 4: Validation loss curves comparing different TCSM variants on OpenWebText. Lower is better.

2022), we focus on scenarios where the reward is either explicitly known or has been successfully learned. Given a pre-trained model $p_1^{\text{pre}}(\mathbf{x}_1)$ trained on the true data distribution $p_1(\mathbf{x}_1)$, our objective is to align it with a reward-modulated target distribution: $p_{\text{target}} := p_1^R(\mathbf{x}_1) = \frac{p_1(\mathbf{x}_1) \exp(R(\mathbf{x}_1)/\beta)}{\sum_{\mathbf{x}_1} p_1(\mathbf{x}_1) \exp(R(\mathbf{x}_1)/\beta)}$, where $\beta$ controls the trade-off between reward maximization and fidelity to the original distribution. A fundamental challenge arises from the lack of ground truth samples from $p_1^R(\mathbf{x}_1)$, as we only have access to unnormalized density evaluations through the reward model.

**Reward-modulated Concrete Score** Let us analyze the score of the reward-modulated target distribution which takes the form: $p_{1|t}^R(\mathbf{x}_1|\mathbf{x}_t) \propto p_{1|t}(\mathbf{x}_1|\mathbf{x}_t) \exp(R(\mathbf{x}_1)/\beta)$. The score is given by $\frac{p_{1|t}^R(\mathbf{y}|\mathbf{x}_t)}{p_{1|t}^R(\mathbf{x}|\mathbf{x}_t)} = \frac{p_{1|t}(\mathbf{y}|\mathbf{x}_t)}{p_{1|t}(\mathbf{x}|\mathbf{x}_t)} \exp\left(\frac{R(\mathbf{y}) - R(\mathbf{x})}{\beta}\right)$ as the partition function cancels out in the ratio.

This indicates that the score of the reward-modulated

target is essentially the original score adjusted by the reward function. Given that we have a pre-trained model trained to align with the target distribution score $\left[\frac{p_{1|t}(\mathbf{y}|\mathbf{x}_t)}{p_{1|t}(\mathbf{x}|\mathbf{x}_t)}\right]$, we can approximate this using the pre-trained model as follows: $\left[\frac{p_{1|t}(\mathbf{y}|\mathbf{x}_t)}{p_{1|t}(\mathbf{x}|\mathbf{x}_t)}\right] \approx \left[\frac{p_{1|t}^{\text{pre}}(\mathbf{y}|\mathbf{x}_t)}{p_{1|t}^{\text{pre}}(\mathbf{x}|\mathbf{x}_t)}\right]$. Similarly, for the target distribution $p_{1|t}^R(x_1^i|\mathbf{x}_1^{\neq i}, \mathbf{x}_t)$ within the $\mathcal{L}_{\text{distrib}}$ objective, we have: $p_{1|t}^R(x_1^i|\mathbf{x}_1^{\neq i}, \mathbf{x}_t) \propto p_{1|t}(x_1^i|\mathbf{x}_1^{\neq i}, \mathbf{x}_t) \exp(R(x_1^i, \mathbf{x}_1^{\neq i})/\beta)$, which can also be approximated using the pre-trained model as: $p_{1|t}^R(x_1^i|\mathbf{x}_1^{\neq i}, \mathbf{x}_t) \propto p_{1|t}^{\text{pre}}(x_1^i|\mathbf{x}_1^{\neq i}, \mathbf{x}_t) \exp(R(x_1^i, \mathbf{x}_1^{\neq i})/\beta)$.

**Experiments** To validate our reward optimization methodology, we conducted experiments on both synthetic and real-world tasks: (1) a synthetic 2D grid experiment demonstrating the model's ability to effectively suppress undesired modes after fine-tuning Fig. 5 and (2) a toxicity mitigation task for language generation where our approach achieved superior performance compared to existing methods like MDLM with Best-of-N sampling, as

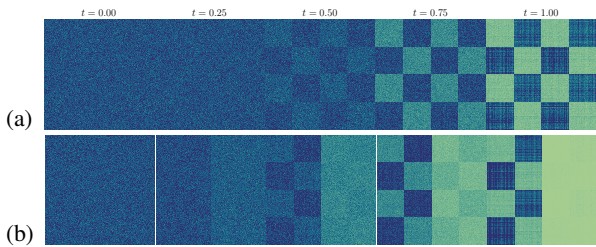

Figure 5: Model generation dynamics: sample distributions at intermediate steps, before and after reward optimization.

shown in Fig. 3. For detailed experimental settings, comprehensive results, and analysis, we refer readers to Sec. F.2 in the appendix. The complete algorithm for reward-guided training is provided in Alg. 3.

### 5.3. Direct Preference Fine-tuning

**Problem Setting** We present a method for fine-tuning pre-trained diffusion models using pairwise preference data $\{(\mathbf{q}, \mathbf{x}_1^w, \mathbf{x}_1^l)\}$, where $\mathbf{q}$ represents a query (instruction), and $\mathbf{x}_1^w$ and $\mathbf{x}_1^l$ denote preferred and non-preferred responses respectively. Our approach directly optimizes for preference alignment without requiring an explicit reward model (Rafailov et al., 2023). The target distribution focuses on preferred responses: $p_{\text{target}}(\mathbf{x}_1|\mathbf{q}) \coloneqq p_1(\mathbf{x}_1^w|\mathbf{q})$, with a pre-trained diffusion model $p_{1|t}^{\text{pre}}(\mathbf{x}_1|\mathbf{q})$ serving as our reference distribution.

**Preference Optimization** Building on the density ratio estimation framework from Sec. 5.1, we learn a new diffusion model $p_{1|t}^{\theta}$ relative to the pre-trained reference. The density ratio model is defined as: $r_{1|t}^{\theta}(x_1^i|\mathbf{x}_1^{\neq i}, \mathbf{x}_t, \mathbf{q}) = \frac{p_{1|t}^{\theta}(x_1^i|\mathbf{x}_1^{\neq i}, \mathbf{x}_t, \mathbf{q})}{p_{1|t}^{\text{pre}}(x_1^i|\mathbf{x}_1^{\neq i}, \mathbf{x}_t, \mathbf{q})}$. Optimization follows the objective in Eq. (12), with Monte Carlo estimates computed using samples $\mathbf{x}_1^w, \mathbf{x}_1^l$ drawn from the pre-trained model. Implementation details are provided in Alg. 4.

**Experiments** We validate our TCSM preference optimization approach by fine-tuning a pre-trained model on the IMDB-sentiment dataset using our density ratio estimation framework (Sec. 5.1). As shown in Fig. 2, stronger preference optimization leads to higher mean rewards but reduced sample diversity. The complete training procedure is detailed in Alg. 4, and further experimental details and results are available in the appendix (Sec. G.2).

### 5.4. AR → Diffusion distillation

**Problem setting** We explore knowledge distillation from a pre-trained autoregressive model (teacher) $p_1^{\text{AR}}(\mathbf{x}_1)$ to a diffusion model (student), where the target distribution is the teacher model's distribution $p_{\text{target}} \coloneqq p_1^{\text{AR}}(\mathbf{x}_1)$.

**Efficient estimation of distillation target** As discussed in Sec. 4.2, we can leverage pre-trained autoregressive language models to estimate $p_1(x_1^i|\mathbf{x}_1^{\neq i}) = \frac{p_1(\mathbf{x}_1)}{\sum_{x_1^i} p_1(x_1^i, \mathbf{x}_1^{\neq i})}$. However, naively computing this requires $O(VL)$ likelihood evaluations of the teacher model for each sequence $\mathbf{y} \in \mathcal{N}^1(\mathbf{x})$. While these evaluations can be parallelized, the computational cost remains prohibitive. We propose two efficient approaches to estimate the target concrete score: *Top-K* and *First-order Taylor* estimation. We leave the details to the appendix Sec. H.

**Experiments** We validate our distillation approach on the OPENWEBTEXT dataset using a transformer-based AR teacher model and an absorbing discrete diffusion student model, where our method achieves faster convergence and lower perplexity compared to baselines. See Sec. H for detailed experimental settings and further results and analysis.

## 6. Conclusion

In this work, we introduced Target Concrete Score Matching (TCSM) as a principled framework for training discrete diffusion models. By estimating the concrete score in the original data space, TCSM enables effective pre-training and seamless post-training with reward functions, preference data, and pre-trained models. Empirical results on language modeling tasks show that TCSM achieves competitive performance with greater flexibility and sample efficiency.

# Acknowledgment

We are grateful to Jiatao Gu, Tatiana Likhomanenko, Dinghuai Zhang, Richard Bai, Zijin Gu, Huangjie Zheng, Dan Busbridge, and Jason Ramapuram for their valuable insights and discussions throughout this project. We would also like to acknowledge Samy Bengio for his guidance and support.

# Impact Statement

The paper introduces a novel objective for training and fine-tuning discrete diffusion models. While discrete diffusion models have broad applicability, including language modeling and structured data generation, we do not foresee immediate ethical concerns beyond those generally associated with advancements in generative modeling, such as potential misuse for generating harmful or biased content. Responsible use and further research into mitigating such risks remain important considerations.

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

# Appendix

## Table of Contents

## A. Extended Preliminaries

**Continuous Time Markov Chains Model**   The Continuous Time Markov Chain (CTMC) model is an $\mathcal{S}$-valued time-dependent family of random variables $(\mathbf{x}_t)_{t \in [0,1]}$ that form a Markov chain characterized by the probability transition kernel $p_{t+\Delta t|t}(\mathbf{y}|\mathbf{x}) = \delta(\mathbf{y}, \mathbf{x}) + u_t(\mathbf{y}, \mathbf{x})\Delta t + o(\Delta t)$ with the initial distribution of the process at time $t = 0$ as $p_0(\mathbf{x}_0)$. $u_t(\mathbf{y}, \mathbf{x}) : \mathcal{S} \times \mathcal{S} \to \mathbb{R}$ is called the velocity or the rate matrix, which indicate the speed at which the probability transitions between states. To make sure the transition probabilities $p_{t+\Delta t|t}(\mathbf{y}|\mathbf{x})$ are normalized, $u_t(\mathbf{y}, \mathbf{x})$ need to satisfy $u_t(\mathbf{y}, \mathbf{x}) \geq 0$ for all $\mathbf{y} \neq \mathbf{x}$ and $\sum_{\mathbf{y}} u_t(\mathbf{y}, \mathbf{x}) = 0$.

**Discrete Flow Matching**   We use the discrete flow matching (Campbell et al., 2024; Gat et al., 2024) as a general framework to introduce the discrete diffusion models. Our goal is to transfer samples $\mathbf{x}_0 \sim p_0(\mathbf{x}_0)$ from a *source* distribution

$p_0$ to samples $\mathbf{x}_1 \sim p_1(\mathbf{x}_1)$ from a *target* distribution $p_1$. Source and target samples can be related by means of the independent coupling $(\mathbf{x}_0, \mathbf{x}_1) \sim p_0(\mathbf{x}_0) p_1(\mathbf{x}_1)$, or associate by means of a general coupling $\pi_{0,1}(\mathbf{x}_0, \mathbf{x}_1)$. For independent coupling, common choices for the source distribution is either $p_0^{\text{unif}}(\mathbf{x}_0) = \prod_{i=1}^{L} \frac{1}{V}$, a uniform distribution over $\mathcal{S}$; and (ii) $p_0^{\text{mask}}(\mathbf{x}_0) = \prod_{i=1}^{L} \delta\{\mathsf{M}, x_0^i\}$, a delta measure concentrated on the absorbing state $\mathsf{M}$.

Similar to the continuous flow matching model (Lipman et al., 2023; Liu et al., 2023), we construct a probability path $p_t(\mathbf{x}_t)$ interpolating between $p_0$ and $p_1$. By conditioning on $\mathbf{x}_1$, we build a probability path $p_t(\mathbf{x}_t) = \mathbb{E}_{p_1(\mathbf{x}_1)} p_{t|1}(\mathbf{x}_t|\mathbf{x}_1)$. The marginal velocity $u_t(\mathbf{y}, \mathbf{x})$ generating probability path $p_t(x_t)$ can be computed by $u_t(\mathbf{y}_t, \mathbf{x}_t) = \mathbb{E}_{p_{1|t}(\mathbf{x}_1|\mathbf{x}_t)} u_t(\mathbf{y}_t, \mathbf{x}_t|\mathbf{x}_1)$, where $p_{1|t}(\mathbf{x}_1|\mathbf{x}_t) = \frac{p_1(\mathbf{x}_1) p_{t|1}(\mathbf{x}_t|\mathbf{x}_1)}{p_t(\mathbf{x}_t)}$ is the true conditional distribution predicting clean data $\mathbf{x}_1$ from noisy data $\mathbf{x}_t$, and $u_t(\mathbf{y}_t, \mathbf{x}_t|\mathbf{x}_1)$ is the conditional velocity generating $p_{t|1}(\mathbf{x}_t|\mathbf{x}_1)$.

**Training** The goal is to approximate the velocity $u_t(\mathbf{y}, \mathbf{x})$ using a neural network. We can parameterize the velocity $u_t^\theta(\mathbf{y}, \mathbf{x})$ directly, and optimize the conditional flow matching loss $\mathcal{L}_{\text{CFM}}^{\text{vel}} = \mathbb{E}_{\omega(t) p_1(\mathbf{x}_1) p_{t|1}(\mathbf{x}_t|\mathbf{x}_1)} \mathcal{D}_F \left( u_t(\mathbf{y}_t, \mathbf{x}_t), u_t^\theta(\mathbf{y}_t, \mathbf{x}_t) \right)$, where we sample time $t$ from distribution $\omega(t)$, and $\mathcal{D}_F(\mathbf{u}, \mathbf{v}) = F(\mathbf{u}) - F(\mathbf{v}) - \langle \nabla F(\mathbf{v}), \mathbf{u} - \mathbf{v} \rangle$ is the Bregman divergence with respect to the strictly convex function $F$. We also need to make sure that $u_t^\theta(\mathbf{y}_t, \mathbf{x}_t)$ satisfies the rate conditions.

As shown above, the velocity is governed by the true denoising distribution $p_{1|t}(\mathbf{x}_1|\mathbf{x}_t)$, so instead of parameterizing the velocity directly, we can use a model $p_{t|1}^\theta(\mathbf{x}_1|\mathbf{x}_t)$ to approximate $p_{1|t}(\mathbf{x}_1|\mathbf{x}_t)$ by minimizing the loss

$$\mathcal{L}_{\text{CFM}}^{\text{d}} = \mathbb{E}_{\omega(t) p_1(\mathbf{x}_1) p_{t|1}(\mathbf{x}_t|\mathbf{x}_1)} \mathbb{D} \left( p_{1|t}(\mathbf{x}_1|\mathbf{x}_t) \,\|\, p_{t|1}^\theta(\mathbf{x}_1|\mathbf{x}_t) \right), \tag{13}$$

where $\mathbb{D}(\cdot \| \cdot)$ is some statistical divergence. For example (Campbell et al., 2024) uses the KL divergence which gives rise to the cross-entropy loss $\mathbb{E}_{t, \mathbf{x}_1, \mathbf{x}_t} - \log p_{1|t}^\theta(\mathbf{x}_1|\mathbf{x}_t)$, which has been shown to be a upper bound on the negative model log-likelihood of the target data distribution. $\mathcal{L}_{\text{CFM}}^{\text{d}}$ is often called the *data-prediction* loss, as the model $p_{1|t}^\theta(\mathbf{x}_1|\mathbf{x}_t)$ is trained to predicts the clean data $\mathbf{x}_1$ from the noisy data $\mathbf{x}_t$ by aligning to the true denoising distribution $p_{1|t}(\mathbf{x}_1|\mathbf{x}_t)$.

**Factorized Probability Paths** The flow formulation and training objective described earlier are applicable to any probability path. However, parameterizing the velocity in $\mathcal{S} \times \mathcal{S}$ is often impractical. To address this, we typically construct factorized conditional paths $p_{t|0,1}(\mathbf{x}_t|\mathbf{x}_0, \mathbf{x}_1) = \prod_{i=1}^{L} p_{t|0,1}^i(x_t^i|\mathbf{x}_0, \mathbf{x}_1)$. A common design (Gat et al., 2024; Shi et al., 2024; Sahoo et al., 2024) is

$$p_{t|0,1}^i(x_t^i|\mathbf{x}_0, \mathbf{x}_1) = \alpha_t \delta(x_t^i, x_1^i) + (1 - \alpha_t) \delta(x_t^i, x_0^i), \tag{14}$$

where $\alpha_t : \mathbb{R}_{[0,1]} \to \mathbb{R}_{[0,1]}$ is the noise schedule function. A straightforward example is the linear schedule $\alpha_t = t$. For each token $x_t^i$ sampled from $p_{t|0,1}^i(\cdot|x_0, x_1)$, there is a probability $\alpha_t$ of it being $x_1^i$ and a probability $(1 - \alpha_t)$ of it being $x_0^i$. When $\alpha_0 = 0$ and $\alpha_1 = 1$, $p_t(\mathbf{x}_t)$ adheres to the boundary conditions at $t = 0$ and $t = 1$. By marginalizing out $\mathbf{x}_0$, the conditional distribution $p_{t|1}^i(x_t^i|\mathbf{x}_1)$ have closed form as: $p_{t|1}^{\text{unif},i}(x_t^i|\mathbf{x}_1) = \text{Cat}(\alpha_t \delta\{x_t^i, x_1^i\} + (1 - \alpha_t) \frac{1}{V})$ for uniform source, $p_{t|1}^{\text{mask},i}(x_t^i|\mathbf{x}_1) = \text{Cat}(\alpha_t \delta\{x_t^i, x_1^i\} + (1 - \alpha_t) \delta\{\mathsf{M}, x_t^i\})$ for mask source. These are known as *forward transition kernel* in score-based diffusion models (Song et al., 2021), allowing for simulation-free sampling of $\mathbf{x}_t$. The corresponding velocity is given by

$$u_t^i(y^i, \mathbf{x}_t) = \mathbb{E}_{p_{1|t}^i(x_1^i|x_t^i)} \frac{\dot{\alpha}_t}{1 - \alpha_t} \left[ \delta(y^i, x_1^i) - \delta(y^i, x^i) \right], \tag{15}$$

and the marginal velocity $u_t(\mathbf{y}_t, \mathbf{x}_t)$ can be factorized as

$$u_t(\mathbf{y}_t, \mathbf{x}_t) = \sum_{i=1}^{L} \delta(\mathbf{y}_t^{\neq i}, \mathbf{x}_t^{\neq i}) u_t^i(y_t^i, \mathbf{x}_t). \tag{16}$$

So we can parameterize the factorized velocity as $u_t^{i,\theta}(y_t^i, \mathbf{x}_t)$ and optimize the loss

$$\mathcal{L}_{\text{CFM}}^{\text{v}} = \mathbb{E}_{t, \mathbf{x}_1, \mathbf{x}_t} \sum_{i=1}^{L} \mathcal{D}_F \left( u_t^i(\mathbf{y}_t^i, \mathbf{x}_t^i), u_t^{i,\theta}(\mathbf{y}_t^i, \mathbf{x}_t^i) \right), \tag{17}$$

which is also an ELBO on the target data distribution when we choose the generalized KL divergence (Nguyen et al., 2010) as the Bregman divergence (Shaul et al., 2024).

**Sampling** Sampling from the target distribution $p_1(\mathbf{x}_1)$ is achieved simulating the CTMC with learned velocity field $u_t^\theta(\mathbf{y}_t, \mathbf{x}_t)$ with Euler methods.

# B. Proofs

## B.1. Proof of Proposition 1

We first establish a key property of the Concrete score through the following lemma.

**Lemma B.1** ((Meng et al., 2022)). *Let $p(\mathbf{x})$ be a discrete probability distribution over $\mathcal{X}$. For any neighborhood structure $\mathcal{N}$ that induces a connected graph, the Concrete score mapping $\mathbf{c}_p(\mathbf{x}; \mathcal{N})$ is complete. Specifically, for any parameterized distribution $p^\theta(\mathbf{x})$ with $\theta \in \Theta$, we have $\mathbf{c}_{p^\theta}(\mathbf{x}; \mathcal{N}) = \mathbf{c}_p(\mathbf{x}; \mathcal{N})$ for all $\mathbf{x} \in \mathcal{X}$ if and only if $p^\theta(\mathbf{x}) = p(\mathbf{x})$ almost everywhere.*

*Proof.* The result follows directly from (Meng et al., 2022). We observe that our definition of $\mathbf{x}_p$ differs from the original by a constant shift of $\mathbf{1}$, which is a bijective transformation and thus preserves the completeness property. $\square$

**Proposition 1.** *Let $\mathcal{N}$ define a neighborhood structure that induces a weakly connected graph $G$ over the support of $p_{1|t}(\cdot|\mathbf{x}_t)$. Assuming mild regularity conditions on the divergence measure $\mathcal{D}$, the global minimum of the TCSM objective $\mathcal{L}_{\mathsf{TCSM}}$ in Eq. (3) guarantees that $p_{1|t}^\theta(\cdot|\mathbf{x}_t)$ equals $p_{1|t}(\cdot|\mathbf{x}_t)$ almost everywhere with respect to $p(\mathbf{x}_t)$.*

*Proof.* We prove the proposition through a bidirectional argument.

($\Rightarrow$) Let us first assume that the TCSM objective $\mathcal{L}_{\mathsf{TCSM}}$ in Eq. (3) achieves its global minimum. The objective is given by:

$$\mathcal{L}_{\mathsf{TCSM}}(\theta; \mathcal{N}, \mathcal{D}, h) = \mathbb{E}_{\omega(t)p(\mathbf{x}_t)h(\mathbf{x}_1|\mathbf{x}_t)} \mathcal{D}\left(\mathbf{c}_{p_{1|t}}, \mathbf{c}_{p_{1|t}^\theta}\right) \tag{18}$$

By construction, the proposal distribution $h(\mathbf{x}_1|\mathbf{x}_t)$ encompasses the support of $p_{1|t}(\mathbf{x}_1|\mathbf{x}_t)$. At the global minimum, we necessarily have:

$$\forall \mathbf{x}_1 \in \mathrm{supp}(p_{1|t}(\mathbf{x}_1|\mathbf{x}_t)): \quad \mathcal{D}\left(\mathbf{c}_{p_{1|t}}, \mathbf{c}_{p_{1|t}^\theta}\right) = 0$$

This implies:

$$\mathbf{c}_{p_{1|t}}(\mathbf{x}_1; \mathcal{N}) = \mathbf{c}_{p_{1|t}^\theta}(\mathbf{x}_1; \mathcal{N}).$$

Given that $\mathcal{N}$ induces a weakly connected graph over $\mathrm{supp}(p_{1|t}(\cdot|\mathbf{x}_t))$, we can apply Lemma B.1 to conclude:

$$p_{1|t}(\mathbf{x}_1|\mathbf{x}_t) = p_{1|t}^\theta(\mathbf{x}_1|\mathbf{x}_t)$$

($\Leftarrow$) For the converse, assume $p_{1|t}(\mathbf{x}_1|\mathbf{x}_t) = p_{1|t}^\theta(\mathbf{x}_1|\mathbf{x}_t)$. Since the Concrete score is a deterministic function of the underlying distribution, this equality immediately implies:

$$\mathbf{c}_{p_{1|t}}(\mathbf{x}_1; \mathcal{N}) = \mathbf{c}_{p_{1|t}^\theta}(\mathbf{x}_1; \mathcal{N})$$

Consequently, the Bregman divergence term vanishes, and the TCSM objective attains its global minimum of zero, completing the proof. $\square$

## B.2. Proof of Proposition 2

**Proposition 2.** *Under mild regularity conditions, the score-based objective $\mathcal{L}_{\mathsf{score}}$ Eq. (4) achieves its global minimum if and only if the distribution-based objective $\mathcal{L}_{\mathtt{distrib}}$ Eq. (5) achieves its global minimum, where the general TCSM objective 3 is minimized.*

*Proof.* We establish the proposition using a bidirectional approach.

($\Rightarrow$) We begin by demonstrating that if the $\mathcal{L}_{\text{score}}$ Eq. (4) reaches its global minimum, then the $\mathcal{L}_{\text{distrib}}$ Eq. (5) also attains its global minimum.

As indicated in Eq. (8), the conditional distribution $p_{1|t}(x_1^i|\mathbf{x}_1^{\neq i}, \mathbf{x}_t)$ in Eq. (5) can be expressed as:

$$p_{1|t}(x_1^i|\mathbf{x}_1^{\neq i}, \mathbf{x}_t) = \text{Cat}\left(x_1^i; \text{softmax}\left(\log \mathbf{c}_{p_{1|t}}(x_1^i|\mathbf{x}_1^{\neq i}, \mathbf{x}_t)\right)\right) \tag{19}$$

Additionally, we have:

$$\mathbf{c}_{p_{1|t}}(x_1^i|\mathbf{x}_1^{\neq i}, \mathbf{x}_t) := \left[\frac{p_{1|t}(y_1^i|\mathbf{x}_1^{\neq i}, \mathbf{x}_t)}{p_{1|t}(x_1^i|\mathbf{x}_1^{\neq i}, \mathbf{x}_t)}\right]_{y_1^i=1}^V = \left[\frac{p_{1|t}(y_1^i, \mathbf{x}_1^{\neq i}|\mathbf{x}_t)}{p_{1|t}(x_1^i, \mathbf{x}_1^{\neq i}|\mathbf{x}_t)}\right]_{y_1^i=1}^V \tag{20}$$

Therefore, when the score-based objective Eq. (4) achieves its global minimum, according to Proposition 1, we have $\mathbf{c}_{p_{1|t}}(\mathbf{x}_1|\mathbf{x}_t) = \mathbf{c}_{p_{1|t}^\theta}(\mathbf{x}_1|\mathbf{x}_t)$. By considering the $i$-th column, we obtain:

$$\mathbf{c}_{p_{1|t}}^i(\cdot|\mathbf{x}_t) := \left[\frac{p_{1|t}(y_1^i, \mathbf{x}_1^{\neq i}|\mathbf{x}_t)}{p_{1|t}(x_1^i, \mathbf{x}_1^{\neq i}|\mathbf{x}_t)}\right]_{y_1^i=1}^V \tag{21}$$

From the above three equations, it follows that when the score-based objective Eq. (4) reaches its global minimum, we have $p_{1|t}(x_1^i|\mathbf{x}_1^{\neq i}, \mathbf{x}_t) = p_{1|t}^\theta(x_1^i|\mathbf{x}_1^{\neq i}, \mathbf{x}_t)$.

($\Leftarrow$) Conversely, by combining Eq. (20) and Eq. (21), it is evident that when the distribution-based objective Eq. (5) achieves its global minimum, we have $p_{1|t}(x_1^i|\mathbf{x}_1^{\neq i}, \mathbf{x}_t) = p_{1|t}^\theta(x_1^i|\mathbf{x}_1^{\neq i}, \mathbf{x}_t)$.

$\square$

## B.3. Proof of Proposition 3

**Proposition 3.** *Under the proposal distribution $h(\mathbf{x}_1|\mathbf{x}_t) = p_{1|t}(\mathbf{x}_1|\mathbf{x}_t)$, the score-based objective with generalized KL divergence is equivalent to the distribution-based objective with a weighted combination of forward KL and Itakura-Saito (IS) divergences:*

$$\mathcal{L}_{\text{score}}(\theta; h = p_{1|t}, \mathcal{D} = \mathcal{D}_{\text{GKL}}(\,,\,)) \equiv$$
$$\mathcal{L}_{\text{distrib}}(\theta; h = p_{1|t}, \mathbb{D} = V\mathbb{D}_{\text{KL}} + \mathbb{D}_{\text{IS}})$$

*where $\mathbb{D}_{\text{KL}}$ represents the forward KL divergence, and $\mathbb{D}_{\text{IS}}$ denotes the Itakura-Saito divergence.*

*Proof.* Consider the objective function:

$$\mathcal{L}_{\text{score}}\left(\theta; \mathcal{N}^1, \mathcal{D}, h\right) = \mathbb{E}_{\omega(t)p(\mathbf{x}_t)h(\mathbf{x}_1|\mathbf{x}_t)} \sum_{i=1}^L \ell_{\text{score}}^i, \tag{22}$$

$$\ell_{\text{score}}^i = \mathcal{D}\left(\left[\frac{p_{1|t}(y_1^i, \mathbf{x}_1^{\neq i}|\mathbf{x}_t)}{p_{1|t}(x_1^i, \mathbf{x}_1^{\neq i}|\mathbf{x}_t)}\right]_{y_1^i=1}^V, \left[\frac{p_{1|t}^\theta(y_1^i, \mathbf{x}_1^{\neq i}|\mathbf{x}_t)}{p_{1|t}^\theta(x_1^i, \mathbf{x}_1^{\neq i}|\mathbf{x}_t)}\right]_{y_1^i=1}^V\right)$$

Utilizing the definition of the generalized KL divergence: $\mathcal{D}_F(\mathbf{u}, \mathbf{v}) = \sum_j u_j \log \frac{u_j}{v_j} - u_j + v_j$, we substitute this into

the objective function to obtain:

$$
\begin{aligned}
\ell^i_{\text{score}} &= \mathcal{D}_F\left(\left[\frac{p_{1|t}(y_1^i, \mathbf{x}_1^{\neq i}|\mathbf{x}_t)}{p_{1|t}(x_1^i, \mathbf{x}_1^{\neq i}|\mathbf{x}_t)}\right]^V_{y_1^i=1}, \left[\frac{p^\theta_{1|t}(y_1^i, \mathbf{x}_1^{\neq i}|\mathbf{x}_t)}{p^\theta_{1|t}(x_1^i, \mathbf{x}_1^{\neq i}|\mathbf{x}_t)}\right]^V_{y_1^i=1}\right) \\
&= \sum_{y_1^i}\left(\frac{p_{1|t}(y_1^i, \mathbf{x}_1^{\neq i}|\mathbf{x}_t)}{p_{1|t}(x_1^i, \mathbf{x}_1^{\neq i}|\mathbf{x}_t)}\left[\log\frac{p_{1|t}(y_1^i, \mathbf{x}_1^{\neq i}|\mathbf{x}_t)}{p_{1|t}(x_1^i, \mathbf{x}_1^{\neq i}|\mathbf{x}_t)} - \log\frac{p^\theta_{1|t}(y_1^i, \mathbf{x}_1^{\neq i}|\mathbf{x}_t)}{p^\theta_{1|t}(x_1^i, \mathbf{x}_1^{\neq i}|\mathbf{x}_t)}\right] - \frac{p_{1|t}(y_1^i, \mathbf{x}_1^{\neq i}|\mathbf{x}_t)}{p_{1|t}(x_1^i, \mathbf{x}_1^{\neq i}|\mathbf{x}_t)} + \frac{p^\theta_{1|t}(y_1^i, \mathbf{x}_1^{\neq i}|\mathbf{x}_t)}{p^\theta_{1|t}(x_1^i, \mathbf{x}_1^{\neq i}|\mathbf{x}_t)}\right) \\
&= \sum_{y_1^i}\left(\frac{p_{1|t}(y_1^i|\mathbf{x}_1^{\neq i}, \mathbf{x}_t)}{p_{1|t}(x_1^i|\mathbf{x}_1^{\neq i}, \mathbf{x}_t)}\left[\log\frac{p_{1|t}(y_1^i|\mathbf{x}_1^{\neq i}, \mathbf{x}_t)}{p_{1|t}(x_1^i|\mathbf{x}_1^{\neq i}, \mathbf{x}_t)} - \log\frac{p^\theta_{1|t}(y_1^i|\mathbf{x}_1^{\neq i}, \mathbf{x}_t)}{p^\theta_{1|t}(x_1^i|\mathbf{x}_1^{\neq i}, \mathbf{x}_t)}\right] - \frac{p_{1|t}(y_1^i|\mathbf{x}_1^{\neq i}, \mathbf{x}_t)}{p_{1|t}(x_1^i|\mathbf{x}_1^{\neq i}, \mathbf{x}_t)} + \frac{p^\theta_{1|t}(y_1^i|\mathbf{x}_1^{\neq i}, \mathbf{x}_t)}{p^\theta_{1|t}(x_1^i|\mathbf{x}_1^{\neq i}, \mathbf{x}_t)}\right)
\end{aligned}
$$

Given the proposal distribution $h(\mathbf{x}_1|\mathbf{x}_t) = p_{1|t}(\mathbf{x}_1|\mathbf{x}_t) = p_{1|t}(\mathbf{x}_1^{\neq l}|\mathbf{x}_t)p_{1|t}(x_1^l|\mathbf{x}_1^{\neq l}, \mathbf{x}_t)$, we have:

$$
\begin{aligned}
&\mathbb{E}_{p(\mathbf{x}_t)\,p_{1|t}(\mathbf{x}_1|\mathbf{x}_t)}\,\ell^i_{\text{score}} \\
&= \mathbb{E}_{p(\mathbf{x}_t)\,p_{1|t}(\mathbf{x}_1^{\neq i}|\mathbf{x}_t)\,p_{1|t}(x_1^i|\mathbf{x}_1^{\neq i}, \mathbf{x}_t)}\,\ell^i_{\text{score}} \\
&= \mathbb{E}\sum_{x_1^i, y_1^i}\left[p_{1|t}(y_1^i|\mathbf{x}_1^{\neq i}, \mathbf{x}_t)\left(\log\frac{p_{1|t}(y_1^i|\mathbf{x}_1^{\neq i}, \mathbf{x}_t)}{p_{1|t}(x_1^i|\mathbf{x}_1^{\neq i}, \mathbf{x}_t)} - \log\frac{p^\theta_{1|t}(y_1^i|\mathbf{x}_1^{\neq i}, \mathbf{x}_t)}{p^\theta_{1|t}(x_1^i|\mathbf{x}_1^{\neq i}, \mathbf{x}_t)}\right)\right. \\
&\qquad\qquad\qquad\left. - p_{1|t}(y_1^i|\mathbf{x}_1^{\neq i}, \mathbf{x}_t) + \frac{p_{1|t}(x_1^i|\mathbf{x}_1^{\neq i}, \mathbf{x}_t)}{p^\theta_{1|t}(x_1^i|\mathbf{x}_1^{\neq i}, \mathbf{x}_t)}p^\theta_{1|t}(y_1^i|\mathbf{x}_1^{\neq i}, \mathbf{x}_t)\right] \\
&= \mathbb{E}_{p(\mathbf{x}_t)\,p_{1|t}(\mathbf{x}_1^{\neq i}|\mathbf{x}_t)}\sum_{x_1^i}\underbrace{\mathbb{D}_{\text{KL}}\left(p_{1|t}(\cdot|\mathbf{x}_1^{\neq i}, \mathbf{x}_t)\,\|\,p^\theta_{1|t}(\cdot|\mathbf{x}_1^{\neq i}, \mathbf{x}_t)\right)}_{\mathbb{D}_{\text{KL}}(\cdot\,\|\,\cdot)} \\
&\quad + \mathbb{E}_{p(\mathbf{x}_t)\,p_{1|t}(\mathbf{x}_1^{\neq i}|\mathbf{x}_t)}\underbrace{\sum_{x_1^i}\left(-\log\frac{p_{1|t}(x_1^i|\mathbf{x}_1^{\neq i}, \mathbf{x}_t)}{p^\theta_{1|t}(x_1^i|\mathbf{x}_1^{\neq i}, \mathbf{x}_t)} - 1 + \frac{p_{1|t}(x_1^i|\mathbf{x}_1^{\neq i}, \mathbf{x}_t)}{p^\theta_{1|t}(x_1^i|\mathbf{x}_1^{\neq i}, \mathbf{x}_t)}\right)}_{\mathbb{D}_{\text{IS}}(\cdot\,\|\,\cdot)} \\
&= \mathbb{E}_{p(\mathbf{x}_t)\,p_{1|t}(\mathbf{x}_1^{\neq i}|\mathbf{x}_t)}\left[V\,\mathbb{D}_{\text{KL}}\left(p_{1|t}(\cdot|\mathbf{x}_1^{\neq i}, \mathbf{x}_t)\,\|\,p^\theta_{1|t}(\cdot|\mathbf{x}_1^{\neq i}, \mathbf{x}_t)\right) + \mathbb{D}_{\text{IS}}\left(p_{1|t}(\cdot|\mathbf{x}_1^{\neq i}, \mathbf{x}_t)\,\|\,p^\theta_{1|t}(\cdot|\mathbf{x}_1^{\neq i}, \mathbf{x}_t)\right)\right]
\end{aligned}
$$

Thus, the original objective is to minimize the KL divergence and IS divergence between $p_{1|t}(\cdot|\mathbf{x}_1^{\neq l}, \mathbf{x}_t)$ and $p^\theta_{1|t}(\cdot|\mathbf{x}_1^{\neq l}, \mathbf{x}_t)$:

$$
\mathcal{L}_{\text{score}}(\theta; h = p_{1|t}, \mathcal{D} = \mathcal{D}_{\text{GKL}}(\,,\,)) \equiv \mathcal{L}_{\text{distrib}}(\theta; h = p_{1|t}, \mathbb{D} = V\mathbb{D}_{\text{KL}} + \mathbb{D}_{\text{IS}})
$$

When we select the proposal distribution $h(\mathbf{x}_1|\mathbf{x}_t) = p_{1|t}$ and $\mathcal{D} = \mathcal{D}_{\text{GKL}}(\,,\,)$ in the score-based objective, it is equivalent to the distribution-based objective with $\mathbb{D}(\,\|\,) = V\mathbb{D}_{\text{KL}} + \mathbb{D}_{\text{IS}}$. $\qquad\square$

### B.4. Proof of Proposition 4

**Proposition 4.** *When using forward generalized KL divergence as the discrepancy measure and setting the proposal distribution to the true conditional distribution $p_{1|t}(\mathbf{x}_1|\mathbf{x}_t)$, the score-based $\mathcal{L}_{\text{score}}$ objective in Eq. (4) can be expressed as:*

$$
\begin{aligned}
\ell^i_{\text{score}} &= [\ell^i_{pseudo} + \ell^i_{entropy}] + C \\
\ell^i_{pseudo} &= \left(-\log p^\theta_{1|t}(x_1^i|\mathbf{x}_1^{\neq i}, \mathbf{x}_t) + \frac{1}{Vp^\theta_{1|t}(x_1^i|\mathbf{x}_1^{\neq i}, \mathbf{x}_t)}\right) \\
\ell^i_{entropy} &= \sum_{y_1^i}\frac{1}{V}\log p^\theta_{1|t}(y_1^i|\mathbf{x}_1^{\neq i}, \mathbf{x}_t)
\end{aligned}
$$

*Proof.* The score-based Target Concrete Score Matching ($\mathcal{L}_{\text{score}}$) objective, as defined in Eq. (4), aims to minimize the divergence between the concrete score of the true denoising distribution $p_{1|t}(\mathbf{x}_1|\mathbf{x}_t)$ and the model's denoising distribution $p^\theta_{1|t}(\mathbf{x}_1|\mathbf{x}_t)$. Proposition 3 establishes that when using the generalized KL divergence ($\mathcal{D}_{\text{GKL}}(\,,\,)$) as the discrepancy

measure $\mathcal{D}$ and the true conditional distribution $p_{1|t}(\mathbf{x}_1|\mathbf{x}_t)$ as the proposal distribution $h(\mathbf{x}_1|\mathbf{x}_t)$, the *expected* value of the $\mathcal{L}_{\text{score}}$ objective over the data distribution is equivalent to minimizing a weighted sum of the expected forward KL divergence and the Itakura-Saito (IS) divergence between the true conditional $p_{1|t}(x_1^i|\mathbf{x}_1^{\neq i}, \mathbf{x}_t)$ and the model conditional $p_{1|t}^\theta(x_1^i|\mathbf{x}_1^{\neq i}, \mathbf{x}_t)$:

$$\mathbb{E}_{\omega(t)p(\mathbf{x}_t)p_{1|t}(\mathbf{x}_1|\mathbf{x}_t)} \sum_{i=1}^{L} \ell_{\text{score}}^i[\mathcal{D}_{\mathsf{GKL}}(,)] = \mathbb{E}_{\omega(t)p(\mathbf{x}_t)p_{1|t}(\mathbf{x}_1^{\neq i}|\mathbf{x}_t)} \sum_{i=1}^{L} \left( V\mathbb{D}_{\mathsf{KL}}\left(p_{1|t}(\cdot|\dots) \,\|\, p_{1|t}^\theta(\cdot|\dots)\right) \right.$$
$$\left. + \mathbb{D}_{\mathsf{IS}}\left(p_{1|t}(\cdot|\dots) \,\|\, p_{1|t}^\theta(\cdot|\dots)\right) \right), \tag{23}$$

where $(\cdot|\dots)$ is shorthand for $(x_1^i|\mathbf{x}_1^{\neq i}, \mathbf{x}_t)$.

However, this expected loss formulation involves the true, unknown distribution $p_{1|t}$ and cannot be directly computed during training when we only have access to samples $\mathbf{x}_1 \sim p_1(\mathbf{x}_1)$ (the target data distribution). Therefore, we resort to Monte Carlo estimation, minimizing a loss function evaluated on individual samples $(t, \mathbf{x}_1, \mathbf{x}_t)$ drawn according to $\omega(t)$, $p_1(\mathbf{x}_1)$, and $p_{t|1}(\mathbf{x}_t|\mathbf{x}_1)$.

Proposition 4 presents the specific form of this practical, per-sample objective that is minimized during training. This form is particularly relevant and aligns directly with the objective derived for the common case of a *factorized model parameterization*, as detailed in Eq. (10). Under factorization, the model assumes $p_{1|t}^\theta(\mathbf{x}_1|\mathbf{x}_t) = \prod_{j=1}^{L} p_{1|t}^\theta(x_1^j|\mathbf{x}_t)$, which implies $p_{1|t}^\theta(x_1^i|\mathbf{x}_1^{\neq i}, \mathbf{x}_t) = p_{1|t}^\theta(x_1^i|\mathbf{x}_t)$. Let $q(y|\mathbf{x}_t) := p_{1|t}^\theta(y|\mathbf{x}_t)$ denote the factorized model's output distribution for any position.

The objective stated in Eq. (10) for a single sample $\mathbf{x}_1$ and position $i$ is:

$$\ell_{\text{score}}^i[\text{factorized}] = \left( -\log q(x_1^i|\mathbf{x}_t) + \frac{1}{Vq(x_1^i|\mathbf{x}_t)} \right) + \frac{1}{V} \sum_{y=1}^{V} \log q(y|\mathbf{x}_t). \tag{24}$$

Here, $x_1^i$ is the specific token at position $i$ in the sampled clean sequence $\mathbf{x}_1$.

Proposition 4 decomposes this per-sample loss into two terms:

- $\ell_{\text{pseudo}}^i = \left( -\log p_{1|t}^\theta(x_1^i|\mathbf{x}_1^{\neq i}, \mathbf{x}_t) + \frac{1}{Vp_{1|t}^\theta(x_1^i|\mathbf{x}_1^{\neq i}, \mathbf{x}_t)} \right)$
- $\ell_{\text{entropy}}^i = \sum_{y_1^i=1}^{V} \frac{1}{V} \log p_{1|t}^\theta(y_1^i|\mathbf{x}_1^{\neq i}, \mathbf{x}_t)$

When applied to the factorized model where $p_{1|t}^\theta(y_1^i|\mathbf{x}_1^{\neq i}, \mathbf{x}_t) = q(y_1^i|\mathbf{x}_t)$, these terms become:

- $\ell_{\text{pseudo}}^i = \left( -\log q(x_1^i|\mathbf{x}_t) + \frac{1}{Vq(x_1^i|\mathbf{x}_t)} \right)$
- $\ell_{\text{entropy}}^i = \frac{1}{V} \sum_{y=1}^{V} \log q(y|\mathbf{x}_t)$

Summing these two components precisely recovers the objective $\ell_{\text{score}}^i[\text{factorized}]$ given in Eq. (24).

Thus, the objective $\ell_{\text{pseudo}}^i + \ell_{\text{entropy}}^i$ as presented in Proposition 4 represents the practical, per-sample loss function derived from the $\mathcal{L}_{\text{score}}$ principle using the generalized KL divergence. It is the objective minimized via Monte Carlo estimation when training from data samples, and its structure directly corresponds to the objective used for factorized models. The constant $C$ represents terms from the full expected GKL divergence (related to the entropy of the true distribution $p_{1|t}$) that do not depend on the model parameters $\theta$ and are therefore omitted during optimization. $\qquad\square$

## C. TCSM Pre-training from data

### C.1. Experimental Details and Results

In this section, we present the experimental results obtained from our datasets, followed by a comprehensive analysis and summary of our findings at the conclusion of this section.

TEXT8   The TEXT8 dataset is a character-level text dataset featuring a limited vocabulary of 27 tokens, which includes the letters a-z and the _ whitespace token. We adhere to the standard practice of training and evaluating on TEXT8 in segments of 256 characters without any preprocessing, as outlined by Hoogeboom et al. (2021). Our experiments on the TEXT8 dataset, a compact character-level language modeling task, follow the network hyperparameters and dataset splits specified by Austin et al. (2021). We compare our results with methods that utilize models of similar size. Consistent with previous studies (Austin et al., 2021; Lou et al., 2024), we trained discrete diffusion models on TEXT8 and assessed their performance by measuring bits-per-character on the test set.

OPENWEBTEXT   To assess our approach in large-scale language modeling, we conducted extensive experiments using the OPENWEBTEXT dataset. Given that the original WebText dataset used for training GPT-2 (Radford et al., 2019) is not publicly accessible, we followed the common practice of using OPENWEBTEXT.

Our evaluation involved testing TCSM-trained discrete diffusion models against GPT-2 using zero-shot testing on five standard benchmarks: LAMBADA (Paperno et al., 2016), WikiText (Merity et al., 2017), Penn Tree Bank (PTB) (Marcus et al., 1993), and One Billion Words (LM1B). These datasets encompass a wide array of language understanding tasks and were initially employed to assess GPT-2's zero-shot perplexity performance.

For training, we utilized a batch size of 512 and a sequence length of 1024, maintaining the evaluation setup consistent with that of Lou et al. (2024).

The results indicate that TCSM significantly surpasses existing diffusion methods and closely approaches the performance of autoregressive baselines. It is important to note that our evaluation methodology slightly deviates from previous work, as we compute likelihood unconditionally without employing a sliding window, which typically results in higher perplexity values than those reported in earlier studies.

## D. TCSM Pre-training with Parametric Model $p_1$

**Experiments**   To assess the efficacy of parametric target estimation in expediting the training of discrete diffusion models, we conducted extensive experiments on language modeling tasks using the TEXT8 and OPENWEBTEXT datasets. Our empirical findings reveal substantial improvements across all proposed estimation methods.

To explore whether the parametric model $p_1$ enhances the sample efficiency of discrete diffusion model training, we employed this model to train the discrete diffusion model from scratch on the OPENWEBTEXT dataset, processing 26 billion tokens. The results of these experiments are presented in Fig. 1a.

The data clearly indicate that our TCSM framework, incorporating the parametric model $p_1$, consistently surpasses existing discrete diffusion methodologies. Notably, the hollow transformer variant (TCSM-Hollow) delivered the best performance. Both the BERT-based (TCSM-Bert) and autoregressive-based (TCSM-AR) target estimations also demonstrated strong results. These outcomes signify a significant advancement over previous diffusion methods such as SEDD and MDLM, enhancing both the learning process and sample efficiency.

The robust performance of our TCSM variants supports our hypothesis that operating within the clean target space and utilizing parametric estimation can significantly improve discrete diffusion model training. Furthermore, the results suggest that different architectural choices for target estimation present various trade-offs between performance and computational efficiency.

## E. TCSM Post-training with Parametric Model $p_{1|t}$

### E.1. Derivation of Density Ratio Estimation Objectives

This section provides a detailed derivation of the objective functions used for density ratio estimation (DRE) within the TCSM framework, as outlined in Sec. 5.1. The core idea is to estimate the ratio between the true conditional data distribution $p_{1|t}(x_1^i | \mathbf{x}_1^{\neq i}, \mathbf{x}_t)$ and a reference distribution $p_{1|t}^{\text{ref}}(x_1^i | \mathbf{x}_1^{\neq i}, \mathbf{x}_t)$, denoted by $r(x_1^i | \mathbf{x}_1^{\neq i}, \mathbf{x}_t) := \frac{p_{1|t}(x_1^i | \mathbf{x}_1^{\neq i}, \mathbf{x}_t)}{p_{1|t}^{\text{ref}}(x_1^i | \mathbf{x}_1^{\neq i}, \mathbf{x}_t)}$. We employ the Bregman divergence for this estimation task, aiming to find the parameters $\phi$ of a model $r^\phi(x_1^i | \mathbf{x}_1^{\neq i}, \mathbf{x}_t)$ that minimize the divergence to the true ratio $r$.

The general Bregman divergence objective for density ratio estimation is given by (Sugiyama et al., 2012):

$$\min_{\phi} \mathbb{E}_{p_{1|t}^{\mathrm{ref}}(x_1^i|\mathbf{x}_1^{\neq i}, \mathbf{x}_t)} \left[ \mathcal{D}_F \left( r(x_1^i|\mathbf{x}_1^{\neq i}, \mathbf{x}_t), \, r^{\phi}(x_1^i|\mathbf{x}_1^{\neq i}, \mathbf{x}_t) \right) \right], \tag{25}$$

where $F$ is a strictly convex function defining the divergence, $\mathcal{D}_F(u, v) = F(u) - F(v) - F'(v)(u - v)$.

Expanding the Bregman divergence and using the property that $\mathbb{E}_{p_{1|t}^{\mathrm{ref}}}[F'(r^{\phi})r] = \mathbb{E}_{p_{1|t}}[F'(r^{\phi})]$, we can derive a practical objective function by omitting terms independent of the model parameters $\phi$. Minimizing Eq. (25) is equivalent to minimizing:

$$\mathcal{L}_{\mathrm{DRE}}(\phi) = \mathbb{E}_{p_{1|t}^{\mathrm{ref}}(x_1^i|\dots)} \left[ F'(r^{\phi}(x_1^i|\dots))r^{\phi}(x_1^i|\dots) - F(r^{\phi}(x_1^i|\dots)) \right] - \mathbb{E}_{p_{1|t}(x_1^i|\dots)} \left[ F'(r^{\phi}(x_1^i|\dots)) \right], \tag{26}$$

where $(\dots)$ is shorthand for the conditioning variables $(\mathbf{x}_1^{\neq i}, \mathbf{x}_t)$. Note that in practice, the expectations are estimated using Monte Carlo sampling from $p_{1|t}$ (using data samples) and $p_{1|t}^{\mathrm{ref}}$ (using the reference model).

We now instantiate this general objective for the specific choices of $F$ mentioned in the main text:

**Least-Squares Importance Fitting (LSIF):** Using $F(r) = \frac{(r-1)^2}{2}$, we have $F'(r) = r - 1$. Substituting into Eq. (26):

$$\begin{aligned}
\mathcal{L}_{\mathrm{LSIF}}(\phi) &= \mathbb{E}_{p_{1|t}^{\mathrm{ref}}} \left[ (r^{\phi} - 1)r^{\phi} - \frac{(r^{\phi} - 1)^2}{2} \right] - \mathbb{E}_{p_{1|t}}[r^{\phi} - 1] \\
&= \mathbb{E}_{p_{1|t}^{\mathrm{ref}}} \left[ (r^{\phi})^2 - r^{\phi} - \frac{1}{2}((r^{\phi})^2 - 2r^{\phi} + 1) \right] - \mathbb{E}_{p_{1|t}}[r^{\phi}] + \mathrm{const.} \\
&= \mathbb{E}_{p_{1|t}^{\mathrm{ref}}} \left[ \frac{(r^{\phi})^2}{2} - \frac{1}{2} \right] - \mathbb{E}_{p_{1|t}}[r^{\phi}] + \mathrm{const.} \\
&\propto \mathbb{E}_{p_{1|t}^{\mathrm{ref}}} \left[ \frac{(r^{\phi})^2}{2} \right] - \mathbb{E}_{p_{1|t}}[r^{\phi}]. \quad \text{(Ignoring constants)}
\end{aligned}$$

**Binary Cross-Entropy (BCE) related / KL Divergence:** The objective associated with BCE often arises from $f$-divergence dual forms rather than directly from this specific $F(r)$ in the Bregman DRE literature. A common choice leading to BCE is related to the Jensen-Shannon divergence. Alternatively, considering the standard GAN objective for distinguishing $p_{1|t}$ (label 1) from $p_{1|t}^{\mathrm{ref}}$ (label 0) using a discriminator $D(x) = \sigma(\log r^{\phi}(x))$, where $\sigma(z) = 1/(1+\exp(-z))$ is the sigmoid function. Maximizing the log-likelihood $\mathbb{E}_{p_{1|t}}[\log D] + \mathbb{E}_{p_{1|t}^{\mathrm{ref}}}[\log(1 - D)]$ is equivalent to minimizing:

$$\mathcal{L}_{\mathrm{BCE-like}}(\phi) = -\mathbb{E}_{p_{1|t}}[\log(\sigma(\log r^{\phi}))] - \mathbb{E}_{p_{1|t}^{\mathrm{ref}}}[\log(1 - \sigma(\log r^{\phi}))].$$

This formulation is commonly used and corresponds to the objective derived from $F(r) = r \log r - (r + 1) \log(r + 1)$ in some DRE contexts via duality.

**Generalized Kullback-Leibler (Gen. KL):** Using $F(r) = r \log r - r$, we have $F'(r) = \log r$. Substituting into Eq. (26):

$$\begin{aligned}
\mathcal{L}_{\mathrm{GenKL}}(\phi) &= \mathbb{E}_{p_{1|t}^{\mathrm{ref}}}[(\log r^{\phi})r^{\phi} - (r^{\phi} \log r^{\phi} - r^{\phi})] - \mathbb{E}_{p_{1|t}}[\log r^{\phi}] \\
&= \mathbb{E}_{p_{1|t}^{\mathrm{ref}}}[r^{\phi} \log r^{\phi} - r^{\phi} \log r^{\phi} + r^{\phi}] - \mathbb{E}_{p_{1|t}}[\log r^{\phi}] \\
&= \mathbb{E}_{p_{1|t}^{\mathrm{ref}}}[r^{\phi}] - \mathbb{E}_{p_{1|t}}[\log r^{\phi}].
\end{aligned}$$

These objectives are summarized in Table 6.

**Implicit Parameterization Strategies**

As discussed in Sec. 5.1, we consider two main strategies for parameterizing the density ratio and the denoising model, where $\theta$ represents the parameters being optimized.

**(i) Parameterizing Ratio via Model:** Here, we set $\phi := \theta$ and define the ratio implicitly through the denoising model $p_{1|t}^{\theta}$ and the reference model $p_{1|t}^{\mathrm{ref}}$:

$$r_{1|t}^{\theta}(x_1^i|\dots) := \frac{p_{1|t}^{\theta}(x_1^i|\dots)}{p_{1|t}^{\mathrm{ref}}(x_1^i|\dots)}. \tag{27}$$

Table 6: Objective functions $\mathcal{L}_{\mathrm{DRE}}(\phi)$ derived from minimizing Eq. (26) for different Bregman divergence choices $F(r)$. Constants independent of $\phi$ are ignored.

| Method | Objective $\mathcal{L}_{\mathrm{DRE}}(\phi)$ |
|---|---|
| LSIF ($F(r) = \frac{(r-1)^2}{2}$) | $\mathbb{E}_{p_{1|t}^{\mathrm{ref}}}\left[\frac{(r^\phi)^2}{2}\right] - \mathbb{E}_{p_{1|t}}[r^\phi]$ |
| BCE-like (related to JSD/GAN) | $-\mathbb{E}_{p_{1|t}}[\log(\sigma(\log r^\phi))] - \mathbb{E}_{p_{1|t}^{\mathrm{ref}}}[\log(1 - \sigma(\log r^\phi))]$ |
| Gen. KL ($F(r) = r\log r - r$) | $\mathbb{E}_{p_{1|t}^{\mathrm{ref}}}[r^\phi] - \mathbb{E}_{p_{1|t}}[\log r^\phi]$ |

We substitute this definition of $r^\phi \equiv r^\theta$ into the objectives in Table 6. For example, the Gen. KL objective becomes $\mathbb{E}_{p_{1|t}^{\mathrm{ref}}}[p_{1|t}^\theta/p_{1|t}^{\mathrm{ref}}] - \mathbb{E}_{p_{1|t}}[\log(p_{1|t}^\theta/p_{1|t}^{\mathrm{ref}})]$.

**(ii) Parameterizing Model via Ratio:** Here, we directly parameterize the ratio, typically ensuring non-negativity, e.g., $r_{1|t}^\theta(x_1^i|\dots) = \exp(f_\theta(x_1^i|\dots))$, where $f_\theta$ is a neural network parameterized by $\theta$. The denoising model is then implicitly defined (up to normalization) as $p_{1|t}^\theta(x_1^i|\dots) \propto p_{1|t}^{\mathrm{ref}}(x_1^i|\dots)r_{1|t}^\theta(x_1^i|\dots)$. The optimization minimizes the DRE objectives from Table 6 with $r^\phi \equiv r^\theta = \exp(f_\theta)$. For instance, the Gen. KL objective becomes $\mathbb{E}_{p_{1|t}^{\mathrm{ref}}}[\exp(f_\theta)] - \mathbb{E}_{p_{1|t}}[f_\theta]$.

The resulting objectives for both strategies and all three choices of $F$ are compiled in Table 7, which mirrors Table 4 in the main text for consistency.

Table 7: Final objective functions for TCSM post-training via DRE under different Bregman divergences $F(r)$ and parameterization strategies. Here $f_\theta = \log r_{1|t}^\theta$, where $r_{1|t}^\theta$ is the parameterized ratio (explicit in (ii), implicit in (i)), and $\sigma(x)$ is the sigmoid function.

| $F(r)$ | Strategy (i) Objective: $r^\theta = p_{1|t}^\theta/p_{1|t}^{\mathrm{ref}}$ | Strategy (ii) Objective: $p_{1|t}^\theta \propto p_{1|t}^{\mathrm{ref}}\exp(f_\theta)$ |
|---|---|---|
| LSIF: $(r-1)^2/2$ | $\mathbb{E}_{p_{1|t}^{\mathrm{ref}}}\left[\frac{1}{2}\left(p_{1|t}^\theta/p_{1|t}^{\mathrm{ref}}\right)^2\right] - \mathbb{E}_{p_{1|t}}\left[p_{1|t}^\theta/p_{1|t}^{\mathrm{ref}}\right]$ | $\mathbb{E}_{p_{1|t}^{\mathrm{ref}}}\left[\frac{\exp(2f_\theta)}{2}\right] - \mathbb{E}_{p_{1|t}}[\exp(f_\theta)]$ |
| BCE-like: $r\log r - (r+1)\log(r+1)$ | $-\mathbb{E}_{p_{1|t}}[\log(\sigma(\log p_{1|t}^\theta/p_{1|t}^{\mathrm{ref}}))] - \mathbb{E}_{p_{1|t}^{\mathrm{ref}}}[\log(1 - \sigma(\log p_{1|t}^\theta/p_{1|t}^{\mathrm{ref}}))]$ | $-\mathbb{E}_{p_{1|t}}[\log(\sigma(f_\theta))] - \mathbb{E}_{p_{1|t}^{\mathrm{ref}}}[\log(1 - \sigma(f_\theta))]$ |
| Gen. KL: $r\log r - r$ | $\mathbb{E}_{p_{1|t}^{\mathrm{ref}}}\left[p_{1|t}^\theta/p_{1|t}^{\mathrm{ref}}\right] - \mathbb{E}_{p_{1|t}}[\log p_{1|t}^\theta/p_{1|t}^{\mathrm{ref}}]$ | $\mathbb{E}_{p_{1|t}^{\mathrm{ref}}}[\exp(f_\theta)] - \mathbb{E}_{p_{1|t}}[f_\theta]$ |

### E.2. Connections to $f$-divergence TCSM

A straightforward method involves independently parameterizing both the density ratio model $r_{1|t}^\phi(\mathbf{x}_1|\mathbf{x}_t)$ and the denoising model $p_{1|t}^\theta(\mathbf{x}_1|\mathbf{x}_t)$. Once the density ratio model is optimized using Bregman divergence minimization, resulting in the optimal model $r^\star(\mathbf{x}_1, \mathbf{x}_t)$, we face the task of solving the optimization problem

$$\min_\theta \mathcal{D}(r^\star p^{\mathrm{ref}}, p^\theta)$$

to align $p^\theta$ with $p$. However, this two-stage process, alternating between density ratio estimation and divergence minimization, is not stable and is difficult to converge.

As shown in (Uehara et al., 2016), minimizing the objective

$$\mathbb{E}_{p_{1|t}^{\mathrm{ref}}(x_1^i|\mathbf{x}_1^{\neq i}, \mathbf{x}_t)}\left(F'(r^\phi(x_1^i|\mathbf{x}_1^{\neq i}, \mathbf{x}_t))r^\phi(x_1^i|\mathbf{x}_1^{\neq i}, \mathbf{x}_t) - F(r^\phi(x_1^i|\mathbf{x}_1^{\neq i}, \mathbf{x}_t))\right) - \mathbb{E}_{p_{1|t}(x_1^i|\mathbf{x}_1^{\neq i}, \mathbf{x}_t)}F'(r^\phi(x_1^i|\mathbf{x}_1^{\neq i}, \mathbf{x}_t)) \quad (28)$$

for estimating the density ratio model $r^\phi$ would lead to $f$-divergence maximization, thus such two-stage process will yield GAN-like adversarial training. This motivates us to parameterize the density ratio model in terms of the denoising model, or vice versa, as shown in Sec. 5.1.

**Reference Models** With the density ratio model parameterized, the next crucial step is selecting an appropriate reference distribution $p^{\mathrm{ref}}$. We explore two compelling options.

Weaker model as reference   At each optimization step $k$, we can set the reference distribution to be the previous step denoising distribution $p^{\text{ref}} = p_{1|t}^{\theta_{k-1}}$, and the density ratio model is parameterized as

$$r_{1|t}^{\theta}(x_1^i|\mathbf{x}_1^{\neq i}, \mathbf{x}_t) = \frac{p_{1|t}^{\theta}(x_1^i|\mathbf{x}_1^{\neq i}, \mathbf{x}_t)}{p_{1|t}^{\theta_{k-1}}(x_1^i|\mathbf{x}_1^{\neq i}, \mathbf{x}_t)}.$$

This will give us a procedure similar to SPIN (Chen et al., 2024). Alternatively, we can use the exponential moving average of the denoising distribution as the reference distribution, $p^{\text{ref}} = p_{1|t}^{\theta_{\text{ema}}}$. In this case, we naturally use the (i) parameterization strategy for the density ratio model.

Pre-trained model as reference   We can also set the reference distribution to be a pre-trained discrete diffusion model $p_{1|t}^{\text{ref}}(\mathbf{x}_1|\mathbf{x}_t) := p_{1|t}^{\text{pre}}(\mathbf{x}_1|\mathbf{x}_t)$. We can use the (ii) parameterization strategy to parameterize the density ratio model as

$$r_{1|t}^{\theta}(\mathbf{x}_1|\mathbf{x}_t) = \frac{p_{1|t}^{\theta}(\mathbf{x}_1|\mathbf{x}_t)}{p_{1|t}^{\text{pre}}(\mathbf{x}_1|\mathbf{x}_t)}.$$

The training objective becomes

$$\mathbb{E}_{p_{1|t}^{\text{ref}}(x|\mathbf{x}_1^{\neq i}, \mathbf{x}_t)} \left( F'(r^{\theta}(x))r^{\theta}(x) - F(r^{\theta}(x)) \right) - \mathbb{E}_{p_{1|t}(x|\mathbf{x}_1^{\neq i}, \mathbf{x}_t)} F'(r^{\theta}(x)).$$

---

**Algorithm 1** TCSM Post-Training with Density Ratio Estimation

---

**Require:** Dataset $\mathbf{D} := \{\mathbf{x}_1\}$
**Require:** Pre-trained model $p_{1|t}^{\text{pre}}$
**Require:** Proposal distribution $h$
**Require:** Bregman divergence function $F$
**Require:** Density ratio model $r_{1|t}^{\theta} = f_{\theta}$
**Require:** Learning rate $\eta$
1: $\mathbf{x}_1 \sim \mathbf{D}$ $\qquad\qquad\qquad\qquad\qquad\qquad\qquad\qquad\qquad\qquad\qquad$ ▷ Sample data point
2: $t \sim \omega(t)$ $\qquad\qquad\qquad\qquad\qquad\qquad\qquad\qquad\qquad\qquad\qquad$ ▷ Sample diffusion time
3: $\mathbf{x}_t \sim p_{t|1}(\mathbf{x}_t|\mathbf{x}_1)$ $\qquad\qquad\qquad\qquad\qquad\qquad\qquad\qquad\qquad\qquad$ ▷ Sample noisy data
4: $\mathbf{x}_1^{\text{ref}} \leftarrow p_{1|t}^{\text{ref}}(\mathbf{x}_1|\mathbf{x}_t)$ $\qquad\qquad\qquad\qquad\qquad\qquad\qquad$ ▷ Sample from reference distribution
5: **if** $F = $ LSIF **then** $\qquad\qquad\qquad\qquad\qquad$ ▷ Compute density ratio based on Bregman divergence
6: $\qquad \mathcal{L} \leftarrow \left( \frac{\exp(2f_{\theta}(\mathbf{x}_1^{\text{ref}}))}{2} \right) - \exp(f_{\theta}(\mathbf{x}_1))$
7: **else if** $F = $ BCE **then**
8: $\qquad \mathcal{L} \leftarrow \log(1 - \sigma(f_{\theta}(\mathbf{x}_1^{\text{ref}}))) + \log(\sigma(f_{\theta}(\mathbf{x}_1)))$
9: **else if** $F = $ Gen. KL **then**
10: $\qquad \mathcal{L} \leftarrow \exp(f_{\theta}(\mathbf{x}_1^{\text{ref}})) - f_{\theta}(\mathbf{x}_1)$
11: **end if**
12: $\theta \leftarrow \theta - \eta \nabla_{\theta} \mathcal{L}$ $\qquad\qquad\qquad\qquad\qquad\qquad\qquad\qquad\qquad$ ▷ Update parameters

---

### E.3. Experimental Details and Results

We present a thorough empirical evaluation of our density ratio estimation-based post-training methodology within the TCSM framework. While Sec. 5.3 investigates parameterization strategy (i), we concentrate here on evaluating parameterization strategy (ii), which characterizes the denoising model through density ratio estimation.

Our experimental framework utilizes a pre-trained GPT2-small model with $\mathcal{L}_{\text{distrib}}$ for language modeling tasks, implementing an absorbing state formulation as outlined in Sec. 4.1. Building upon the work of Xu et al. (2024a), we initialize our density ratio model $r_{1|t}^{\theta}(\mathbf{x}_1|\mathbf{x}_t)$ using the pre-trained diffusion model. The initialization process involves projecting mean-pooled last token embeddings to scalar values, while the partition function is estimated following the methodology proposed by Nowozin (2018).

To ensure a comprehensive evaluation, we investigate three distinct Bregman divergence measures for training the density ratio model:

- Least Squares Importance Fitting (LSIF)

- Binary Cross-Entropy (BCE)

- Generalized KL divergence

For a complete algorithmic description of our approach, we refer readers to Alg. 1.

The comparative performance of these measures is documented in Table Table 5. Notably, our implementation of TCSM with BCE shares similarities with the EDLM model - in fact, EDLM NCE (Xu et al., 2024a) can be viewed as a specific case of our framework when BCE serves as the chosen Bregman divergence.

Our experimental analysis yields several significant findings. Most prominently, the post-training approach incorporating density ratio estimation consistently outperforms the pre-trained baseline model, as demonstrated by improved perplexity metrics across all configurations. While both generalized KL divergence and binary cross-entropy achieve particularly strong results, the relatively uniform performance across all tested variants highlights the fundamental robustness of our methodology, regardless of the specific divergence measure employed. This consistency across different mathematical formulations provides strong evidence for the stability and reliability of our approach.

## F. TCSM Post-training with Reward Function

### F.1. Derivation of Objectives for Reward Tuning

In this section, we provide more comprehensive derivations of the TCSM objectives introduced in Sec. 5.2, with particular focus on their practical implementations.

$\mathcal{L}_{\texttt{score}}$ and $\mathcal{L}_{\texttt{distrib}}$ with $\mathcal{N}^1$  For the score-based TCSM objective with target distribution $p_1^R(\mathbf{x}_1)$, we can directly apply the formulation from Eq. (4):

$$\mathcal{L}_{\text{score}}\left(\theta; \mathcal{N}^1, \mathcal{D}, h\right) = \mathbb{E}_{t, \mathbf{x}_1, \mathbf{x}_t} \sum_{i=1}^{L} \mathcal{D}\left(\left[\frac{p_{1|t}^R(y_1^i, \mathbf{x}_1^{\neq i}|\mathbf{x}_t)}{p_{1|t}^R(x_1^i, \mathbf{x}_1^{\neq i}|\mathbf{x}_t)}\right]_{y_1^i=1}^{V}, \left[\frac{p_{1|t}^\theta(y_1^i, \mathbf{x}_1^{\neq i}|\mathbf{x}_t)}{p_{1|t}^\theta(x_1^i, \mathbf{x}_1^{\neq i}|\mathbf{x}_t)}\right]_{y_1^i=1}^{V}\right)$$

Let us define $\mathbf{y} := \left[y_1^i, \mathbf{x}_1^{\neq i}\right]$ and $\mathbf{x} := \left[x_1^i, \mathbf{x}_1^{\neq i}\right]$, where $y_1^i \neq x_1^i$. The ratio between reward-modulated conditional probabilities can be expressed as:

$$\frac{p_{1|t}^R(\mathbf{y}|\mathbf{x}_t)}{p_{1|t}^R(\mathbf{x}|\mathbf{x}_t)} = \frac{p_1(\mathbf{y})p_{t|1}(\mathbf{x}_t|\mathbf{y})\exp(R(\mathbf{y})/\beta)}{p_1(\mathbf{x})p_{t|1}(\mathbf{x}_t|\mathbf{x})\exp(R(\mathbf{x})/\beta)} = \frac{p_{1|t}(\mathbf{y}|\mathbf{x}_t)}{p_{1|t}(\mathbf{x}|\mathbf{x}_t)}\exp\left(\frac{R(\mathbf{y}) - R(\mathbf{x})}{\beta}\right)$$

Given access to a pre-trained model $p_{1|t}^{\text{pre}}$ that approximates $p_{1|t}$, we can reformulate the objective as:

$$\mathcal{L}_{\text{score}}\left(\theta; \mathcal{N}^1, \mathcal{D}, h\right) = \mathbb{E}_{t, \mathbf{x}_1, \mathbf{x}_t} \sum_{i=1}^{L} \mathcal{D}\left(\left[\frac{p_{1|t}^{\text{pre}}(y_1^i, \mathbf{x}_1^{\neq i}|\mathbf{x}_t)}{p_{1|t}^{\text{pre}}(x_1^i, \mathbf{x}_1^{\neq i}|\mathbf{x}_t)}\exp\left(\frac{R(y_1^i, \mathbf{x}_1^{\neq i}) - R(x_1^i, \mathbf{x}_1^{\neq i})}{\beta}\right)\right]_{y_1^i=1}^{V}, \left[\frac{p_{1|t}^\theta(y_1^i, \mathbf{x}_1^{\neq i}|\mathbf{x}_t)}{p_{1|t}^\theta(x_1^i, \mathbf{x}_1^{\neq i}|\mathbf{x}_t)}\right]_{y_1^i=1}^{V}\right)$$

For models with factorized denoising parameterizations, this objective simplifies to:

$$\mathcal{L}_{\text{score}}\left(\theta; \mathcal{N}^1, \mathcal{D}, h\right) = \mathbb{E}_{t, \mathbf{x}_1, \mathbf{x}_t} \sum_{i=1}^{L} \mathcal{D}\left(\left[\frac{p_{1|t}^{\text{pre}}(y_1^i|\mathbf{x}_t)}{p_{1|t}^{\text{pre}}(x_1^i|\mathbf{x}_t)}\exp\left(\frac{R(y_1^i, \mathbf{x}_1^{\neq i}) - R(x_1^i, \mathbf{x}_1^{\neq i})}{\beta}\right)\right]_{y_1^i=1}^{V}, \left[\frac{p_{1|t}^\theta(y_1^i|\mathbf{x}_t)}{p_{1|t}^\theta(x_1^i|\mathbf{x}_t)}\right]_{y_1^i=1}^{V}\right)$$

This formulation enables efficient computation of all terms involving $p_{1|t}^{\text{pre}}$ and $p_{1|t}^\theta$.

For the distribution-based $\mathcal{L}_{\texttt{distrib}}$ approach, we derive a similar approximation:

$$p_{1|t}^R(x_1^i|\mathbf{x}_1^{\neq i}, \mathbf{x}_t) \propto p_{1|t}^{\text{pre}}(x_1^i|\mathbf{x}_1^{\neq i}, \mathbf{x}_t)\exp(R(x_1^i, \mathbf{x}_1^{\neq i})/\beta)$$

The detailed implementation is presented in Alg. 2.

$\mathcal{L}_{\texttt{distrib}}$ **with** $\mathcal{N}^{\textbf{full}}$  When employing $\mathcal{N}^{\text{full}}$, the $\mathcal{L}_{\texttt{distrib}}$ objective takes the form:

$$\mathcal{L}_{\texttt{distrib}}(\theta; \mathcal{N}^{\text{full}}, \mathcal{D}, h) = \mathbb{E}_{\omega(t)p(\mathbf{x}_t)}\mathbb{D}\left(p_{1|t}^R(\cdot|\mathbf{x}_t) \,\|\, p_{1|t}^\theta(\cdot|\mathbf{x}_t)\right)$$

Using the approximation $p_{1|t}^{\text{pre}} \approx p_{1|t}$, we can derive:

$$\mathbb{D}_{\text{KL}}\left(p_{1|t}^R(\cdot|\mathbf{x}_t) \,\|\, p_{1|t}^\theta(\cdot|\mathbf{x}_t)\right) = \mathbb{E}_{p_{1|t}^R(\mathbf{x}_1|\mathbf{x}_t)}\log\frac{p_{1|t}^R(\mathbf{x}_1|\mathbf{x}_t)}{p_{1|t}^\theta(\mathbf{x}_1|\mathbf{x}_t)}$$

$$= \sum_{\mathbf{x}_1} p_{1|t}^R(\mathbf{x}_1|\mathbf{x}_t)\log\frac{p_{1|t}^R(\mathbf{x}_1|\mathbf{x}_t)}{p_{1|t}^\theta(\mathbf{x}_1|\mathbf{x}_t)}$$

$$= \sum_{\mathbf{x}_1} \frac{p_{1|t}(\mathbf{x}_1|\mathbf{x}_t)\exp(R(\mathbf{x}_1)/\beta)}{\sum_{\mathbf{x}_1} p_{1|t}(\mathbf{x}_1|\mathbf{x}_t)\exp(R(\mathbf{x}_1)/\beta)}\log\frac{p_{1|t}^R(\mathbf{x}_1|\mathbf{x}_t)}{p_{1|t}^\theta(\mathbf{x}_1|\mathbf{x}_t)}$$

$$= \mathbb{E}_{p_{1|t}(\mathbf{x}_1|\mathbf{x}_t)}\frac{\exp(R(\mathbf{x}_1)/\beta)}{\mathcal{Z}(\mathbf{x}_t)}\log\frac{p_{1|t}^R(\mathbf{x}_1|\mathbf{x}_t)}{p_{1|t}^\theta(\mathbf{x}_1|\mathbf{x}_t)}$$

The complete algorithm is detailed in Alg. 3.

**Connection to Reinforcement Learning**  An interesting connection emerges when we set $h_{1|t}(\mathbf{x}_1|\mathbf{x}_t) = p_1^\theta(\mathbf{x}_1|\mathbf{x}_t)$ and use $\mathbb{D}\left(p \,\|\, q\right) := \mathbb{D}_{\text{KL}}\left(q \,\|\, p\right)$ as the reverse KL divergence. The $\mathcal{L}_{\texttt{distrib}}$ objective then takes the form of a traditional RL objective:

$$\mathbb{D}\left(p_{1|t}^R(\cdot|\mathbf{x}_1^{\neq i}, \mathbf{x}_t) \,\|\, p_{1|t}^\theta(\cdot|\mathbf{x}_1^{\neq i}, \mathbf{x}_t)\right) = \mathbb{D}_{\text{KL}}\left(p_{1|t}^\theta(\cdot|\mathbf{x}_1^{\neq i}, \mathbf{x}_t) \,\|\, p_{1|t}^R(\cdot|\mathbf{x}_1^{\neq i}, \mathbf{x}_t)\right)$$

$$= \mathbb{E}_{p_{1|t}^\theta(x_1^i|\mathbf{x}_1^{\neq i}, \mathbf{x}_t)}\log\frac{p_{1|t}^\theta(x_1^i|\mathbf{x}_1^{\neq i}, \mathbf{x}_t)}{p_{1|t}^R(x_1^i|\mathbf{x}_1^{\neq i}, \mathbf{x}_t)}$$

$$= \mathbb{E}_{p_{1|t}^\theta(x_1^i|\mathbf{x}_1^{\neq i}, \mathbf{x}_t)}\log\frac{p_{1|t}^\theta(x_1^i|\mathbf{x}_1^{\neq i}, \mathbf{x}_t)}{p_{1|t}^{\text{pre}}(x_1^i|\mathbf{x}_1^{\neq i}, \mathbf{x}_t)\exp(R(x_1^i, \mathbf{x}_1^{\neq i})/\beta)} + C$$

$$= \mathbb{D}_{\text{KL}}\left(p_{1|t}^\theta(x_1^i|\mathbf{x}_1^{\neq i}, \mathbf{x}_t) \,\|\, p_{1|t}^{\text{pre}}(x_1^i|\mathbf{x}_1^{\neq i}, \mathbf{x}_t)\right) - \frac{1}{\beta}\mathbb{E}_{p_{1|t}^\theta(x_1^i|\mathbf{x}_1^{\neq i}, \mathbf{x}_t)}R(x_1^i, \mathbf{x}_1^{\neq i}) + C$$

This formulation closely resembles the standard RLHF objective, highlighting the theoretical connections between our approach and traditional reinforcement learning methods.

For practical implementation, we employ $h_{1|t}(\mathbf{x}_1|\mathbf{x}_t) = p_1^{\text{pre}}(\mathbf{x}_1|\mathbf{x}_t)$ as the proposal distribution. Since the new model $p_1$ follows a product distribution, its support must necessarily be contained within the support of $p_1^{\text{pre}}$.

### F.2. Experimental Details and Results

**Synthetic Experiments**  To assess the effectiveness of our reward function tuning methodology, we conducted experiments using a synthetic dataset. This dataset is structured as a 2D discrete grid, specifically a $128 \times 128$ grid. Initially, we pre-train a discrete diffusion model, denoted as $p^{\text{pre}}$, on this grid using the $\mathcal{L}_{\texttt{distrib}}$ objective with a uniform source distribution. Subsequently, we define a reward function $R$ designed to eliminate modes located in the right half of the grid. Concretely, we assign $R(x) = 0$ for all points $x$ in the left half, and $R(x) = -10^5$ for those in the right half. Following this setup, we fine-tune the model using the $\mathcal{L}_{\texttt{distrib}}$ objective with $\mathcal{N}^{\text{full}}$, adhering to the procedure detailed in Alg. 3.

---

**Algorithm 2** Reward-Guided Post-Training with $\mathcal{N}^1$

---

**Require:** Pre-trained model $p_{1|t}^{\text{pre}}$, proposal distribution $h$, reward function $R$, temperature $\beta$
**Require:** Model parameters $\theta$, learning rate $\eta$, sequence length $L$
1: Sample diffusion time $t \sim \omega(t)$             ▷ Sample diffusion time and generate noisy sequence
2: Sample clean sequence $\mathbf{x}_1 \sim h(\cdot|\mathbf{x}_t)$
3: Generate noisy sequence $\mathbf{x}_t \sim p(\cdot|\mathbf{x}_1)$
4: **for** $i = 1$ **to** $L$ **do**             ▷ Compute reward-modulated target distribution
5:      $p_{1|t}^R(x_1^i|\mathbf{x}_1^{\neq i}, \mathbf{x}_t) \leftarrow \frac{p_{1|t}^{\text{pre}}(x_1^i|\mathbf{x}_1^{\neq i}, \mathbf{x}_t) \exp(R(x_1^i, \mathbf{x}_1^{\neq i})/\beta)}{\sum_{x'} p_{1|t}^{\text{pre}}(x'|\mathbf{x}_1^{\neq i}, \mathbf{x}_t) \exp(R(x', \mathbf{x}_1^{\neq i})/\beta)}$
6: **end for**
7: $\mathcal{L} \leftarrow \mathcal{L}_{\texttt{distrib}}(\theta; \mathcal{N}^1, \mathcal{D}, h)$             ▷ Compute loss and update parameters
8: $\theta \leftarrow \theta - \eta \nabla_\theta \mathcal{L}$             ▷ Gradient descent step

---

**Algorithm 3** Reward-Guided Training with $\mathcal{N}^{\text{full}}$

---

**Require:** Pre-trained model $p_{1|t}^{\text{pre}}$, proposal distribution $h$, reward function $R$, temperature $\beta$
**Require:** Model parameters $\theta$, learning rate $\eta$
1: $t \sim \omega(t)$             ▷ Sample diffusion time
2: $\mathbf{x}_t \sim p(\mathbf{x}_t)$             ▷ Sample noise
3: Sample mini-batch $\{\mathbf{x}_{1,b}\}_{b=1}^B \sim h(\mathbf{x}_1|\mathbf{x}_t)$             ▷ Draw samples from proposal
4: $\mathcal{Z} \leftarrow \sum_{b=1}^B \exp(R(\mathbf{x}_{1,b})/\beta)$             ▷ Compute normalization
5: $w_b \leftarrow \exp(R(\mathbf{x}_{1,b})/\beta)/\mathcal{Z}$ for $b = 1, \ldots, B$             ▷ Importance weights
6: $\mathcal{L} \leftarrow \sum_{b=1}^B w_b \log \frac{p_{1|t}^R(\mathbf{x}_{1,b}|\mathbf{x}_t)}{p_{1|t}^\theta(\mathbf{x}_{1,b}|\mathbf{x}_t)}$             ▷ Weighted objective
7: $\theta \leftarrow \theta - \eta \nabla_\theta \mathcal{L}$             ▷ Gradient update

---

The results of this process are illustrated in Figure 5, which displays the intermediate samples generated by the model both before and after fine-tuning. Initially, during the pre-training phase, the model successfully captures all modes present in the data distribution. However, after applying reward-guided fine-tuning, the model effectively suppresses the modes in the right half of the grid, resulting in final samples that exclusively generate the left half of the grid.

**Toxicity Mitigation** A critical challenge in deploying language models is effectively controlling and mitigating toxic content in their outputs. Although toxic generations occur relatively infrequently, their potential negative impact on users and downstream applications makes this an essential area of research (Singhal et al., 2025). Even a small proportion of toxic outputs can significantly undermine the safety, reliability, and trustworthiness of language models in real-world scenarios.

Our experimental methodology builds upon recent advances in controlled text generation (Zhao et al., 2024a; Rector-Brooks et al., 2024; Singhal et al., 2025). To ensure reproducibility, we conduct our experiments using a standardized story-beginning prompt: "Once upon a time, there was a". The foundation of our experimental framework is a pre-trained diffusion model developed in Sec. 4.1, which implements $\mathcal{L}_{\texttt{distrib}}$ with absorbing discrete diffusion. To further enhance the model's capabilities and robustness, we perform comprehensive fine-tuning on the Tinystories dataset (Eldan & Li, 2023). This fine-tuning process utilizes the Adam optimizer with ($\beta_1 = 0.9$, $\beta_2 = 0.95$) and a learning rate of $1 \times 10^{-4}$, continuing for 100,000 training steps.

For measuring and controlling toxicity, we implement a sophisticated reward function based on a pre-trained RoBERTa classifier (Logacheva et al., 2022). During our evaluation phase, we employ this classifier as our primary metric for assessing content safety, with outputs scored on a continuous scale from 0 (completely non-toxic) to 1 (highly toxic). This granular scoring system allows for precise measurement of our mitigation strategies' effectiveness.

The results of our comprehensive evaluation are presented in Fig. 3, where we analyze two critical metrics: the toxicity score and the generative perplexity of the samples. To assess the quality and coherence of the generated text, we measure perplexity using GPT-2 Large (Radford et al., 2019) as an independent evaluator.

We fine-tune the model using the $\mathcal{L}_{\texttt{distrib}}$ objective with $\mathcal{N}^{\text{full}}$, following the procedure outlined in Alg. 3. To investigate

the impact of sampling density, we conduct experiments with varying numbers of Monte Carlo samples $N \in \{2, 4, 8, 16\}$ for estimating the importance weights, with results displayed in Fig. 3. For comparative analysis, we include benchmark results from the pre-trained MDLM (Sahoo et al., 2024) model using Best-of-N sampling with $N \in \{4, 8\}$, as reported in (Singhal et al., 2025).

Our experimental results demonstrate several key findings. First, our approach exhibits superior scaling properties with respect to the number of Monte Carlo samples used for importance weight estimation. Second, our fine-tuning methodology achieves more effective toxicity mitigation compared to the pre-trained MDLM model, even when the latter employs Best-of-N sampling techniques. Notably, since our approach is based on fine-tuning rather than inference-time scaling, it eliminates the need for multiple reward function evaluations during inference, resulting in reduced computational overhead and improved efficiency in practical applications.

## G. TCSM Post-training with Preference Optimization

### G.1. Detailed Algorithm

**Problem Setting** We introduce a methodology for fine-tuning pre-trained diffusion models using pairwise preference data, denoted as $\{(\mathbf{q}, \mathbf{x}_1^w, \mathbf{x}_1^l)\}$. In this formulation, $\mathbf{q}$ represents a query or instruction, while $\mathbf{x}_1^w$ and $\mathbf{x}_1^l$ represent the preferred (winning) and non-preferred (losing) responses, respectively.

The underlying preferences are assumed to emerge from a latent reward model that is not directly observable. Among various approaches for modeling such preferences, we adopt the widely-recognized Bradley-Terry (BT) model (Bradley & Terry, 1952). This model provides an elegant framework for capturing human preference distributions. Specifically, the BT model expresses the probability of one response being preferred over another as:

$$p^*(\mathbf{x}_1^w \succ \mathbf{x}_1^l | \mathbf{q}) = \frac{\exp(R^*(\mathbf{q}, \mathbf{x}_1^w))}{\exp(R^*(\mathbf{q}, \mathbf{x}_1^w)) + \exp(R^*(\mathbf{q}, \mathbf{x}_1^l))} \tag{29}$$

where $R^*(\mathbf{q}, \mathbf{x})$ represents the underlying reward function that quantifies the quality of response $\mathbf{x}$ given query $\mathbf{q}$.

Building on this foundation, we define our target distribution to emphasize preferred responses. This distribution can be formally expressed as:

$$p_{\text{target}}(\mathbf{x}_1 | \mathbf{q}) := p_1(\mathbf{x}_1^w | \mathbf{q}) := p_1(\mathbf{x}_1 \text{ is winner} | \mathbf{q}) = p_1(\mathbf{x}_1 | \mathbf{q}) \sum_{\mathbf{y}_1} p_1(\mathbf{y}_1 | \mathbf{q}) p^*(\mathbf{x}_1 \succ \mathbf{y}_1 | \mathbf{q}), \tag{30}$$

For practical implementation, we leverage a pre-trained diffusion model $p_{1|t}^{\text{pre}}(\mathbf{x}_1 | \mathbf{q})$ as our reference distribution, which serves as the starting point for our fine-tuning process.

Based on the TCSM with density ratio estimation approach in Sec. 5.1, we learn a new diffusion model $p_{1|t}^\theta$ relative to the pre-trained reference. The detailed algorithm is shown in Alg. 4, where we use BCE loss to estimate the density ratio as an example.

---

**Algorithm 4** Preference Optimization with TCSM using BCE loss

---

**Require:** Pre-trained diffusion model $p_{1|t}^{\text{pre}}$

**Require:** Preference dataset $\mathcal{D} = \{(\mathbf{c}, \mathbf{x}^w, \mathbf{x}^l)\}$

**Require:** Model parameters $\theta$, learning rate $\eta$, time distribution $\omega(t)$, coefficient $\beta$

1: **for** each training iteration **do**
2:      $t \sim \omega(t)$                          ▷ Sample diffusion time
3:      $(\mathbf{c}, \mathbf{x}^w, \mathbf{x}^l) \sim \mathcal{D}$            ▷ Sample preference triplet
4:      $\mathbf{x}_t^w \sim p_{t|1}(\cdot|\mathbf{x}_1^w)$          ▷ Sample noisy sequence for preferred response
5:      $\mathbf{x}_t^l \sim p_{t|1}(\cdot|\mathbf{x}_1^l)$        ▷ Sample noisy sequence for non-preferred response
6:      ▷ Compute density ratios for preferred and non-preferred responses
7:      $r_{1|t}^w(\mathbf{c}) \leftarrow \frac{p_{1|t}^\theta(\mathbf{x}^w|\mathbf{c})}{\beta p_{1|t}^{\text{pre}}(\mathbf{x}^w|\mathbf{c})}$
8:      $r_{1|t}^l(\mathbf{c}) \leftarrow \frac{p_{1|t}^\theta(\mathbf{x}^l|\mathbf{c})}{\beta p_{1|t}^{\text{pre}}(\mathbf{x}^l|\mathbf{c})}$
9:      ▷ Compute loss
10:     $\mathcal{L} \leftarrow -\log \frac{r_{1|t}^w(\mathbf{c})}{1+r_{1|t}^w(\mathbf{c})} - \log \frac{1}{1+r_{1|t}^l(\mathbf{c})}$
11:     $\theta \leftarrow \theta - \eta\nabla_\theta \mathcal{L}$            ▷ Update model parameters
12: **end for**

---

## G.2. Experimental Details and Results

To evaluate the effectiveness of preference optimization, we employed the IMDB-sentiment dataset (Maas et al., 2011) as our primary evaluation benchmark, with the SiEBERT model (Hartmann et al., 2023) serving as our reward function. For training data, we utilized a carefully curated preference dataset constructed in prior work (Rafailov et al., 2023; Wang et al., 2023). As our foundation model, we used the pre-trained model from Sec. 4.1, which had been extensively trained on the OPENWEBTEXT dataset.

The fine-tuning process implemented our density ratio estimation framework, as detailed in Sec. 5.1, with Binary Cross-Entropy (BCE) loss serving as our optimization objective. We adopted parameterization strategy (i) from Sec. 5.1, which defines the density ratio as:

$$r_{1|t}^{\phi:=\theta}(x_1^i|\mathbf{x}_1^{\neq i}, \mathbf{x}_t) = \frac{p_{1|t}^\theta(x_1^i|\mathbf{x}_1^{\neq i}, \mathbf{x}_t)}{\beta p_{1|t}^{\text{ref}}(x_1^i|\mathbf{x}_1^{\neq i}, \mathbf{x}_t)}$$

Here, the coefficient $\beta$ plays a crucial role in balancing two competing objectives: maximizing preference reward optimization while maintaining fidelity to the original pre-trained model. The complete training procedure is outlined in Alg. 4.

Our training protocol consisted of 10 full epochs with a batch size of 256. We employed the Adam optimizer with a learning rate of $1 \times 10^{-5}$ and weight decay of $1 \times 10^{-5}$. To ensure stable training, we implemented a linear learning rate warmup for the first 10% of training steps, with momentum parameters $\beta_1 = 0.9$ and $\beta_2 = 0.95$. The noise schedule remained consistent with that of the pre-trained model to maintain continuity in the diffusion process.

To thoroughly investigate the effects of preference optimization, we conducted experiments across a range of $\beta$ values: $\{0.1, 0.5, 1, 5\}$. Our evaluation focused on two key metrics: the mean reward achieved by the fine-tuned model and the entropy of generated samples. As shown in Fig. 2, we observed that models with stronger preference optimization (higher $\beta$ values) achieved both higher mean rewards and lower sample entropy. This suggests that our approach improves alignment with desired preferences but also leads to less diverse generation of preferred samples.

## H. TCSM Post-training with AR → Diffusion Distillation

**Problem setting** In this case, we assume we have a pre-trained autoregressive model $p_1^{\text{AR}}(\mathbf{x}_1)$ trained on the target distribution $p_1(\mathbf{x}_1)$, and we show that we can use TCSM to distill it to a diffusion model $p_1^\theta(\mathbf{x}_1)$. Note that this deviates from the regular diffusion models setting, that we have the knowledge of the target distribution $p_1(\mathbf{x}_1) \approx p^{\text{AR}}(\mathbf{x}_1)$, and we can use it as a teacher model. In this section, we set the target distribution to be the AR teacher model distributoin $p_1(\mathbf{x}_1) := p_1^{\text{AR}}(\mathbf{x}_1)$.

---

**Algorithm 5** Top-K Estimation

---

1: **procedure** tcs_estimate($\mathbf{x}_0$, teacher_model, $L, V, K$, tcs)
2:                                ▷ $\mathbf{x}_0$: Input tokens; $L$: Sequence length; $V$: Vocabulary size; $K$: Top-$K$ tokens to select; tcs: list
3:      logits $\leftarrow$ teacher_model($\mathbf{x}_0$) $\in \mathbb{R}^{V \times L}$; original_log_prob $\leftarrow$ teacher_model_log_prob($\mathbf{x}_0$)
4:      **for** $l = 1$ to $L$ **do**
5:          Get top-$K$ tokens: top_tokens $\leftarrow$ TopK(logits[:, $l$], $K$)
6:          If $\mathbf{x}_0[l] \notin$ top_tokens, add it to top_tokens
7:          Construct a batch of new sequences $\widehat{\mathbf{x}}_0 \leftarrow [\mathbf{x}_0^{<l}, \text{top\_tokens}, \mathbf{x}_0^{>l}]$
8:          Compute log probability of sequences log_prob from new_logits $\leftarrow$ teacher_model($\widehat{\mathbf{x}}_0$)
9:          Compute log-density ratio: log_density_ratio $\leftarrow$ log_prob $-$ orig_log_prob
10:         Append log-density ratio to list: tcs $\leftarrow$ tcs $+$ log_density_ratio
11:      **end for**
12:      **return** tcs
13: **end procedure**

---

And akin to classical knowledge distillation, we are interested in how to distill the knowledge from the AR teacher model to the diffusion student model.

**TCSM objectives for distillation** We show that our TCSM objectives can naturally integrate the knowledge of the AR teacher model into the training objective.

We have

$$p_{1|t}(\mathbf{x}_1|\mathbf{x}_t) = p_1^{\text{AR}}(\mathbf{x}_1)p_{t|1}(\mathbf{x}_t|\mathbf{x}_1)\Big/{\textstyle\sum_{\mathbf{x}_1}} p_1^{\text{AR}}(\mathbf{x}_1)p_{t|1}(\mathbf{x}_t|\mathbf{x}_1)$$

.

We can also use $p_1^{\text{AR}}(\mathbf{x}_1)$ to estimate

$$p_{1|t}(x_1^i|\mathbf{x}_1^{\neq i}, \mathbf{x}_t) = \frac{p_1^{\text{AR}}(x_1^i, \mathbf{x}_1^{\neq i})p_{t|1}(\mathbf{x}_t|x_1^i, \mathbf{x}_1^{\neq i})}{\sum_{y_1^i} p_1^{\text{AR}}(y_1^i, \mathbf{x}_1^{\neq i})p_{t|1}(\mathbf{x}_t|y_1^i, \mathbf{x}_1^{\neq i})}$$

.

Both score-based and distribution-based TCSM objectives can be used to distill the AR teacher model to the diffusion student model, we use the distribution-based TCSM objective in our experiments and assume it is the default setting in following discussions.

**Efficient estimation of distillation target** To optimize the TCSM objective, we need to compute the distillation target $p_1^{\text{AR}}(\mathbf{x}_1)$. Naively, this requires $(V - 1) \times L + 1$ likelihood evaluations of the teacher autoregressive model for each sequence $\mathbf{y} \in \mathcal{N}^1(\mathbf{x})$. Even though that the likelihood evaluation can be done in parallel for the autoregressive model, this procedure is still computationally prohibitive. To address this challenge, we introduce two approaches to efficiently estimate the target concrete score, *Top-$K$* estimation and *First-order Taylor* estimation.

Top-$K$ approximation Our empirical analysis reveals that distribution $p_{1|t}(x_1^i|\mathbf{x}_1^{\neq i}, \mathbf{x}_t)$ are naturally sparse. As illustrated in Fig. 6, tokens with high density ratios closely resemble the one-hot encoding of original tokens in the simplex space, but enriched with distributional information. This observation motivates approximating the score vector with only the top-$K$ items, treating the rest as zero, for efficient computation. We leverage this property to propose an efficient top-$K$ approximation that reduces computational complexity from $O(VL)$ to $O(KL)$ by considering only the $K$ most probable tokens at each position. This approximation can be efficiently implemented using batched forward passes and proves effective even with $K \leq 128$ - for detailed implementation and the complete algorithm, we refer readers to Alg. 5 in the appendix.

First-order Taylor approximation We leverage the fact that autoregressive language models, despite operating on discrete tokens, are differentiable functions that can be approximated using Taylor expansion. For sequences that differ by only one position, we can efficiently estimate the likelihood ratio using first-order Taylor approximation: $\log p_{1|t}(y_1^i, \mathbf{x}_1^{\neq i}|\mathbf{x}_t) \approx \log p_{1|t}(x_1^i, \mathbf{x}_1^{\neq i}|\mathbf{x}_t) + \nabla_{\mathbf{e}_{\mathbf{x}_1}} \log p_{1|t}(\mathbf{x}_1|\mathbf{x}_t)^\top (\mathbf{e}_{\mathbf{y}_1} - \mathbf{e}_{\mathbf{x}_1})$. This gradient-based estimation requires just one forward and backward pass through the teacher model; for detailed derivations and implementation, please refer to Alg. 7.

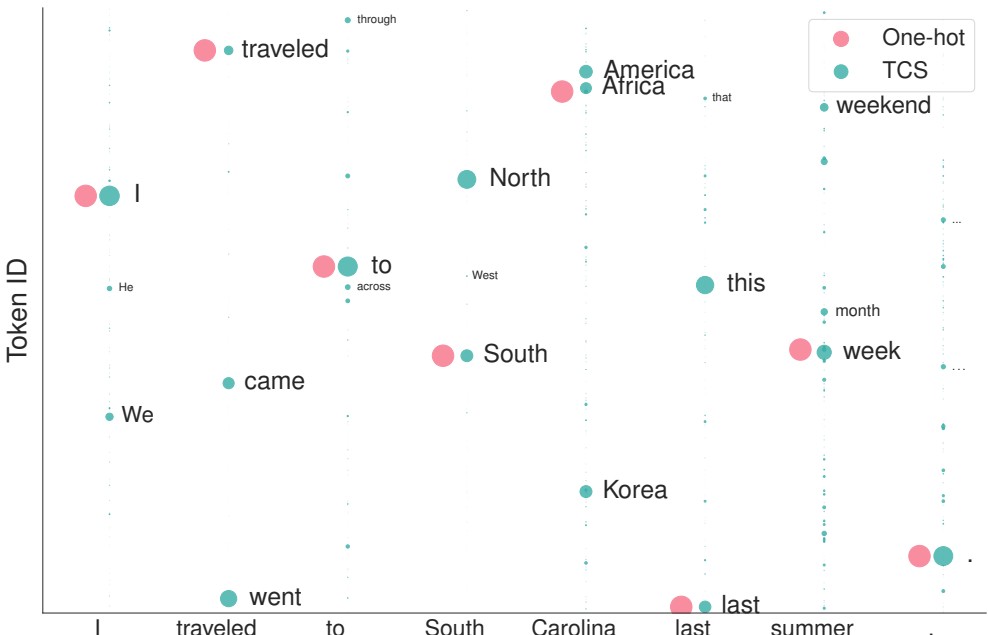

Figure 6: Visualization of the concrete score for sequence "I traveled to South Carolina last summer.". The x-axis represents the position in the sequence, and the y-axis represents the log-probability ratio. The red line represents the original token, and the blue lines represent the top-K tokens with the highest log-probability ratios. The concrete score is highly sparse, with most of the probability mass concentrated on a few tokens.

**Experimental** To validate our distillation approach, we conducted comprehensive experiments focusing on language modeling capabilities using the OPENWEBTEXT dataset. Our experimental setup involved two key components: a teacher model and a student model. For the teacher, we pre-trained a transformer-based autoregressive model following the architectural configurations described in (Sahoo et al., 2024). As our student model, we employed an absorbing discrete diffusion model.

The training process utilized our Top-K estimation strategy with $K = 128$, training the student model from scratch. To assess performance, we tracked the validation negative log-likelihood (NLL) loss on the OPENWEBTEXT dataset, which we visualize in Figure Fig. 4. The empirical results demonstrate two significant findings: First, our distillation approach substantially accelerates the student model's learning trajectory compared to standard training. Second, and perhaps more importantly, models trained with our distillation loss consistently achieve lower perplexity scores than baseline approaches throughout the entire training process, indicating improved model quality.

## I. Connection to Continuous Target Score Matching

In this section, we elaborate on the relationship between the proposed Target Concrete Score Matching (TCSM) framework for discrete data and the established Target Score Matching (TSM) objective (Bortoli et al., 2024) used in continuous diffusion models. We first briefly review TSM in the context of language modeling via continuous diffusion and then demonstrate how TCSM can be viewed as its discrete analogue under certain approximations.

Continuous diffusion models for language often operate in a continuous embedding space. Let $\mathbf{x}_1 = [x_1^1, \ldots, x_1^L]$ be a discrete sequence from the vocabulary $\mathcal{X} = \{1, \ldots, V\}$. Let $\mathbf{E} \in \mathbb{R}^{d \times V}$ be a word embedding matrix, where $d$ is the embedding dimension. The one-hot vector for token $k$ is $\mathbf{e}_k \in \{0, 1\}^V$. The embedding for token $x_1^l$ is $\mathbf{E}^\top \mathbf{e}_{x_1^l}$. The forward noising process typically acts independently on these embeddings:

$$q_{t|1}(\mathbf{z}_t|\mathbf{x}_1) = \prod_{l=1}^{L} q_{t|1}(\mathbf{z}_t^l|x_1^l) = \prod_{l=1}^{L} \mathcal{N}(\mathbf{z}_t^l; \alpha_t \mathbf{E}^\top \mathbf{e}_{x_1^l}, \sigma_t^2 \mathbf{I}_d), \tag{31}$$

---

**Algorithm 6** Top-K with N-Gram Estimation

---

1: **procedure** tcs_estimate($\mathbf{x}_1$, teacher_model, ngram_model, $L, V, K$, tcs)
2:                                    ▷ $\mathbf{x}_1$: Input tokens; $L$: Sequence length; $V$: Vocabulary size; $K$: Top-$K$ tokens to select; tcs: list
3:     logits $\leftarrow$ teacher_model($\mathbf{x}_1$) $\in \mathbb{R}^{V \times L}$; original_log_prob $\leftarrow$ teacher_model_log_prob($\mathbf{x}_1$)
4:     **for** $l = 1$ to $L$ **do**
5:         Get top-$K$ tokens: top_tokens $\leftarrow$ TopK(logits[:, $l$], $K$)
6:         Get N-Gram score for all tokens: n-gram_scores $\leftarrow$ ngram_model($[\mathbf{x}_1^{l+1}, \ldots, \mathbf{x}_1^{l+N-1}]$)
7:         Add another top-$K$ tokens: top_tokens $\leftarrow$ top_tokens + TopK(n-gram_scores, $K$)
8:         If $\mathbf{x}_1[l] \notin$ top_tokens, add it to top_tokens
9:         Construct a batch of new sequences $\widehat{\mathbf{x}}_1 \leftarrow [\mathbf{x}_1^{<l}, \text{top\_tokens}, \mathbf{x}_1^{>l}]$
10:        Compute log probability of sequences log_prob from new_logits $\leftarrow$ teacher_model($\widehat{\mathbf{x}}_1$)
11:        Compute log-density ratio: log_density_ratio $\leftarrow$ log_prob $-$ orig_log_prob
12:        Append log-density ratio to list: tcs $\leftarrow$ tcs + log_density_ratio
13:     **end for**
14:     **return** tcs
15: **end procedure**

---

**Algorithm 7** Concrete Score Estimation with first-order Taylor approximation

---

1: **procedure** tcs_estimate(teacher_model, tokens, $V, \tau$)
2:                                  ▷ tokens: Input tokens of shape $(B, L)$; $V$: Vocabulary size; $\tau$: Temperature
3:     $\mathbf{x}_1 \leftarrow$ one_hot(tokens, $V$)                                           ▷ Convert to one-hot vectors
4:     Enable gradient computation for $\mathbf{x}_1$
5:     logits $\leftarrow$ teacher_model($\mathbf{x}_1$)
6:     log_prob $\leftarrow$ log_softmax(logits)
7:     log_prob $\leftarrow \sum(\mathbf{x}_1[:, 1 :, :] \cdot \text{log\_prob}[:, : -1, :])$
8:     Compute gradient: grad_log_prob $\leftarrow \nabla_{\mathbf{x}_1}$log_prob
9:                                                  ▷ Compute log-density ratios
10:     log_prob_ratio $\leftarrow$ grad_log_prob $- \sum_{\dim=-1}(\mathbf{x}_1 \cdot \text{grad\_log\_prob})$
11:     Scale by temperature: log_prob_ratio $\leftarrow$ log_prob_ratio$/\tau$
12:     prob_ratio $\leftarrow \exp($log_prob_ratio$)$
13:     **return** prob_ratio
14: **end procedure**

---

where $(\mathbf{z}_t^l)_{l=1}^L$ forms the sequence of noisy embeddings $\mathbf{z}_t \in \mathbb{R}^{L \times d}$, and $\alpha_t, \sigma_t$ are schedule parameters. The goal is to learn the score function $\nabla_{\mathbf{z}_t} \log q_t(\mathbf{z}_t)$ of the marginal distribution $q_t(\mathbf{z}_t) = \int q_{t|1}(\mathbf{z}_t|\mathbf{x}_1)q_1(\mathbf{x}_1)d\mathbf{x}_1$.

Target Score Matching (TSM) provides an objective when the score of the clean data distribution, $\nabla_{\mathbf{z}_1} \log p_1(\mathbf{z}_1)$ (where $\mathbf{z}_1$ represents the clean embeddings and $p_1$ is a density over them), is known or can be estimated. The following identity connects the noisy score to the clean score:

**Lemma I.1** (Target Score Matching Identity, adapted from (Bortoli et al., 2024)). *Let $q_{t|1}(\mathbf{z}_t|\mathbf{z}_1) = \mathcal{N}(\mathbf{z}_t; \alpha_t\mathbf{z}_1, \sigma_t^2\mathbf{I})$ define the forward process conditioned on clean continuous data $\mathbf{z}_1$, and let $p_1(\mathbf{z}_1)$ be a differentiable distribution over $\mathbf{z}_1$. Then, the score of the noisy marginal $q_t(\mathbf{z}_t) = \int q_{t|1}(\mathbf{z}_t|\mathbf{z}_1)p_1(\mathbf{z}_1)d\mathbf{z}_1$ is given by:*

$$\nabla_{\mathbf{z}_t} \log q_t(\mathbf{z}_t) = \frac{1}{\alpha_t}\mathbb{E}_{q_{1|t}(\mathbf{z}_1|\mathbf{z}_t)}\left[\nabla_{\mathbf{z}_1} \log p_1(\mathbf{z}_1)\right], \tag{32}$$

*where $q_{1|t}(\mathbf{z}_1|\mathbf{z}_t)$ is the posterior distribution.*

*Proof.* The proof follows standard arguments, e.g., in Bortoli et al. (2024), adapted for the scaling factor $\alpha_t$. Using the property $\nabla_{\mathbf{z}_1} \log q_{t|1}(\mathbf{z}_t|\mathbf{z}_1) = -\alpha_t\nabla_{\mathbf{z}_t} \log q_{t|1}(\mathbf{z}_t|\mathbf{z}_1)$ and Bayes' rule $q_{t|1}(\mathbf{z}_t|\mathbf{z}_1) = q_{1|t}(\mathbf{z}_1|\mathbf{z}_t)q_t(\mathbf{z}_t)/p_1(\mathbf{z}_1)$, we take gradients w.r.t. $\mathbf{z}_1$: $\nabla_{\mathbf{z}_1} \log q_{t|1}(\mathbf{z}_t|\mathbf{z}_1) = \nabla_{\mathbf{z}_1} \log q_{1|t}(\mathbf{z}_1|\mathbf{z}_t) - \nabla_{\mathbf{z}_1} \log p_1(\mathbf{z}_1)$. Combining these yields $\nabla_{\mathbf{z}_t} \log q_{t|1}(\mathbf{z}_t|\mathbf{z}_1) = -\frac{1}{\alpha_t}(\nabla_{\mathbf{z}_1} \log q_{1|t}(\mathbf{z}_1|\mathbf{z}_t) - \nabla_{\mathbf{z}_1} \log p_1(\mathbf{z}_1))$. Finally, taking the expectation w.r.t. $q_{1|t}(\mathbf{z}_1|\mathbf{z}_t)$: $\nabla_{\mathbf{z}_t} \log q_t(\mathbf{z}_t) =$

$\mathbb{E}_{q_{1|t}(\mathbf{z}_1|\mathbf{z}_t)}[\nabla_{\mathbf{z}_t}\log q_{t|1}(\mathbf{z}_t|\mathbf{z}_1)] = -\frac{1}{\alpha_t}\mathbb{E}_{q_{1|t}}[\nabla_{\mathbf{z}_1}\log q_{1|t}] + \frac{1}{\alpha_t}\mathbb{E}_{q_{1|t}}[\nabla_{\mathbf{z}_1}\log p_1(\mathbf{z}_1)]$. Since $\mathbb{E}_{q_{1|t}}[\nabla_{\mathbf{z}_1}\log q_{1|t}] = \int \nabla_{\mathbf{z}_1} q_{1|t}(\mathbf{z}_1|\mathbf{z}_t)d\mathbf{z}_1 = 0$ (assuming boundary conditions), the identity holds. $\qquad\square$

Using Lemma I.1, a score network $\mathbf{s}_\theta(\mathbf{z}_t, t)$ can be trained by minimizing the TSM loss:

$$\mathcal{L}_{\text{TSM}}(\theta) = \mathbb{E}_{t\sim U(0,1)}\mathbb{E}_{p_1(\mathbf{z}_1)q_{t|1}(\mathbf{z}_t|\mathbf{z}_1)}\left\|\mathbf{s}_\theta(\mathbf{z}_t, t) - \frac{1}{\alpha_t}\nabla_{\mathbf{z}_1}\log p_1(\mathbf{z}_1)\right\|_2^2. \tag{33}$$

Alternatively, using the mean prediction parameterization $\boldsymbol{\mu}_\theta(\mathbf{z}_t, t) \approx \mathbb{E}_{q_{1|t}(\mathbf{z}_1|\mathbf{z}_t)}[\mathbf{z}_1]$, and Tweedie's formula $\mathbb{E}_{q_{1|t}(\mathbf{z}_1|\mathbf{z}_t)}[\mathbf{z}_1] = \frac{1}{\alpha_t}(\sigma_t^2\nabla_{\mathbf{z}_t}\log q_t(\mathbf{z}_t) + \mathbf{z}_t)$, the TSM objective becomes equivalent to minimizing (up to scaling by $\lambda_t = \alpha_t^2/\sigma_t^2$):

$$\mathcal{L}_{\text{TSM}}^{\boldsymbol{\mu}}(\theta) = \mathbb{E}_{t\sim U(0,1)}\mathbb{E}_{p_1(\mathbf{z}_1)q_{t|1}(\mathbf{z}_t|\mathbf{z}_1)}\left\|\boldsymbol{\mu}_\theta(\mathbf{z}_t, t) - \left(\frac{\sigma_t^2}{\alpha_t}\nabla_{\mathbf{z}_1}\log p_1(\mathbf{z}_1) + \frac{1}{\alpha_t}\mathbf{z}_t\right)\right\|_2^2. \tag{34}$$

Note: The exact form depends slightly on conventions; here we target a scaled version of the clean score plus noise term. Let $\mathbf{T}(\mathbf{z}_1, \mathbf{z}_t, t) := \frac{\sigma_t^2}{\alpha_t}\nabla_{\mathbf{z}_1}\log p_1(\mathbf{z}_1) + \frac{1}{\alpha_t}\mathbf{z}_t$ be the target for the mean predictor.

Now, let's connect this to the discrete TCSM objective. Consider the log-probability ratio (concrete score component) for the posterior distribution $q_{1|t}(\mathbf{x}_1|\mathbf{z}_t)$ in the continuous setting, where $\hat{\mathbf{x}}_1$ differs from $\mathbf{x}_1$ only at position $i$ (i.e., $\hat{x}_1^i = j \neq x_1^i$, and $\hat{x}_1^l = x_1^l$ for $l \neq i$):

$$\log\frac{q_{1|t}(\hat{\mathbf{x}}_1|\mathbf{z}_t)}{q_{1|t}(\mathbf{x}_1|\mathbf{z}_t)} = \log\frac{q_1(\hat{\mathbf{x}}_1)}{q_1(\mathbf{x}_1)} + \log\frac{q_{t|1}(\mathbf{z}_t|\hat{\mathbf{x}}_1)}{q_{t|1}(\mathbf{z}_t|\mathbf{x}_1)}. \tag{35}$$

The second term simplifies due to the product structure of $q_{t|1}$:

$$\log\frac{q_{t|1}(\mathbf{z}_t|\hat{\mathbf{x}}_1)}{q_{t|1}(\mathbf{z}_t|\mathbf{x}_1)} = \log\frac{q_{t|1}(\mathbf{z}_t^i|\hat{x}_1^i)}{q_{t|1}(\mathbf{z}_t^i|x_1^i)} \tag{36}$$

$$\propto -\frac{\|\mathbf{z}_t^i - \alpha_t\mathbf{E}^\top\mathbf{e}_{\hat{x}_1^i}\|^2}{2\sigma_t^2} + \frac{\|\mathbf{z}_t^i - \alpha_t\mathbf{E}^\top\mathbf{e}_{x_1^i}\|^2}{2\sigma_t^2} \tag{37}$$

$$= \frac{\alpha_t}{\sigma_t^2}\langle\mathbf{z}_t^i, \mathbf{E}^\top(\mathbf{e}_{\hat{x}_1^i} - \mathbf{e}_{x_1^i})\rangle - \frac{\alpha_t^2}{2\sigma_t^2}\left(\|\mathbf{E}^\top\mathbf{e}_{\hat{x}_1^i}\|^2 - \|\mathbf{E}^\top\mathbf{e}_{x_1^i}\|^2\right). \tag{38}$$

Let's assume embeddings have similar norms, making the last term negligible, or absorb it into the definition.

For the first term, $\log\frac{q_1(\hat{\mathbf{x}}_1)}{q_1(\mathbf{x}_1)}$, we use a first-order Taylor approximation in the continuous embedding space $\mathbf{z}_1 = [\mathbf{E}^\top\mathbf{e}_{x_1^1}, \ldots, \mathbf{E}^\top\mathbf{e}_{x_1^L}]$ corresponding to $\mathbf{x}_1$. Let $p_1(\mathbf{z}_1)$ be the density over these embeddings. Then:

$$\log\frac{p_1(\mathbf{z}_{\hat{\mathbf{x}}_1})}{p_1(\mathbf{z}_{\mathbf{x}_1})} \approx \log p_1(\mathbf{z}_{\mathbf{x}_1}) + \langle\nabla_{\mathbf{z}_1}\log p_1(\mathbf{z}_1), \mathbf{z}_{\hat{\mathbf{x}}_1} - \mathbf{z}_{\mathbf{x}_1}\rangle - \log p_1(\mathbf{z}_{\mathbf{x}_1}) \tag{39}$$

$$= \langle\nabla_{\mathbf{z}_1}\log p_1(\mathbf{z}_1), \mathbf{z}_{\hat{\mathbf{x}}_1} - \mathbf{z}_{\mathbf{x}_1}\rangle \tag{40}$$

$$= \langle(\nabla_{\mathbf{z}_1}\log p_1(\mathbf{z}_1))_i, \mathbf{E}^\top(\mathbf{e}_{\hat{x}_1^i} - \mathbf{e}_{x_1^i})\rangle, \tag{41}$$

where $(\cdot)_i$ denotes the gradient block corresponding to the $i$-th position embedding.

Combining Eq. (38) (simplified) and Eq. (41), the target concrete score is approximately:

$$\mathbf{r}_{q_{1|t}}(\mathbf{x}_1|\mathbf{z}_t)_{i,j} := \log\frac{q_{1|t}(\mathbf{x}_1|x_1^i \leftarrow j|\mathbf{z}_t)}{q_{1|t}(\mathbf{x}_1|\mathbf{z}_t)} \tag{42}$$

$$\approx \langle(\nabla_{\mathbf{z}_1}\log p_1(\mathbf{z}_1))_i + \frac{\alpha_t}{\sigma_t^2}\mathbf{z}_t^i, \mathbf{E}^\top(\mathbf{e}_j - \mathbf{e}_{x_1^i})\rangle. \tag{43}$$

Now, consider the model prediction $p_\theta(\mathbf{x}_1|\mathbf{z}_t)$, often parameterized via logits $\boldsymbol{\mu}_\theta(\mathbf{z}_t, t)$ such that $p_\theta(x_1^i = j|\mathbf{z}_t) = \text{softmax}([\boldsymbol{\mu}_\theta]_{:,i})_j$. The model's concrete score is:

$$\mathbf{r}_{p_\theta}(\mathbf{x}_1|\mathbf{z}_t)_{i,j} = [\boldsymbol{\mu}_\theta]_{j,i} - [\boldsymbol{\mu}_\theta]_{x_1^i,i} = \langle[\boldsymbol{\mu}_\theta]_{:,i}, \mathbf{e}_j - \mathbf{e}_{x_1^i}\rangle. \tag{44}$$

The TCSM objective aims to match $\mathbf{r}_{p_\theta}$ to $\mathbf{r}_{q_{1|t}}$. The TSM objective (Eq. (34)) encourages $\boldsymbol{\mu}_\theta(\mathbf{z}_t, t) \approx \mathbf{T}' \coloneqq \frac{\sigma_t^2}{\alpha_t} \nabla_{\mathbf{z}_1} \log p_1(\mathbf{z}_1) + \frac{1}{\alpha_t} \mathbf{z}_t$. If this holds, then from Eq. (44):

$$\mathbf{r}_{p_\theta}(\mathbf{x}_1 | \mathbf{z}_t)_{i,j} \approx \langle [\mathbf{T}']_{:,i}, \mathbf{e}_j - \mathbf{e}_{x_1^i} \rangle = \langle \left( \frac{\sigma_t^2}{\alpha_t} \nabla_{\mathbf{z}_1} \log p_1(\mathbf{z}_1) \right)_i + \frac{1}{\alpha_t} \mathbf{z}_t^i, \mathbf{e}_j - \mathbf{e}_{x_1^i} \rangle. \tag{45}$$

Comparing this to the target approximation in Eq. (43), we see they align (up to scaling factors and potential embedding norm terms) if $\mathbf{E} = \mathbf{I}$. When $\mathbf{E} \neq \mathbf{I}$, the alignment is approximate.

In summary, under the first-order Taylor approximation for the marginal discrete probability ratio and assuming word embeddings $\mathbf{E}$ behave similarly to an identity mapping (or have negligible impact on the inner products compared to the main terms), minimizing the TCSM objective, which matches discrete concrete scores, serves as an approximation to minimizing the continuous TSM objective. This provides a conceptual link between the two frameworks, highlighting how TCSM adapts score-matching principles to the discrete domain.

## J. Experimental Configuration Summary Table

To enhance clarity and facilitate reproducibility, this section provides a comprehensive summary of the specific models, parameterizations, and training objectives used for each experimental result presented throughout the paper. Sec. J details the configuration for each key experiment, linking the reported results (identified by their table or figure number) to the underlying methodological choices, including the prior distribution (source distribution for diffusion), the structure of the denoising model $p_{1|t}^\theta$, the proposal distribution $h(\mathbf{x}_1 | \mathbf{x}_t)$ used within the loss computation (if applicable), and the specific TCSM training objective function employed.

## K. Related Work

Generative modeling (Austin et al., 2021; Song et al., 2021; Ho et al., 2020; Vincent, 2011; Song & Ermon, 2019) has seen significant advances through diffusion models, initially developed for continuous data like images. Applying these principles effectively to discrete data, such as text or graphs, presents unique challenges due to the non-differentiable nature of discrete spaces and has spurred several distinct lines of research.

**Score Matching and Continuous Diffusion Foundations**  The theoretical underpinning for many modern diffusion models is **Score Matching** (Hyvärinen et al., 2009). This method estimates parameters $\theta$ for models $p(\mathbf{x}; \theta) \propto q(\mathbf{x}; \theta)$ with intractable normalization constants by minimizing the difference between the model's score function $\nabla_{\mathbf{x}} \log q(\mathbf{x}; \theta)$ and the data score $\nabla_{\mathbf{x}} \log p_x(\mathbf{x})$. A key insight by Hyvärinen et al. (2009) showed that this objective can be computed using only the model score and its derivatives on data samples, avoiding the need for the true data density or normalization constant. A crucial practical development was **Denoising Score Matching (DSM)** (Vincent, 2011), which established an equivalence between score matching on noise-perturbed data and training specific denoising autoencoders (DAEs). DSM matches the model's score at a noisy point $\tilde{\mathbf{x}}$ to the score of the conditional denoising distribution, avoiding the second derivatives required by original score matching and making score estimation more tractable.

These principles were central to the development of diffusion models. Early work framed diffusion via forward (noising) and reverse (denoising) Markov processes trained with a variational lower bound (VLB) (Sohl-Dickstein et al., 2015). Subsequently, score-based generative models (Song & Ermon, 2019) directly applied DSM by training a single **Noise Conditional Score Network (NCSN)** $s_\theta(\mathbf{x}, \sigma)$ to estimate scores $\nabla_{\mathbf{x}} \log q_{\sigma_i}(\mathbf{x})$ across multiple noise levels $\{\sigma_i\}$, using annealed Langevin dynamics for sampling. **Denoising Diffusion Probabilistic Models (DDPM)** (Ho et al., 2020) refined this, particularly for images, by parameterizing the reverse process to predict the added noise $\epsilon$ and using a simplified VLB-derived objective shown to be equivalent to DSM over multiple noise scales. While highly successful, standard DSM can suffer from high variance at low noise levels. **Target Score Matching (TSM)** (Bortoli et al., 2024) addresses this by incorporating knowledge of the clean target score $\nabla \log p(\mathbf{x})$ when available, leading to lower variance estimators in the low-noise regime.

**Continuous Diffusion for Discrete Data**  One approach to handle discrete data involves operating within continuous embedding spaces, adapting standard continuous diffusion techniques. This allows leveraging powerful continuous models but requires mapping back to the discrete space. **Diffusion-LM** (Li et al., 2022) applied continuous diffusion to word

embeddings, enabling controllable text generation via gradient guidance during sampling. **Plaid** (Gulrajani & Hashimoto, 2023) focused on likelihood-based training for text, jointly optimizing embeddings and model parameters using the VLB, categorical reparameterization, an output prior, a learned conditional likelihood $p(x|z_0)$, and self-conditioning. **CDCD** (Dieleman et al., 2022) employed a probability flow ODE on embeddings, using score interpolation to jointly train embeddings and a denoising Transformer with a cross-entropy loss, along with time warping. **Bit Diffusion** (Chen et al., 2023) treated the binary representation of discrete data as continuous "analog bits," enhanced by self-conditioning and asymmetric time intervals. While effective, these methods rely on continuous approximations or embeddings, motivating research into models operating directly on discrete domains. Furthermore, many of these works explore non-autoregressive approaches enabling parallel generation (Bowman et al., 2016; Gu et al., 2018; Li et al., 2022; Hoogeboom et al., 2021; Savinov et al., 2022; Che et al., 2017; Zhang et al., 2020; Yu et al., 2017; de Masson d'Autume et al., 2019; Deng et al., 2020), contrasting with sequential autoregressive models.

**Discrete Diffusion Models** A parallel line of research develops diffusion processes inherently designed for discrete state spaces, often using Markov chains. Building on early foundations (Sohl-Dickstein et al., 2015; Hoogeboom et al., 2021), **D3PM** (Austin et al., 2021) generalized discrete diffusion using various structured transition matrices (e.g., uniform, absorbing, Gaussian-like) and trained via a hybrid VLB/cross-entropy loss. Campbell et al. (2022) extended this to Continuous-Time Markov Chains (CTMCs), deriving a continuous-time ELBO and proposing efficient sampling methods like tau-leaping and predictor-corrector schemes, leveraging factorization for high-dimensional data.

**Score-like Analogues and Masking Mechanisms for Discrete Diffusion** Instead of direct Markov chain simulation, other works define score-like quantities for discrete diffusion. The concrete score, defined as the ratio of marginal probabilities $p_t(\mathbf{y})/p_t(\mathbf{x})$, acts as a discrete analogue to the continuous score (Meng et al., 2022; Lou et al., 2024). **SEDD** (Lou et al., 2024) trained models using a score entropy objective ($L_{DSE}$) derived from this ratio, connecting it to the ELBO and using Tweedie $\tau$-leaping for sampling. Sun et al. (2023) developed categorical ratio matching within a CTMC framework, learning singleton conditionals $p_t(x^d|\mathbf{x}^{\backslash d})$ with a tractable loss and an analytical reverse sampler. Building on this, Ou et al. (2024) showed that for absorbing diffusion, the concrete score factorizes into a time-independent conditional and a time-dependent scalar, simplifying the model (**RADD**) and yielding the Denoising Cross-Entropy (DCE) loss.

Masked (or absorbing) diffusion, which replaces tokens with a special [MASK] token during the forward process, has proven particularly effective. **MDLM** (Sahoo et al., 2024) introduced a substitution-based parameterization (SUBS) and derived a simplified Rao-Blackwellized ELBO equivalent to weighted Masked Language Modeling (MLM) losses, enabling generative training of encoder-only models. Shi et al. (2024) (**MD4**) further unified this framework, deriving a simple ELBO with SNR invariance properties similar to continuous diffusion and generalizing to state-dependent masking schedules.

Further research has refined the parameterization and mechanisms of discrete diffusion. **Reparameterized Discrete diffusion Models (RDM)** (Zheng et al., 2023) identified an underlying route-and-denoise mechanism, simplifying the objective to cross-entropy on noisy tokens and enabling adaptive routing during sampling. Liu et al. (2024b) proposed **Discrete Diffusion with Planned Denoising (DDPD)**, factorizing the reverse process into a planner (predicting corruption) and a denoiser, allowing adaptive sampling via the Gillespie algorithm guided by the planner.

Discrete Flow Matching offers another generalization pathway. Gat et al. (2024) defined probability paths interpolating discrete distributions and derived corresponding probability velocities, analogous to continuous flow matching, providing a unified sampling theory. (Campbell et al., 2024) formulated discrete flows using CTMCs, learning scores via cross-entropy and enabling inference-time flexibility by adjusting the rate matrix family without retraining, also unifying multimodal generation. Discrete diffusion principles have also been applied to structured data, such as graphs in **DiGress** (Vignac et al., 2023), using specific noise transitions, auxiliary features, and classifier guidance.

**Scaling and Adapting Pre-trained Models for Diffusion Language Modeling** Significant recent effort has focused on scaling diffusion models for language generation, often by adapting large pre-trained autoregressive (AR) or masked language models (MLMs). **DiffusionBERT** (He et al., 2023) integrated BERT into an absorbing-state diffusion framework, leveraging pre-trained weights and exploring novel noise schedules and time conditioning. Ye et al. (2023) adapted pre-trained MLMs (like XLM-R) for generative tasks by finetuning with an RDM objective, enabling instruction-following capabilities. **AR2Diff** (Han et al., 2024) proposed converting pre-trained AR models to diffusion models by enabling bidirectional attention and continuing training with a diffusion objective. **DiffuLLaMA** (Gong et al., 2024) presented

a continual pre-training method to adapt AR models (like LLaMA) into time-embedding-free diffusion models using attention mask annealing. **LLaDA** (Nie et al., 2025) developed a large masked diffusion model trained with a masking objective, adapting standard pre-training and SFT pipelines for this non-autoregressive paradigm. These works demonstrate the potential of leveraging existing large model architectures and weights to build capable diffusion language models.

**Guidance and Control in Discrete Diffusion**    Controlling the generation process of discrete diffusion models is vital for their application. Several approaches modify the sampling procedure or the model itself. Nisonoff et al. (2024) introduced Discrete Guidance (DG), a principled framework for guidance in CTMC-based models, offering exact predictor guidance (PG), predictor-free guidance (PFG), and an efficient Taylor-Approximated Guidance (TAG) variant by exploiting tractable normalization constants during inference. **FK-steering** (Singhal et al., 2025) provides a general inference-time steering approach using Feynman-Kac interacting particle systems, applicable even with non-differentiable rewards via parallel simulation and resampling. An alternative strategy involves finetuning the model itself to incorporate guidance. Rector-Brooks et al. (2024) proposed **Discrete Denoising Posterior Prediction (DDPP)**, a framework for steering pre-trained Masked Diffusion Models (MDMs) according to a reward function $R(\mathbf{x}_1)$. DDPP reframes steering as learning an amortized sampler (via finetuning the MDM) for a target posterior distribution proportional to $p_\theta^{\text{pre}}(\mathbf{x}_1)R(\mathbf{x}_1)$. By exploiting the relationship between the target denoising posterior, the pre-trained model's posterior, and the reward, DDPP derives several simulation-free training objectives, offering a scalable approach to bake reward-based control into the model. Other methods include informed corrector steps based on confidence scores combined with architectural changes and novel training objectives for masked diffusion (Zhao et al., 2024b), and adaptations of standard classifier-free or classifier-based guidance for discrete domains, sometimes coupled with improved ELBO formulations suitable for guidance (Schiff et al., 2024).

**LLM Distillation**    Our work also relates to LLM distillation (Xu et al., 2024b), which focuses on transferring capabilities from large teacher models to smaller student models. Common techniques involve distribution matching, specialized loss functions (e.g., **MiniLLM** (Gu et al., 2024), **DistiLLM** (Ko et al., 2024)), using rationales (Hsieh et al., 2023), or dynamic data selection (Liu et al., 2024a). While most existing methods distil knowledge between autoregressive models, our research explores knowledge transfer from powerful AR teachers to bidirectional diffusion students. This presents distinct challenges, particularly regarding the mismatch between the teacher's sequential generation process and the student's non-autoregressive, iterative refinement process, but potentially benefits from similar underlying principles aimed at effective knowledge transfer and mitigating distribution discrepancies.

| Model Variant / Name (Defining Section/Eq.) | Experiment (Table/Figure) | Prior (Source Dist.) | Denoising Model Parameterization $p_{1|t}^\theta$ | Proposal distribution $h(\mathbf{x}_1|\mathbf{x}_t)$ | Training Objective (Equation / Description) |
|---|---|---|---|---|---|
| *Experiments on* TEXT8 *(Fig. 1b)* | | | | | |
| TCSM Uniform $\mathcal{L}_{\text{score}}$ (Sec. 4.2) | Fig. 1b | Uniform | Factorized: $p_{1|t}^\theta(\mathbf{x}_1|\mathbf{x}_t) = \prod_{i=1}^L p_{1|t}^\theta(x_1^i|\mathbf{x}_t)$ | $p_{1|t}(\mathbf{x}_1|\mathbf{x}_t)$ | $\mathcal{L}_{\text{score}}$ with Gen KL (Monte Carlo version: Eq. (10)) |
| TCSM Uniform $\mathcal{L}_{\text{distrib}}$ (Sec. 4.2) | Fig. 1b | Uniform | Factorized (as above) | $p_{1|t}(\mathbf{x}_1|\mathbf{x}_t)$ | $\mathcal{L}_{\text{distrib}}$ with KL (Cross-Entropy: Factorized version of Eq. (9)) |
| TCSM Absorb $\mathcal{L}_{\text{score}}$ (Sec. 4.2) | Fig. 1b | Mask (Absorbing) | Factorized (as above) | $p_{1|t}(\mathbf{x}_1|\mathbf{x}_t)$ | $\mathcal{L}_{\text{score}}$ with Gen KL (Monte Carlo version: Eq. (10)) |
| TCSM Absorb $\mathcal{L}_{\text{distrib}}$ (Sec. 4.2) | Fig. 1b | Mask (Absorbing) | Factorized (as above) | $p_{1|t}(\mathbf{x}_1|\mathbf{x}_t)$ | $\mathcal{L}_{\text{distrib}}$ with KL (Cross-Entropy: Factorized version of Eq. (9)) |
| TCSM Absorb $\mathcal{L}_{\text{distrib}}$ (Sec. 5.1) | Fig. 1b | Mask (Absorbing) | Density Ratio (Strategy ii): $p_{1|t}^\theta(\mathbf{x}_1|\mathbf{x}_t) \propto p_{1|t}^{\text{ref}}(\mathbf{x}_1|\mathbf{x}_t)\exp(f_\theta(\mathbf{x}_1|\mathbf{x}_t))$ (Ref = Pre-trained TCSM Absorb $\mathcal{L}_{\text{distrib}}$) | $p_{1|t}^{\text{ref}} = p_{1|t}^{\text{pre}}$ | **Post-training phase:** DRE objective using Gen KL (Table 4, column 3) |
| *Experiments on* OPENWEBTEXT *(Table 3, Fig. 1a, Fig. 4)* | | | | | |
| TCSM Uniform $\mathcal{L}_{\text{score}}$ (Sec. 4.2) | Table 3 | Uniform | Factorized (as above) | $p_{1|t}(\mathbf{x}_1|\mathbf{x}_t)$ | $\mathcal{L}_{\text{score}}$ with Gen KL (Eq. (10)) |
| TCSM Uniform $\mathcal{L}_{\text{distrib}}$ (Sec. 4.2) | Table 3 | Uniform | Factorized (as above) | $p_{1|t}(\mathbf{x}_1|\mathbf{x}_t)$ | $\mathcal{L}_{\text{distrib}}$ with KL (Factorized version of Eq. (9)) |
| TCSM Absorb $\mathcal{L}_{\text{distrib}}$ (Sec. 4.2) | Table 3 | Mask (Absorbing) | Factorized (as above) | $p_{1|t}(\mathbf{x}_1|\mathbf{x}_t)$ | $\mathcal{L}_{\text{distrib}}$ with KL (Factorized version of Eq. (9)) |
| TCSM Absorb $\mathcal{L}_{\text{distrib}}$ (Sec. 5.1) | Table 3 | Mask (Absorbing) | Density Ratio (Strategy ii, as above) (Ref = Pre-trained TCSM Absorb $\mathcal{L}_{\text{distrib}}$) | $p_{1|t}^{\text{ref}} = p_{1|t}^{\text{pre}}$ | **Post-training phase:** DRE objective using Gen KL (Table 4, column 3) |
| TCSM-Bert (Sec. 4.2) | Fig. 1a | Mask (Absorbing) | Factorized (as above) | $p_{1|t}(\mathbf{x}_1|\mathbf{x}_t)$ | $\mathcal{L}_{\text{distrib}}$ with KL (Target $p_{1|t}$ uses BERT approx. for $p_1$) |
| TCSM-AR (Sec. 4.2) | Fig. 1a | Mask (Absorbing) | Factorized (as above) | $p_{1|t}(\mathbf{x}_1|\mathbf{x}_t)$ | $\mathcal{L}_{\text{distrib}}$ with KL (Target $p_{1|t}$ uses AR approx. for $p_1$) |
| TCSM-Hollow (Sec. 4.2) | Fig. 1a | Mask (Absorbing) | Factorized (as above) | $p_{1|t}(\mathbf{x}_1|\mathbf{x}_t)$ | $\mathcal{L}_{\text{distrib}}$ with KL (Target $p_{1|t}$ uses Hollow approx. for $p_1$) |
| TCSM Distillation (Sec. 5.4) | Fig. 4 | Mask (Absorbing) | Factorized (Student Model) | $p_{1|t}(\mathbf{x}_1|\mathbf{x}_t)$ | $\mathcal{L}_{\text{distrib}}$ with KL (Target $p_{1|t}$ uses AR Teacher via Top-K approx.) |
| *Density Ratio Estimation Bregman Comparison (Table 5)* | | | | | |
| TCSM BCE (Reimpl.) (Sec. 5.1) | Table 5 | Mask (Absorbing) | Density Ratio (Strategy ii) | $p_{1|t}^{\text{ref}} = p_{1|t}^{\text{pre}}$ | DRE objective using BCE (Table 4, column 3) |
| TCSM LSIF (Sec. 5.1) | Table 5 | Mask (Absorbing) | Density Ratio (Strategy ii, as above) | $p_{1|t}^{\text{ref}} = p_{1|t}^{\text{pre}}$ | DRE objective using LSIF (Table 4, column 3) |
| TCSM Gen KL (Sec. 5.1) | Table 5 | Mask (Absorbing) | Density Ratio (Strategy ii, as above) | $p_{1|t}^{\text{ref}} = p_{1|t}^{\text{pre}}$ | DRE objective using Gen KL (Table 4, column 3) |
| *Post-training Fine-tuning Experiments* | | | | | |
| TCSM Reward Tuning (Sec. 5.2) | Fig. 5 (Synthetic) | Uniform | Standard denoising model $p_{1|t}^\theta$ (Factorized assumed) | $p_{1|t}^{\text{pre}}$ | Weighted KL objective for $p_{1|t}^R$ with $\mathcal{N}^{\text{full}}$ (Alg. 3, Line 7) |

Table 8: Detailed summary of model configurations for experiments reported in the paper.

