# OpenReview forum: "Target Concrete Score Matching: A Holistic Framework for Discrete Diffusion"
_ICML.cc/2025/Conference — ICML 2025 poster_

### Official Review · Reviewer_wm3g · 2025-03-06

**Overall Recommendation:** 2

**Summary:**

The paper introduces a novel training objective for discrete diffusion models, dubbed Target Concrete Score Matching (TCSM), which is based on the concrete score (Meng et al., 2022).
Specifically, two different objectives are proposed: One is based on some divergence between the target and predicted concrete score, whereas the other is based on some divergence between the target and predicted distribution (conditioned on a noisy version of the data). It is shown not only that the two objectives have the same global minimum, but also that if the global minimum is reached, then the target and model distribution are the same almost everywhere.
By choosing the forward KL-divergence as a specific instance of TCSM and under a factorized parameterization (where every token is modeled independently), the objective is claimed to reduce to the simple cross-entropy reconstruction objective, where the clean data is predicted from its noisy version.
To demonstrate the flexibility of TCSM, the proposed training objective is applied to pre-training a language model with or without some reference model, and also adapted for post-training tasks in the form of (offline) reward optimization, (offline) preference tuning, as well as knowledge distillation from some teacher language model.
The reported empirical results are on-par with or slightly ahead of prior work.

## Update after rebuttal
Thank you to the authors for earnestly and extensively engaging with my questions and feedback. There were a lot of initial concerns, some of which were addressed by the authors' rebuttal and some of which turned into different concerns, some of which in turn have been addressed by the authors' followup response. In the following, I will summarize which of my concerns have and have not been addressed after the rebuttal:
1. **Faster convergence of TCSM:** While this point was initially unclear, it seems that TCSM converges faster than baselines _initially_, but performs worse than the baselines (on masked diffusion) after full convergence. In light of this, claiming "faster convergence" is, IMO, unjustified as _convergence_ usually refers to the point in training where the model no longer improves, i.e. full convergence. In the absence of further evidence to the contrary, I believe that optimizing the diffusion ELBO directly is still superior for pre-training.
2. **TCSM encompasses several existing methods:** This concern has been partially addressed. I now see how TCSM can be used to train various discrete diffusion models, including MDMs. However, claiming that, e.g., MDM is an instance of TCSM still requires a mathematical proof.
3. **Likelihood computation:** TCSM relies on existing theory for likelihood bounds and does not provide a likelihood bound in and of itself. This confusion has been addressed.
3. **Soundness of theoretical claims:** Most of my concerns have been adequately addressed. The intractability of $p\_{1|t}$ remains a concern. This has now been reduced to the intractability of $p\_t(x\_t)$, which is still an issue if not conditioned on $x\_1$. During training, we neither sample from nor approximate $p\_t(x\_t)$ but instead sample from $p\_1(x\_1)$ and then $p\_t(x\_t | x\_1)$. This results in the same joint distribution, but would imply that the theory does not describe what we actually do in practice.
4. **Data prediction:** The authors point towards DFM for this terminology, but the DFM targets are probability _flows_ rather than probability _distributions_. In the DFM sense, TCSM is neither noise nor data prediction, since we are dealing with a distribution over denoised tokens rather than a probability flow. I maintain that the usage of the term in this paper is unnecessary and confusing.

Overall, given that many of the concerns on theoretical soundness have been addressed, I will raise my score from 1 (reject) to 2 (weak reject). Due to the initial extent of issues and given that TCSM for pre-training (and especially the claim of faster convergence) stands on weak evidence, I believe that major revisions in the form of improved clarity and framing are necessary. Unfortunately, this makes it hard for me to recommend an accepting decision, despite the novel and significant contributions.

**Claims And Evidence:**

Section 4.2 claims that the proposed method helps mitigate “slow convergence, high variance gradients, and reduced sample efficiency compared to autoregressive models” by using a parametric model (a pre-trained language model, AR or non-AR). While using a parametric model indeed improves convergence compared to the TCSM baseline, no evidence on gradient variance is provided. Furthermore, the setup requires having access to a capable pre-trained model, which itself requires compute resources to train, thus offsetting any efficiency gains. I therefore cannot confidently conclude any of the above claims from the provided evidence.

Section 4.1 (L268 ff.) claims that TCSM “encompasses several existing discrete diffusion methods, including MD4 (Shi et al., 2024), MDLM (Sahoo et al., 2024), and DFM (Gat et al., 2024)”, with different source distributions and divergence measure supposedly giving rise to various objectives from the literature (as per Table 2). However, this is stated without proof, and the claim is not at all obvious. For starters: Masked diffusion models (MD4 and MDLM) have an ELBO that takes the form of a weighted reconstruction loss, where the weights $\frac{\alpha_t’}{1 - \alpha_t}$ are determined by the noise schedule $\alpha_t$. While the proposed TCSM $l_\mathrm{distrib}$ loss indeed seems to reduce to a reconstruction loss in certain cases, it is entirely unclear where the scaling factor would come from. Adding to the confusion is the fact that the equivalence is claimed for the $\mathcal{L}_\mathrm{distrib}$ loss, which is not actually an instance of TCSM (Eq. 3), but only has the same global minimum (as per Propositions 2).

The paper also claims an improvement in PPL compared to other discrete diffusion models (MDLM and SEDD), but the cited numbers are different from what is reported in the original paper. This calls into question the soundness of either the methodology or the reported numbers.

**Essential References Not Discussed:**

It seems to me like the comparison to [1] and/or [2] would be appropriate, and arguably more pressing than comparing to discrete flow matching.

- [1] Lou et al., 2023. https://arxiv.org/abs/2310.16834
- [2] Sun et al., 2022. https://arxiv.org/abs/2211.16750

**Experimental Designs Or Analyses:**

The paper relies on likelihood as a measure of model performance, which is fine, but the lack of qualitative examples somewhat casts doubt on the sample quality of the proposed model. I suggest providing at least some samples from each experiment (unconditional generation, IMDB sentiment tuning, toxicity tuning) in the appendix.

**Methods And Evaluation Criteria:**

The paper relies heavily on likelihood as an evaluation criterion (BPC, PPL), and while the chosen datasets are consistent with prior work, there is no explanation on how one actually calculates the likelihood under the proposed model. While it is somewhat plausible that one _can_ calculate likelihood from the concrete score, it is not obvious (to me) how to do this, and the paper is lacking detail in this regard. If the intended claim is that the proposed $\mathcal{L}\_\mathrm{score}$ and
$\mathcal{L}_\mathrm{distrib}$ losses are likelihood bounds, this would have to be proven mathematically.

**Other Comments Or Suggestions:**

On a more subjective note, I have found that the clarity of the writing has a lot of room for improvement. Things feel, at times, overly and unnecessarily general, in a way that obfuscates the contributions of the paper (e.g. do we really need to sample $t$ over an arbitrary distribution if this distribution in practice will always be $U(0, 1)$?). The theoretical results could also be put into perspective a bit better, leading the reader to the desired conclusions. For example: It is not clear to me why Proposition 3 gives new insight, considering that we have already proved equivalence of $l_\mathrm{score}$ and $l_\mathrm{distrib}$ in Proposition 2. Further, while Proposition 2 assumes some specific divergence measures, as far as I can tell, these are not the ones that are later used in experiments. Another example: Table 2 is not only stated without proof, but also never elaborated on, and it is left to the reader to “figure out” what it means and entails.

To summarize: I would suggest to streamline the theory by clearly stating all the conclusions we would like to take away, to lead the reader towards the desired conclusions as clearly and directly as possible, and to remove anything that does not contribute towards this goal.

**Other Strengths And Weaknesses:**

The paper provides an interesting and new perspective on discrete diffusion models and, if the claimed performance improvements are real, improves the state-of-the-art. This is the paper’s strength, which should be focused in future revisions.

However, the mathematical rigor and the soundness of claimed results leaves a lot to be desired and makes it hard for me to recommend an accepting decision. I am happy to reevaluate my conclusion if these concerns can be addressed, but the rebuttal period constitutes a tight deadline given the extent of the issues. It may be deemed the better, less stressful option to take the necessary time to improve the writeup and promptly resubmit it at a future date.

**Questions For Authors:**

- How does one compute likelihood under the proposed model? Are objectives (4) and (5) supposed to be likelihood bounds?
- How does one generate samples with the proposed model?
- How is Eq. (10) derived?
- What is the proposal distribution $h(x_1 | x_t)$ during pretraining? This seems like an essential component of the training objective that is (as far as I can tell) never specified until Section 5.
- How are baseline numbers for SEDD and MDLM obtained? The reported numbers (Figure 1) are different from what is claimed in each respective original paper: Sahoo et al. report a PPL of 23.21, not >30, for MDLM and 24.10, not >34, for SEDD. What is the reason for this discrepancy?
- Table 6: What data is used to compute PPL? Also, the table is not referenced in the main text.
Where is the “data-prediction objective” (Eq. 12) from? I cannot find it in either of the two references (Campbell et al., 2024; Gat et al., 2024).

**Relation To Broader Scientific Literature:**

The paper combines ideas from “Target Score Matching” (Bortoli et al., 2024) and “Concrete Score Matching” (Meng et al., 2022). While I am not deeply familiar with these two papers, it seems like a more detailed discussion on how the presented method is similar/different from and builds upon these two prior works would be appropriate, especially considering that the papers in question are only referenced in passing and are not mentioned in Section 2 (Preliminaries).

**Theoretical Claims:**

Proposition 4.1 refers to the appendix for proof, but I cannot find this proof anywhere. Therefore the claim is stated without proof.

For Eq. (9), it is claimed that the derivation is straightforward starting at Eq. (5), but I do not see how this simplification occurs. Specifically, while there are indeed some constant terms (which we cannot simply “drop” and claim _equality_), the expectation/sum over $p_{1|t}$ that is present in the KL-divergence but not Eq. (9) cannot (trivially) be ignored. Hence, and as far as I can tell, while Eq. (9) can still serve as a training objective, the claimed equality does not hold.

The simplified objectives arising from a factorized parameterization (Eq. 10 and L263) are also stated without proof, and it is non-obvious how they follow from the stated assumption.

The proof of Proposition 1 (App. B.1) assumes “mild regularity conditions”, which is a term that I am not familiar with, and I am unable to find any further elaboration in the paper. Clarifying these conditions would improve the mathematical rigor. Other than this, the proof seems correct.

Proof of Proposition 2:
- RHS of Eq. (19) is repeated twice.
- RHS of Eq. (20) is identical to Eq. (19). It seems to me like at least one of them should be different.
- Consequently, I have a hard time coming to the claimed conclusion.

Proof of Proposition 3 seems fine after skimming.

---

> ### Author Rebuttal · Authors · 2025-04-01
>
> We thank the reviewer for their thorough and insightful review. We address each point in detail below, quoting relevant comments, and will incorporate all suggestions into revision.
>
> Due to the paper's density, we aimed to balance presenting our method and providing essential background information. We omitted some standard practices for brevity, the concerns arising from these omissions can be clarified below with minor revisions, as they do not reflect flaws in our method or experiments.
>
> # Q5
> >Inconsistent Fig 1 baseline numbers
>
> >cited numbers are different from original paper. calls into question soundness of methodology or reported numbers.
>
> The discrepancy comes from different experimental setups. Fig 1 shows models trained on 26B OpenWebText tokens, making scores incomparable to original papers. We mentioned this in Sec 4.2 (L301) and Fig 1's caption, but agree it needs more emphasis. See response to Q2 of vK3t for more details.
>
> # Q1
> >How to compute likelihood?
>
> We use same ELBO methods as prior works:
> - Masked diffusion: MD4/MDLM approach (Eq.4 in [1])
> - Uniform diffusion: DFM method (Eq.7.32 in [3])
>
> >Are Eq(4,5) likelihood bounds?
>
> >Writing is overly general (is sampling t from arbitrary $\omega(t)$ necessary?)
>
> >Masked diffusion models have ELBO as weighted loss based on noise schedule, the source of the scaling factor in TCSM isn't clear.
>
> $\omega(t)$ provides necessary weighting for valid ELBO and scaling factors, following diffusion literature (Eq.7 in [5], [4 Sec2.1]). For masked diffusion, ELBO weighting is $\lambda(t) = \frac{d\alpha_t/dt}{1-\alpha_t}$, so $\omega(t) \propto \lambda(t)$. We sample $t$ uniformly and apply $\lambda(t)$, equivalent to $t \sim \omega(t)$.
> Eq (4, 5) are likelihood bounds only when $\omega(t)$ matches the ELBO weighting scheme.
>
> # Q2
> Refer to Q1 of vK3t.
>
> # Q3
> Eq.10 comes from Eq.9 using conditional independence
> $p^{\theta}_{1|t}(\mathbf{x}_1 | \mathbf{x}_t) = \prod\_{i=1}^{L} p^{\theta}\_{1|t} (x^i_1 | \mathbf{x}_t)$.
> Thus $p^{\theta}\_{1|t}(\mathbf{x}\_1 | \mathbf{x}\_1^{\neq i}, \mathbf{x}\_t)= p^{\theta}\_{1|t}(\mathbf{x}\_1 | \mathbf{x}\_t)$, yielding Eq.10.
>
> # Q4
> In pre-training, $h_{1|t} = p_{1|t}$ sampling from true data distribution, noted in Prop. 4.1 (L195) and L239.
>
> # Q6
> Table shows OWT val PPL from OWT-trained models. We'll add main text references.
>
> >Where is data-prediction (Eq.12) from?
>
> "Data-pred" follows DDPM conventions, used alongside "noise-prediction" or "v-prediction". It refers to predicting clean data $\mathbf{x}_1$ from noisy $\mathbf{x}_t$, versus predicting noise. See Eq. 109 in [4] for more.
>
> # Q7
> In Sec 4.2, we discuss gradient variance to motivate using parametric models for faster convergence. Prior works (e.g., Fig 2 in [6]) shows high gradient variance can slow convergence in diffusion models. While we don't directly measure variance reduction, our experiments demonstrate TCSM's faster convergence, addressing this core issue. We'll clarify in the revision that sample efficiency is our direct objective and demonstrated benefit.
>
> Using existing LLMs (e.g. LLaMa) as parametric models avoids pretraining costs. Though we trained from scratch on OWT for fair comparisons, results show potential for leveraging pretrained LLMs.
>
> # Q8
> Simplification comes from using $p_{1|t}$ as $h_{1|t}$ and forward KL. We'll add complete notation: $l^i\_{\text{distrib}} = - \mathbb{E}\_{p_{1|t}(x\_1^i | \mathbf{x}\_t)} \log p\_{1|t}^{\theta}(x^i\_1| \mathbf{x}\_t) + C$.
>
> # Q9
> >RHS of Eq.19 repeated twice
>
> RHS terms differ by $\mathbf{x}_1^{\neq i}$ positions.
>
> >RHS of Eq.20 is identical to Eq.19
>
> The proof shows equivalence between two concrete score views: Eq.19's LHS gives conditional score $p(x_1^i | \mathbf{x}_1^{\neq i}, \mathbf{x}_t)$, while Eq.20's LHS gives $i$-th component of joint score $p(\mathbf{x}_1 | \mathbf{x}_t)$. They're equal as normalization constants cancel in ratio-based concrete score.  Though the RHS expressions appear identical, the key difference lies in their LHS interpretations
>
> # Q11
> See response to Q5 of vK3t.
>
> # Q12
> >Prop 3 insight, Prop 4.1 Proof
>
> Prop 3 enables practical TCSM objective estimation from data. Since $\mathbf{c}\_{p\_{1|t}}$ is unknown, it reformulates the objective using KL and IS divergences, leading to the objective in Prop 4.1 used in experiments. It thus connects theory to practical estimation.
>
> We'll include the proof of Prop 4.1 in appendix, which straightforwardly writes out the explicit forms of the KL and IS divergences.
>
> # Q13
> >Discussed divergence not used in exp
>
> Gen. KL used in "TCSM Absorb $L_{score}$" (Table 4). Forward KL used in $\ell_{\text{distrib}}$ in "TCSM Absorb $L_{distrib}$" (Tables 3-4). Post-training uses f-divergence, giving objective (Eq. 11), tested in Tab 5-6 and App E.2.
>
> # Q14
> >Table 2 not elaborated
>
> Please refer to Q4 of vK3t.
>
> # Q15
> >Samples
>
> Will be added in the revision.
>
> # Ref
> [1] 2406.04329
> [2] 2406.07524
> [3] 2412.06264
> [4] 2303.00848
> [5] 2011.13456
> [6] 2503.09573

---

> > ### Comment · Reviewer_wm3g · 2025-04-02
> >
> > Thank you to the authors for responding to many of my concerns. In the following I will outline which concerns have and have not been addressed.
> >
> > - Q5: Based on the caption of Fig. 1, it seems like the validation set, not the training set, consists of 26B tokens. Indeed, the main text clarifies this, but training for 26B OWT tokens is surprisingly little to validate the convergence properties of a pre-training method, considering the literature commonly trains for 5-10x longer.
> > - Q1, Q2: As I understand from the authors' response, the TCSM model based on a masking prior uses both the masked diffusion model (MDM) ELBO and sampling algorithm and only during training relies on the TCSM framework as a surrogate objective. If that is indeed the case, it needs to be clearly stated that TCSM serves only as a surrogate objective for existing diffusion approaches. However, it would also entail a weakening of the claim that "TCSM provides a unifying perspective that encompasses several existing discrete diffusion methods" (L268), since it does not unify existing approaches under one theory but only provides a surrogate objective that is versatile enough to be applied to different discrete diffusion settings.
> > - Q3: Assuming that $p^\theta\_{1|t}(\mathbf{x}\_1|\mathbf{x}\_1^{\neq i},\mathbf{x}\_t)=\prod\_j p^\theta\_{1|t}(x^j\_1|\mathbf{x}\_1^{\neq i},\mathbf{x}_t)$, we have $p^\theta\_{1|t}(x^j\_1|\mathbf{x}\_1^{\neq i},\mathbf{x}_t)=\delta\_{x^j,x\_1^j}$ if $i\neq j$. For $i=j$, $p^\theta\_{1|t}(x^j\_1|\mathbf{x}\_1^{\neq i},\mathbf{x}\_t)$ does not simplify without any further assumptions. It is unclear what the model probability is when we are given all but one token.
> > - Q4: Strictly speaking, the true denoising distribution $p_{1|t}$ (L127) is intractable, and if we can sample from it, we have already solved the problem of unconditionally denoising. Presumably, during training, we use the conditional denoising distribution $p_{1|t}(x | x_t, x_1)$, which is trivially equal to $\delta_{x, x_1}$. If this is indeed the intended claim, the dependence of $h$ on $x_1$ needs to be clearly stated.
> > - Q6: Unlike for continuous (Gaussian) diffusion, it does not make sense to distinguish between _data_ and _noise_ prediction for discrete diffusion, since there is no notion of "distance" between two (noisy) samples. As such, it is the first time I have heard the term "data-prediction" being used in the context of discrete diffusion, and I recommend removing the corresponding claim in L87, Col. 2 to avoid confusion.
> > - Q7: Even if high gradient variance causes slow convergence, which is likely true, the faster convergence observed in TCSM cannot be seen as evidence for lower gradient variance. Changing the writing appropriately should address this concern. In terms of using a teacher model, this cannot be seen as way to speed up pre-training, since pre-training assumes training a model from scratch. If the authors indeed intend to claim that using a parametric teacher model _speeds up_ pre-training, the training cost of the teacher model has to be taken into account. Instead, it is probably best framed as a form of distillation of a (non-diffusion) teacher model into a diffusion student model, which is also a valuable contribution.
> > - Q8: The updated equality is better. However, strictly speaking, the dependence on $\mathbf{x}_1^{\neq i}$ cannot trivially be dropped at this point.
> > - Q9: Thank you for the clarification, the proof seems sound now. Specifying the "mild regularity conditions" should further improve its rigor.
> > - Q12: I am willing to take the authors' word regarding the missing proof of Prop. 4.1. Regarding Prop. 3, given that $p_{1|t}$ is also unknown, its purpose remains unclear.
> >
> > To summarize, while some of my concerns on soundness have been addressed, some remain. Most prominently, the short training horizon of 26B tokens feels insufficient to make strong claims about convergence properties compared to baselines that are usually trained for much longer. Further, the intractability of $p\_{1|t}(\mathbf{x}\_1|\mathbf{x}\_t)$ is a major theoretical issue if this distribution is to be used as the proposal distribution during pre-training. Many remaining theoretical soundness concerns can be remedied by clarifying the role of TCSM as a surrogate objective on top of existing discrete diffusion models. This would remove the need for, and in fact make unnecessary, the theoretical justifications, since empirical validation is all the justification needed to motivate a surrogate objective. Releasing reproducible training code would also bolster the paper in this regard. Besides soundness, framing and writing remain significant weaknesses that can be improved in future versions.

---

> > > ### Author Response · Authors · 2025-04-09
> > >
> > > We appreciate the reviewer's follow-up.
> > >
> > > To summarize:
> > > - Q[9, 11, 12, 13, 14, 15] are resolved. We'll incorporate suggestions in revision.
> > > - Remaining concerns addressed below.
> > >
> > > # Q4 Role of $p_{1|t}$
> > >
> > > We appreciate comments on $h_{1|t} = p_{1|t}$ and clarify.
> > >
> > > >true denoising distribution $p_{1|t}$ is intractable, and if we can sample from it, we have already solved the problem of unconditionally denoising.
> > >
> > > This interpretation is incorrect. As explicitly defined in Sec 2, Line 107, $p_{1|t}$ is $p_{1|t}(x_1 | x_t) = \frac{p_1(x_1)p_{t|1}(x_t | x_1)}{p_t(x_t)}$, where $p_t(x_t) = \mathbb{E}_{p_1(x_1)} p_{t|1}(x_t | x_1)$
> > >
> > > Though $p_{1|t}$ is unknown, we can sample from it using the training dataset and probability path $p_{t|1}$. TCSM training involves sampling from $t \sim \omega(t), x_t \sim p_t(x_t), x_1 \sim h_{1|t}(x_1|x_t)$, specifically when $h_{1|t}=p_{1|t}$, the process is as follows:
> > > 1. $t \sim \omega(t)$
> > > 2. Sample $x_1$ from dataset
> > > 3. Sample $x_t$ from $p_{t|1}(x_t | x_1)$
> > > 4. Train with TCSM
> > >
> > > >during training we use $p_{1|t}(x|x_t, x_1)$, which is $\delta_{x, x_1}$
> > >
> > > >dependence of $h$ on $x_1$ needs to be stated
> > >
> > > While $h_{1|t}$ is $p_{1|t}$ during pre-training, its definition doesn't depend on $x_1$. For example we use $h_{1|t}=p_{1|t}^{pre}$ in post-training, differing from original data distribution (see Alg 1, App E.2).
> > >
> > > In our understanding of the notation $p_{1|t}(x|x_t, x_1)$, it seems you are borrowing the notation of the posterior $p(x_{t-1}|x_t, x_1)$ in denoising diffusion models which shares similar form. If that's the case, we would like to clarify that:
> > > - $h_{1|t}$ does not play the same role as $p(x_{t-1}|x_t, x_1)$, which is only relevant in **denoising score matching**, leading to conditional denoising score matching $\nabla_{x_t} \log p_{t|1}(x_t|x_1)$. Our TCSM, however, is based on **target score matching**, which is based on the score $\nabla_{x_1} \log p_{1}(x_1)$. We have discussion in Sec 4 Lines 209-219, and Table 1.
> > >
> > > >intractability of $p_{1|t}(x_1|x_t)$ is a major theoretical issue
> > >
> > > We hope our explanation clarifies that $h_{1|t}=p_{1|t}$ does not pose a theoretical or practical issue.
> > >
> > > # Q5
> > > >26B tokens is surprisingly little to validate the convergence properties
> > >
> > > - As mentioned in our previous response (Q2 vK3t), the experiment in Fig 1 is designed to evaluate **sample efficiency** during pre-training, not full convergence. Training for 26B tokens demonstrates these efficiency gains. The comparison in Fig 1 is fair as all methods used the same 26B token budget, similar to the 33B tokens in the MDLM paper.
> > > Validation loss curves in Fig 4 further support our approach's improved training efficiency.
> > >
> > > - For evaluations results **after full-convergence**, we have results presented in Table 3 and Table 6, where we compare against baseline models all after full convergence.
> > >
> > > # Q3
> > > We'd like to respectfully clarify that the assumption you brought up is not the one we make, either in our paper or rebuttal.
> > >
> > > To elucidate the derivation of Eq 10, we restate it for clarity. The derivation is based on the conditional independence assumption used in prior works on denoising models, that the model can be factorized as $p^{\theta}\_{1|t}(x_1 | x_t) = \prod\_{i=1}^{L} p^{\theta}\_{1|t}(x^i\_1 | x\_t)$. This indicates that $x_1$, when conditioned on the noisy state $x_t$, is independent of tokens $x_1^{\neq i}$. Therefore, we have $p^{\theta}\_{1|t}(x\_1 | x\_1^{\neq i}, x\_t) = p^{\theta}\_{1|t}(x\_1 | x\_t)$. By substituting this simplification into Eq 9, we directly obtain Eq 10
> > >
> > > # Q6
> > > >it does not make sense to distinguish between data and noise prediction for discrete diffusion
> > >
> > > >first time hearing "data-prediction" in discrete diffusion, suggest removal to avoid confusion
> > >
> > > The distinction between data and noise prediction is well established in the foundational work we built on.
> > >
> > > Discrete Flow Matching (DFM) (2407.15595) explicitly distinguishes these notions (see abstract and Table 1). The equivalence between noise and data prediction only holds for masked diffusion. Since both DFM and our TCSM are designed for general priors, "data-prediction" is necessary to describe one of the valid prediction targets, aligning with the DFM framework.
> > >
> > > # Q1 Q2
> > >
> > > We are glad concerns about likelihood and sampling are resolved. We clarify why TCSM is beyond a surrogate. By choosing different source distributions (masking, uniform, etc.) and divergence measures (KL, Gen KL, f-divergence), our framework can derive objective functions for various discrete diffusion models. There is no existing framework that TCSM can be considered a surrogate of. See Q4 in vK3t for details.
> > >
> > > # Q7
> > > Thanks for the suggestion. We will revise to align more with distillation.
> > >
> > > We hope this clarifies. If so, we kindly ask the reviewer to update review reflecting most concerns are addressed, and specify if remaining issues are about flaws in our own methods or just background/notation when discussing with AC and reviewers.

---

### Official Review · Reviewer_GgYJ · 2025-03-13

**Overall Recommendation:** 4

**Summary:**

This work presents a new paradigm for modeling discrete data, titled Target Concrete Score Matching (TCSM). Unlike recent works that match a denoising concrete score, starting from an objective inspired by discrete flow matching, the authors propose to model the concrete score of ‘clean’ (i.e., target) data. The framework of TCSM is claimed to admit other recent discrete diffusion / flow matching papers as special cases. After deriving tractable Monte Carlo formulations of the TCSM objective, the authors conduct experiments demonstrating the competitive language modeling capabilities of TCSM-trained models, as well as their amenability to post-training adaptations.

## update after rebuttal
During the rebuttal periods the authors addressed my big concern on whether the results represented "fair comparisons". I therefore increased my score to 4.

I believe the authors should commit to adding the missing proof for Prop 4.1 and make the edits from the rebuttal period which are important for clarity and reproducibiliyt

**Claims And Evidence:**

The claims are well supported by the experiments, however, as noted below I have some questions about the details of the experiments and believe more information should be provided in the manuscript.

**Essential References Not Discussed:**

Throughout the paper, I would recommend adding Ou et al., 2024 [1] to the reference lists that include MDLM and MD4, as Ou et al was concurrent with these works and derives a similar formulation.

The authors should cite works such as [2], [3], [4] for discussing the use of the taylor approx. in the AR parameterization/distillation

Additionally, references for the text8 [5] and OWT [6]  datasets should be added.

Finally, it would be helpful to add references to the first sentence in Section 4.2.

---

[1] Ou, Jingyang, et al. "Your absorbing discrete diffusion secretly models the conditional distributions of clean data." arXiv preprint arXiv:2406.03736 (2024).

[2] Vignac, Clement, et al. "Digress: Discrete denoising diffusion for graph generation." arXiv preprint arXiv:2209.14734 (2022).

[3] Nisonoff, Hunter, et al. "Unlocking guidance for discrete state-space diffusion and flow models." arXiv preprint arXiv:2406.01572 (2024).

[4] Schiff, Yair, et al. "Simple Guidance Mechanisms for Discrete Diffusion Models." arXiv preprint arXiv:2412.10193 (2024).

[5] Matt Mahoney. Text8 dataset, 2011. URL http://mattmahoney.net/dc/textdata.

[6] Aaron Gokaslan, Vanya Cohen, Ellie Pavlick, and Stefanie Tellex. Openwebtext corpus. http:
//Skylion007.github.io/OpenWebTextCorpus, 2019.

**Experimental Designs Or Analyses:**

My biggest questions / concerns are regarding the main NLP experiments. Specifically, I do not believe that the current manuscript contains enough information for reproducibility:
-  The authors should provide details on how perplexity (or its lower bounds) were computed for TCSM methods.
- More details for Figure 1 and Table 4 should be provided, i.e, model sizes and hyperparameters for training.
    - More specifically, for Section 4.2, it is unclear whether the TCSM models used in Figure 1 are randomly initialized or come from pre-trained BERT/AR/Hollow transformer models. If the latter, the authors should more explicitly state the details for the pre-trained model, and moreover should clarify why the comparisons to the baselines are still “fair” if pre-trained models are used (i.e., TCSM would be receiving more training / FLOPs budget in that case).
- All of the fine-tuning / post-training experiments should have more explicit details about the specific model used for initialization and the hyperparameter setup of the fine-tuning

**Methods And Evaluation Criteria:**

Yes, benchmarks and tasks are well suited to the tasks at hand.

**Other Comments Or Suggestions:**

**Suggestions**
1. It would be useful to have algorithms for training and generation with TCSM.
2. Throughout the text, I think you can consider dropping the time subscripts when using the variable $y$. To me, this represents any alternative from the vocab and so the subscript is unnecessary.
3. It would be helpful to make explicit what the “mild regularity assumptions” on the divergence are.
4. Citations should be added to Tables 1 and 2.
5. In Def 3.1, perhaps the notation for $\mathcal{N}$ on Line 105-RHS should be changed to: $\mathcal{N}(\mathbf{x}) = \{\mathbf{x\}_{n\_1}, \ldots, \mathbf{x}\_{n\_{K\_{\mathbf{x}}}} \}$.
6. On Line 198 LHS, one can surmise what is meant by $[c]_{\mathbf{y}_1}$ but this notation is not strictly well-defined in the text.
7. Consider adding an explicit definition for the cross entropy notation $\mathbb{H}$ on line 211 RHS.
8. I know it is technically “concurrent” work according to ICLR guidelines, but it might be nice to add UDLM (Schiff et al., 2024) to Tables 2 and 4.
9. The best values per column should be bolded in Table 3.

**Typos / Formatting issues**
1. Figure titles, legends, axes labels, and tickmarks are quite hard to read. Should be made bigger / more legible. Additionally in Figures 2 and 3, the dotted lines, the shading, and the meaning of the legend text (e.g. TCSM xx) are not explained anywhere.
2. There are several places where $p_{1|t}^{\theta}$ is replaced with $p_{t|1}^{\theta}$, e.g.,  Line 78-RHS, Line84-RHS, Line89-RHS, Line 676…
4. Several places have minor formatting issues after equations where the commas appear on a new line, e.g., after Equations (1) and (3)
5. References in the main text to Eq (12) should instead refer to Eq (1).
6. Line 134 LHS should say “ensures” instead of “ensuring”.
7. On Lines 115-116 RHS I understand what is meant by describing the matrix representation of the concrete score, but I think this part of the sentence is a bit confusing “by replicating the original sequence $\mathbf{x}$ $L$ times.” It’s not the values of $\mathbf{x}$ that are actually used, it’s just the right shape? Is that what is trying to be conveyed here?
8. Line 116 RHS is missing  "$\mid \mathbf{x}_t$" in the vector expression for each column $i$ of the concrete score.
9. In the first two rows of Table 2, should “KL” instead be “GKL” or does the caption need to be updated?
10. The numbering for Tables 3 and 4 should be switched.
11. The actual parameterization of TCSM in Tables 3 and 4 should be indicated (i.e. BERT, vs. AR, vs Hollow).

**Other Strengths And Weaknesses:**

**Strengths**

Overall, I think this is innovative work and find the new directions of post-training and various parameterizations compelling. Additionally, if the concerns / confusions I have about the experimental details are resolved, this work would represent a very big step forward towards closing the gap to the dominant AR approach for language modeling. My current score below reflects this and I would be more than happy to increase it if the concerns raised above and detailed below are addressed.

**Weaknesses**

1. The authors should provide more explicit derivations of equivalence between TCSM and the works listed in Table 2.
2. The authors should clarify whether the comparison between TCSM and other models in Figure 1 and Tables 3 and 4 is “fair”. Was TCSM initialized from a pre-trained model here? If so, which one and additionally is the training budget comparable to baselines after accounting for the train of the initialization model.
3. The presentation of an “experiments” section following Section 4.1 seems out of place, since my understanding is that the actual models used require the parameterization introduced in Section 4.2.
4. An AR baseline should be added to Figure 1.
5. Are the gains in Table 6 just a product of additional training? Or are the baselines indeed comparable to fine-tuned TCSM?
6. The presentation of distillation in Section 5.4 as a post-training technique is somewhat confusing, considering this is the implementation of the AR parameterization of $p_1$ in Section 4.2, if I understand correctly. It seems the same methodology is being “pitched” for both pre and post training.
7. More experimental details are required to ensure reproducible results. Specifically, for all experiments the specific probability path schedule (I am assuming linear is used) should be mentioned as well as other missing model size, training length, and hyperparameter details (many / all of these can go in the appendix).
8. All the post-/fine-tuning experimental details should provide more information about the pre-trained model being used.

**Questions For Authors:**

1. How would one use the parameterizations in Section 4.2 for generation? For example, the pre-trained BERT model seems to require ground truth data to compute  $p_{1|t}(x_1^i\mid \mathbf{x}_1^{\neq i})$?
2. Are the parameterizations in Section 4.2 indeed “Pre-trained” (as indicated in the gray paragraph header in Line 256 RHS) or are they initialized from scratch?
3. How expensive are the parameterizations in Section 4.2? Seems like it requires $L$ forward passes for each sentence when using BERT / Hollow models?
4. Is it obvious that the divergence from Eq (3) can be decomposed into the component parts, as in Equations (4) and (5)? Aren’t there some assumptions being made about the divergence operator here that render these equivalent? If so, these should be explicitly stated.
5. Why is it important to define $h(\mathbf{x}_1 \mid \mathbf{x}_t)$? It seems that in all places, this is assumed to just be equal to $p(\mathbf{x}_1 \mid \mathbf{x}_t)$.
6. In Definition 3.1, why is it possible to have a “multi”set? Is this ever used?
7. In Proposition 4.1, is it correct to use the term “pseudo-likelihood" for $\ell^{i}_{pseudo}$ since it also has the added $1 / (V\cdot p)$ term?
8. For Figure 1, why do the authors only report PPL after 26B tokens. I believe this is only ~10% of the training of the baseline models? Additionally, were SEDD and MDLM re-trained for this figure? If so, that should be clarified.
9. In the Density Ration Parameterization of Section 5.1, wouldn’t this cause $\theta$ to not be updated since the $p_{ref}$ in the parameterization of $r_{1|t}$ would cancel out with that in the first entry of the Bregman Divergence in Eq. 11?
10. What does “Reimpl.” mean in the TCSM BCE line of Table 6? What is being reimplemented? Additionally, where is the MDLM number taken from? This is different than the one reported in Table 2 of Sahoo et al. 2024.
11. Can the authors elaborate on what is meant by the differing methods of compute PPL noted in the Appendix (Liines 875-879)? Some notation/formulas explaining the differences would be useful here as well as an explicit mention of which baseline uses which methodology.

**Relation To Broader Scientific Literature:**

This work stands in contrast to the successful denoising score matching work of SEDD and is a generalization of other recent discrete diffusion and flow matching papers.

**Theoretical Claims:**

Proofs were checked. However, there is a proof missing in the manuscript. Currently the referenced proof for Proposition 4.1 is not found in the appendix. This should be remedied during the upcoming discussion period.

Additionally, I believe that the claim that TCSM unifies previous discrete diffusion and flow matching works under a single framework (i.e., Table 2) deserves more explicit derivation that demonstrates equivalence (could be added to the Appendix).

---

> ### Author Rebuttal · Authors · 2025-04-01
>
> We deeply appreciate the reviewer's thorough, high-quality, and insightful feedback. We'll address all suggestions and correct typos in our revision.
>
> We are particularly encouraged by the reviewer's positive assessment of our work as "innovative", "compelling", and "representing a very big step forward." We also appreciate their willingness to increase the score once the concerns on experimental details are addressed. To directly address these concerns, we have organized our response to focus first on the experimental details.
>
> Below we respond to each point (Q=Question, W=Weakness)
>
> # Experimental Details Concerns
>
> ## [W3, Q1, Q2, Q3]
> * Sec 4.1 introduces TCSM using **only data samples** from training dataset, same as prior works (SEDD, MD4)
> * Sec 4.2 presents an alternative method using auxiliary parametric models (AR, BERT, Hollow) to estimate Target Concrete Score (TCS) via $p_1(x_1^i|\mathbf{x}_1^{\neq i})$
>
> They describe orthogonal techniques within TCSM.
>
> ## W3
> Experiments following Sec 4.1 use only **data-only** methodology, without auxiliary models from Sec 4.2.
>
> ## Q1
> Parametric models in Sec 4.2 are used **only during training** to estimate TCS. They're **not needed for inference/generation**. After training, standard sampling techniques are used. (Please refer to Q1 of vK3t for more details).
>
> ## Q2
>
> Our procedure:
> 1. Initialize parametric model from scratch and train on same dataset
> 2. Use this trained model (frozen) to provide TCS estimates
>
> "Pre-trained" means auxiliary model is trained before being used in diffusion model training.
>
> ## Q3
> Yes, it requires additional forward passes during training. However:
> 1. The overhead applies only during training, not inference, preserving generation speed
> 2. We observed just a ~20% increase in training time
> 3. As Fig 4 shows, these models significantly accelerate convergence, requiring fewer total training steps (and FLOPs) to reach target performance compared to data-only baselines
>
>
> ## Q5
> * Pre-training: $h_{1|t}$ is true data posterior $p_{1|t}(x_1 | x_t)$
> * Post-training with reward: $h_{1|t}$ is pre-trained denoising model $p_{1|t}^{pre}$
>
> ## Q8
> Yes, we re-trained both baselines with same training budget as TCSM.
> Please refer to Q2 of vK3t for more details.
>
> ## Q10
> "Reimpl." refers to our re-implementation of EDLM NCE. As TCSM with BCE in Eq.11 is equivalent to EDLM's NCE loss and their code isn't public, we implemented it ourselves (detailed in App E.3, L1105-L1110).
>
> The MDLM number in Tab 6 comes from Table 2 in the EDLM paper. Thanks for noting this discrepancy. We'll update with the original number from Sahoo et al. 2024.
>
> ## Q11
> Like SEDD Sec 5.2.3, we divide text into fixed-length, non-overlapping segments. AR models use sliding window approach.
>
>
> ## W2
> Comparisons are fair. All TCSM models were trained from scratch on identical datasets and evaluated under the same conditions using the same validation set. TCSM was not initialized from any pre-trained model trained on any extra data.
>
> Baseline models were trained to convergence across multiple epochs, TCSM's gains come from our density ratio parameterization (see W5).
>
> ## W4
> We implemented AR baseline with 26B tokens, achieving 22.51 PPL. We'll add to Fig 1.
>
> ## W5
> Tab 6 gains come from our density ratio model $r_{1|t}^{\theta}$, not extra training. This model captures token dependencies other approaches miss with conditional independence assumption.
>
> ## W6
> Please refer to Q3 in vK3t.
>
> ## W7, W8
> All experiments use linear probability paths. We built on MD4's codebase, maintaining identical architectures, sizes, training steps, and optimizers for fair comparison. Code will be released for reproducibility.
>
>
> # Other Concerns
>
> ## Q4
> Eq.4 and 5 are indeed specific instances of the general Eq.3, simplified by using the 1-Hamming neighborhood.
> This adds no additional assumptions on the divergence operator, as the goal of matching model and TCS can use any divergence measure. The 1-Hamming structure only defines what scores we match, not how we measure their differences through the divergence.
>
> ## Q6, Formatting issues 6
> 'Multiset' in Def 3.1 refers to how we construct the concrete score matrix (L114, RHS) by duplicating the original sequence. This enables transition from score space to normalized probability distribution space (Sec 3, L185-L219, 'Target Concrete Score' paragraph). Our approach parallels concepts in prior work (Proposition 1, Eq. 9 in MD4).
>
> ## Q7
> We agree "pseudo-likelihood" in Prop 4.1 isn't used in exact sense. This term was chosen because minimizing $l_{pseudo}$ directly corresponds to maximizing standard pseudo-likelihood. We'll revise.
>
>
> ## Q9
> As in Table 5, when implementing Eq. 11, we treat samples from $p^{ref}$ as "negative samples" for training model $\theta$. We don't backpropagate through sampling from $p^{ref}$.
>
> ## W1
> Please refer to Q4 of vK3t.
>
> ## Extra References
> Thank you for pointing out the extra references. We will add all suggested references.

---

> > ### Comment · Reviewer_GgYJ · 2025-04-03
> >
> > Thank you for the detailed response. I think I originally misunderstood the role of pre-trained models in Section 4.2 parameterizations. Can the authors clarify whether the reported validation ELBO numbers in Table 4 or Figure 1 rely on these external pre-trained models during `eval` or is the validation loop conducted using only TCSM without the pre-trained models?
> >
> > Overall, I think several of the reviewers have noted issues with clarity. I believe that if these are addressed the paper would be much stronger. For me, in particular the two main opportunity areas for improvement are:
> > - Algorithms / explanation for training, generation, and ELBO calculation would be very useful.
> > - I had some difficulty understanding how the experiments were conducted: which models / parameterizations were used for each experiment.

---

> > > ### Author Response · Authors · 2025-04-09
> > >
> > > We sincerely thank the reviewer for their time and insightful comments, which help us improve the work.
> > > # Q1
> > > >Can the authors clarify whether the reported validation ELBO numbers in Table 4 or Figure 1 rely on these external pre-trained models during eval?
> > >
> > > >Is the validation loop conducted using only TCSM without the pre-trained models?
> > >
> > > The validation ELBO results in Table 4 and Figure 1 **do not** use external pre-trained models during evaluation. These models are only used during *training* to help estimate the target concrete score (Section 4.2). Validation relies solely on TCSM without external models.
> > >
> > > # Q2
> > > > Algorithms / explanation for training, generation, and ELBO calculation would be very useful.
> > >
> > > We agree. Below, we outline the algorithms for pre-training (using the $\ell_{score}$ loss from Eq. 10 with a masked prior as an illustrative example) and generation.
> > >
> > > #### Algorithm for Training
> > >
> > > #### Training Algorithm
> > > **Input:** Dataset $\mathcal{D} = \{\mathbf{x}_1\}$, Denoising model $p_{1|t}^{\theta}$, Noise schedule $\{\alpha_t\}_{t}$
> > > 1. Sample $t \sim \omega(t) \propto \frac{d\alpha_t/dt}{1-\alpha_t}$
> > > 2. Sample $\mathbf{x}_1$ from dataset $\mathcal{D}$
> > > 3. Sample $\mathbf{x}_t$ from $p_{t|1}(\mathbf{x}_t | \mathbf{x}_1)$
> > > 4. Compute loss using $\ell_{score}$ (Eq. 10)
> > > 5. Update parameters $\theta$ via gradient descent
> > >
> > > #### Generation Algorithm
> > > 1. **Initialize:** Set time steps $\{t(i)\}_{i=0}^{T} \leftarrow \text{discretize}([0, 1])$, initialize $\mathbf{x}_{t(T)}$ as $N$ mask tokens $\mathbf{m}$.
> > > 2. **for** $i = T, T-1, \ldots, 1$ **do**
> > >    - Set $t \leftarrow t(i)$ and $s \leftarrow t(i-1)$
> > >    - Predict distribution: $p_{1|t}^{\theta}(\mathbf{x}_1 | \mathbf{x}_t) = \text{Cat}(\mu(\mathbf{x}_t, t))$
> > >    - Update elements: $\text{for } n \in [N]$, $\text{ if } x_t^n = \mathbf{m}$, sample $x_s^n \sim \text{Cat}\left(\frac{\alpha_s - \alpha_t}{1 - \alpha_t} \mu^n(\mathbf{x}_t, t) + \frac{1 - \alpha_s}{1 - \alpha_t} \mathbf{e}_m\right)$; otherwise, keep $x_s^n \leftarrow x_t^n$.
> > > 3. **Output:** **return** final sequence $\mathbf{x}_0$.
> > >
> > > For ELBO calculation, please see our response to Q1 wm3g.
> > >
> > > Post-training procedures are detailed in:
> > > - Algorithm 1 (Appendix E.2)
> > > - Algorithms 2 and 3 (Appendix F.1) for reward-based fine-tuning
> > > - Algorithm 4 (Appendix G.1) for preference-based fine-tuning
> > >
> > > # Q3
> > > > which models / parameterizations were used for each experiment.
> > >
> > > We provide a detailed table below showing models and parameterizations for all results in the paper.
> > >
> > > | Model | Experiment | Prior | Denoising Model Parameterization | Proposal distribution $h$ | Training objective |
> > > |-------|------------|-------|----------------------------------|---------------------------|-------------------|
> > > | TCSM Uniform $L_{score}$ (Sec. 4.2) | Table 4 | Uniform | Factorized $p_{1\|t}^{\theta}(\mathbf{x}_1 \| \mathbf{x}_t) = \prod_{i=1}^L p_{1\|t}^{\theta}(x_{1}^i \| \mathbf{x}_{t})$ | $p_{1\|t}$ | Eq. 10 |
> > > | TCSM Uniform $L_{distrib}$ (Sec. 4.2) | Table 4 | Uniform | Factorized | $p_{1\|t}$ | Eq. 9 |
> > > | TCSM Absorb $L_{score}$ (Sec. 4.2) | Table 4 | Mask | Factorized | $p_{1\|t}$ | Eq. 10 |
> > > | TCSM Absorb $L_{distrib}$ (Sec. 4.2) | Table 4 | Mask | Factorized | $p_{1\|t}$ | Eq. 9 |
> > > | TCSM Absorb $L_{distrib}$ (Sec. 5.1) | Table 4 | Mask | Density ratio parameterization $p_{1\|t}^{\theta}(\mathbf{x}_1 \| \mathbf{x}_t) \propto p_{1\|t}^{\text{ref}}(\mathbf{x}_1 \| \mathbf{x}_t) r_{1\|t}^{\theta}(\mathbf{x}_1 \| \mathbf{x}_t)$  | $p_{1\|t}^{pre}$ | Eq. 9 (pre-training), Table 5 Gen KL (post-training) |
> > > | TCSM $L_{score}$ (Sec. 4.2) | Table 3 | Uniform | Factorized | $p_{1\|t}$ | Eq. 10 |
> > > | TCSM $L_{distrib}$ (Sec. 4.2) | Table 3 | Uniform | Factorized | $p_{1\|t}$ | Eq. 9 |
> > > | TCSM $L_{distrib}$ (Sec. 4.2) | Table 3 | Mask | Factorized | $p_{1\|t}$ | Eq. 9 |
> > > | TCSM $L_{distrib}$ (Sec. 5.1) | Table 3 | Mask | Density ratio parameterization $p_{1\|t}^{\theta}(\mathbf{x}_1 \| \mathbf{x}_t) \propto p_{1\|t}^{\text{ref}}(\mathbf{x}_1 \| \mathbf{x}_t) r_{1\|t}^{\theta}(\mathbf{x}_1 \| \mathbf{x}_t)$  | $p_{1\|t}^{pre}$ | Eq. 9 (pre-training), Table 5 Gen KL (post-training) |
> > > | TCSM-Bert | Figure 1 | Mask | Factorized | $p_{1\|t}$ | Eq. 9 |
> > > | TCSM-AR | Figure 1 | Mask | Factorized | $p_{1\|t}$ | Eq. 9 |
> > > | TCSM-Hollow | Figure 1 | Mask | Factorized | $p_{1\|t}$ | Eq. 9 |
> > > | TCSM BCE (Reimpl.) | Table 6 | Mask | Density ratio (same as above) | $p_{1\|t}^{pre}$ | Table 5 BCE (ii) |
> > > | TCSM LSIF | Table 6 | Mask | Density ratio (same as above) | $p_{1\|t}^{pre}$ | Table 5 LSIF (ii) |
> > > | TCSM Gen KL | Table 6 | Mask | Density ratio (same as above) | $p_{1\|t}^{pre}$ | Table 5 Gen KL (ii) |
> > > | TCSM | Figure 5 | Uniform | $p_{1\|t}^\theta$ | $p_{1\|t}^{pre}$ | Sec. 5.2 L 373 |
> > >
> > >
> > > We hope these clarifications and added details effectively address the reviewer's concerns. We believe these clarifications further strengthen our paper, and we would be grateful if the reviewer would take these points into consideration in their updated review.

---

### Official Review · Reviewer_vK3t · 2025-03-14

**Overall Recommendation:** 4

**Summary:**

Recent works have proposed various diffusion modeling frameworks for discrete data, this paper proposes target concrete score matching, a framework that unifies various discrete diffusion approaches, such as discrete flow matching, masked diffusion language modeling, etc. The unified framework allows for using various model parameterizations and inference/noising processes with either uniform or absorbing terminal distributions in a single framework. The paper also demonstrates post-training methods such as sampling reward tilted models using RL or preference fine-tuning, and distillation from auto-regressive models can be encompassed in the TCSM framework.

**Claims And Evidence:**

See below.

**Essential References Not Discussed:**

Sampling from discrete diffusion models is an active area of research as well as relevant for any unifying framework. Recent works have proposed various forms of sampling schemes, such as predictor correctors (zhao et al 2024), or gradient-based guidance (shi et al 2024).

Zhao, Yixiu, et al. "Informed correctors for discrete diffusion models." arXiv preprint arXiv:2407.21243 (2024).

Shi, Jiaxin, et al. "Simplified and generalized masked diffusion for discrete data." Advances in neural information processing systems 37 (2024): 103131-103167.

**Experimental Designs Or Analyses:**

The experiments contain the relevant diffusion baselines, as well as experimenting with different objectives and choices in the TCSM framework. However, the paper could benefit from a discussion of the experimental results. For instance, for several baselines considered the inference process, objectives and model parameterization are similar (for instance, in table 3), therefore a discussion explaining the results or guidance on which component would benefit the paper.

**Methods And Evaluation Criteria:**

The paper enables studying various existing approaches, with different inference processes, model parameterization and objectives in a single framework.

**Other Comments Or Suggestions:**

None

**Other Strengths And Weaknesses:**

Strengths:
1. a single framework to study multiple inference processes, model parameterizations, divergences is appealing since discrete diffusion models have shown promising performance. TCSM allows for making modeling choices without having to manually derive an objective or parameterization for each different choice.

Weaknesses:
1. A discussion on sampling methods from masked diffusion models, including correctors (Zhao et al 2024), re-masking, etc is missing.


Zhao, Yixiu, et al. "Informed correctors for discrete diffusion models." arXiv preprint arXiv:2407.21243 (2024).

Shi, Jiaxin, et al. "Simplified and generalized masked diffusion for discrete data." Advances in neural information processing systems 37 (2024): 103131-103167.

**Questions For Authors:**

None

**Relation To Broader Scientific Literature:**

The authors consider several discrete space diffusion models and propose a unified framework for studying and making modeling choices.

**Theoretical Claims:**

The theoretical claims are justified and detailed and self-contained proofs are provided in the appendix.

---

> ### Author Rebuttal · Authors · 2025-04-01
>
> We sincerely thank the reviewer for their positive review and valuable feedback. We greatly appreciate the opportunity to address the questions and provide additional clarifications about our work. In the following sections, we carefully respond to each point raised by the reviewer, quoting the relevant comments to ensure clear and direct responses. We have also included additional explanations that we believe will help further clarify our approach and results. Please don't hesitate to let us know if you have any follow-up questions or need further details on any aspect of our work.
>
> # Experimental Details Clarifications
>
> > the paper could benefit from a discussion of the experimental results.
>
> We agree that more discussion of the experimental results would be helpful. We layout following additional details on the implementation details and experimental results.
>
>
> ## Q1 Sampling
> > A discussion on sampling methods from masked diffusion models is missing
>
> We agree that including explicit introduction of sampling methods used in this work would be helpful.
> TCSM's flexibility allows it to work with any diffusion model parameterization, compatible with standard sampling techniques like ancestral sampling and reverse CTMC simulation via Euler method.
>
> In our experiments:
> * Masked diffusion models (Sec 5.2, Fig 3): Used ancestral sampling
> * Uniform diffusion models (Fig 5): Used reverse CTMC simulation with Euler method
>
> These are standard approaches briefly noted in App A. We'll provide more details in the revision.
>
>
> ## Q2 Fig 1 evaluation
> Fig 1 shows how parametric models (Sec 4.2) enhance sample efficiency during training. We aimed to test if this approach accelerates convergence vs data-only baselines. To clearly show performance differences, we deliberately restricted all models to 26 billion tokens, creating a controlled comparison within this limited training budget.
> In contrast, the results reported by Sahoo et al. (23.21 for MDLM, 24.10 for SEDD) baseline were obtained using full training data. Consequently, the PPL scores in our Fig 1 are not directly comparable to those specific results from the original papers.
> For fair comparisons under identical conditions (with full training convergence), see Tab 3 and 6.
>
>
>
> # Further Clarifications
>
> ## Q3 AR to Diffusion Distillation
> We placed distillation in post-training to align with LLM community where distillation is typically a post-training technique.
>
> Distillation section highlights a key TCSM advantage: it's the only discrete diffusion framework enabling effective AR-to-diffusion distillation—valuable given today's many pre-trained LLMs. We developed novel techniques for this process (Top-K, first-order approximation) that enhance scalability beyond Sec 4.2.
>
> ## Q4 Table 2 details
>
> While the key connections summarized in Table 2 are currently present in the manuscript, they are distributed across different sections. For instance, we establish that our simplified $\ell_{distrib}$ objective (Eq. 9) corresponds to the standard cross-entropy loss employed by MD4/MDLM/DFM. Furthermore, the relationship with EDLM (2410.21357) is detailed in App E.3 (L. 1122), linking their objective (Eq. 10) to the BCE-based Bregman divergence shown in Table 2 (row BCE (ii)).
>
> To improve readability, we will add a consolidated paragraph in the revised manuscript. This new paragraph will explicitly outline these connections and clearly explain the significance and utility of Table 2.
>
> ## Q5 SEDD, SDDM comparison
> SEDD was included as a baseline in all language modeling experiments, with results shown in Tables 3, 4 and Fig 1.
>
> For SDDM, we followed prior discrete diffusion studies by excluding it, as its focus is image generation. Comparable language modeling results and official code were unavailable, preventing direct comparison.
>
> We selected DFM for Sec 2 due to its general framework covering various discrete diffusion approaches, including SEDD (score parameterization) and MD4/MDLM (mask source distribution). This makes DFM ideal for introducing core concepts before our specific contributions.
>
> ## Extra references
>
> We will make sure to include all the references pointed out by the reviewer.

---

> > ### Comment · Reviewer_vK3t · 2025-04-09
> >
> > After going through the author's rebuttal and response to other reviewers, I maintain my score.

---

### Decision · Program_Chairs · 2025-05-01

**Decision:**

Accept (poster)

**Comment:**

This paper proposes target concrete score matching methods for discrete diffusion models. The study covers pre-training and post-training to demonstrate the advantage of target concrete score matching.

The proposed score estimation and training pipeline is the major contribution of the paper. This is well-recognized as novel and important contributions by all the reviewers. Concerns of the paper center around the clarity of the presentation and completeness of the results. For example, the formal justification of Proposition 4.1 is missing. Rough edits around equations, where comma appears in the beginning of a line. Faster convergence of target concrete score matching using a parametric model for $p_1$ seems to be not fully justified and involves unfair comparison.

Given these debates, the recommendation should be considered as a conditional acceptance. The positive recommendation owes to the complete pipeline provided in the paper and relatively sufficient empirical results. Nonetheless, the aforementioned issues must be resolved before the next revision.